# FASTER GRADIENT-FREE METHODS FOR ESCAPING SADDLE POINTS

**Hualin Zhang [1], Bin Gu [1,2]**
[1]Nanjing University of Information Science & Technology [2]MBZUAI
{zhanghualin98, jsgubin}@gmail.com

## ABSTRACT

Escaping from saddle points has become an important research topic in non-convex optimization. In this paper, we study the case when calculations of explicit gradients are expensive or even infeasible, and only function values are accessible. Currently, there have two types of gradient-free (zeroth-order) methods based on random perturbation and negative curvature finding proposed to escape saddle points efficiently and converge to an $\epsilon$-approximate second-order stationary point. Nesterov's accelerated gradient descent (AGD) method can escape saddle points faster than gradient descent (GD) which have been verified in first-order algorithms. However, whether AGD could accelerate the gradient-free methods is still unstudied. To unfold this mystery, in this paper, we propose two accelerated variants for the two types of gradient-free methods of escaping saddle points. We show that our algorithms can find an $\epsilon$-approximate second-order stationary point with $\tilde{\mathcal{O}}(1/\epsilon^{1.75})$ iteration complexity and $\tilde{\mathcal{O}}(d/\epsilon^{1.75})$ oracle complexity, where $d$ is the problem dimension. Thus, our methods achieve a comparable convergence rate to their first-order counterparts and have smaller oracle complexity compared to prior derivative-free methods for finding second-order stationary points.

## 1 INTRODUCTION

Non-convex optimization has received increasing attention in recent years because lots of modern machine learning (ML) and deep learning (DL) tasks can be formulated as optimizing models with non-convex loss functions. In this paper, we consider non-convex optimization with the following general form:

$$\min_{\mathbf{x} \in \mathbb{R}^d} f(\mathbf{x}), \tag{1}$$

where $f(\mathbf{x})$ is differentiable and has Lipschitz continuous gradient and Hessian.

In this paper, we focus on situations when first-order information (gradient) is not always directly accessible. Many machine learning and deep learning applications often encounter settings where the calculation of explicit gradients is expensive or even infeasible, such as black-box adversarial attack on deep neural networks (Papernot et al., 2017; Madry et al., 2018; Chen et al., 2017; Bhagoji et al., 2018; Tu et al., 2019), policy search in reinforcement learning (Salimans et al., 2017; Choromanski et al., 2018; Jing et al., 2021), hyper-parameter optimization (Bergstra & Bengio, 2012). Therefore, zeroth-order optimization, which utilizes only the zeroth-order information (function value) to optimize the non-convex problem Eq. (1), has gained increasing attention in machine learning.

In general, the goal of a non-convex optimization problem Eq. (1) is to find an $\epsilon$-approximate first-order stationary point (FOSP, see Definition 3), since finding the global minimum is NP-hard. Gradient descent is proven to be an optimal first-order algorithm for finding an $\epsilon$-approximate FOSP of non-convex problem Eq. (1) under the gradient Lipschitz assumption (Carmon et al., 2020; 2021), which needs a gradient query complexity of $\Theta(\frac{1}{\epsilon^2})$. However, for non-convex functions, FOSPs can be local minima, global minima and saddle points. The ubiquity of saddle points makes high-dimensional non-convex optimization problems extremely difficult and will lead to highly suboptimal solutions (Jain et al., 2017; Sun et al., 2018). Therefore, many recent research works have focused on escaping saddle points and studying properties of converging to an $\epsilon$-approximate second-order stationary point (SOSP, see Definition 4) using first-order methods.

A recent line of work showed that first-order methods can efficiently escape saddle points and converge to SOSPs. Specifically, Jin et al. (2017) proposed the perturbed gradient descent (PGD) algorithm by adding uniform random perturbation into the standard gradient descent algorithm that can find an $\epsilon$-approximate SOSP in $\tilde{\mathcal{O}}(\log^4 d/\epsilon^2)$ gradient queries. Under the zeroth-order setting, Jin et al. (2018a) proposed a zeroth-order perturbed stochastic gradient descent (ZPSGD) method, which studied the power of Gaussian smoothing and stochastic perturbed gradient for finding local minima. The role of Gaussian smoothing is to reduce zeroth-order optimization to a stochastic first-order optimization of a Gaussian smoothed function of problem Eq. (1). They proved their method can find an $\epsilon$-approximate SOSP with a function query complexity of $\tilde{\mathcal{O}}\left(d^2/\epsilon^5\right)$. Vlatakis-Gkaragkounis et al. (2019) proposed the perturbed approximate gradient descent (PAGD) method using the forward difference of the coordinate-wise gradient estimators, which finds an $\epsilon$-approximate SOSP in $\tilde{\mathcal{O}}\left(d\log^4 d/\epsilon^2\right)$ function queries. Recently, Lucchi et al. (2021) proposed a random search power iteration (RSPI) method, which alternatively runs the random search step and zeroth-order power iteration step, and can find an $(\epsilon, \epsilon^{2/3})$-approximate SOSP ($\|\nabla f(\mathbf{x})\| \leq \epsilon$, $\lambda_{\min}(\nabla^2 f(\mathbf{x})) \geq -\epsilon^{2/3}$) in $\mathcal{O}(d \log d/\epsilon^{\frac{8}{3}})$ function queries. Zhang et al. (2022) proposed a zeroth-order gradient descent method with zeroth-order negative curvature finding that can find an $(\epsilon, \delta)$-approximate SOSP ($\|\nabla f(\mathbf{x})\| \leq \epsilon$, $\lambda_{\min}(\nabla^2 f(\mathbf{x})) \geq -\delta$) in $\mathcal{O}(\frac{d}{\epsilon^2} + \frac{d\log d}{\delta^{3.5}})$ function queries.

Table 1: Comparison of different zeroth-order methods for finding $\epsilon$-approximate second-order stationary points.

| Algorithm | Reference | Main Technique | Function Queries |
|---|---|---|---|
| ZPSGD | Jin et al. (2018a) | Random perturbation | $\tilde{\mathcal{O}}\left(\frac{d^2}{\epsilon^5}\right)$ |
| PAGD | Vlatakis-Gkaragkounis et al. (2019) | Random perturbation | $\mathcal{O}\left(\frac{d\log^4 d}{\epsilon^2}\right)$ |
| RSPI | Lucchi et al. (2021) | Negative curvature finding | $\mathcal{O}(\frac{d\log d}{\epsilon^{8/3}})$ * |
| ZO-GD-NCF | Zhang et al. (2022) | Negative curvature finding | $\mathcal{O}(\frac{d}{\epsilon^2} + \frac{d\log d}{\delta^{3.5}})$ ** |
| Algorithm 1 | **Theorem 1** | Random perturbation | $\mathcal{O}\left(\frac{d\log^6 d}{\epsilon^{7/4}}\right)$ |
| Algorithm 3 | **Theorem 2** | Negative curvature finding | $\mathcal{O}\left(\frac{d\log d}{\epsilon^{7/4}}\right)$ |

$*$ guarantees $(\epsilon, \epsilon^{2/3})$-approximate SOSPs; $**$ guarantees $(\epsilon, \delta)$-approximate SOSPs.

Although gradient descent has achieved an optimal convergence rate for finding FOSPs under gradient Lipschitz assumption, potential improvements are achievable under additional Hessian Lipschitz assumption (Carmon et al., 2021). Nesterov's AGD combined with some special mechanisms, has been proved to be able to find $\epsilon$-approximate FOSPs with less query complexity. Carmon et al. (2017) proposed a variant of Nesterov's AGD with a "convex until guilty" mechanism, which can find an $\epsilon$-approximate FOSP with gradient query complexity $\mathcal{O}(\frac{1}{\epsilon^{7/4}} \log \frac{1}{\epsilon})$. Recently, Li & Lin (2022) proposed a restarted accelerated gradient descent method that can find an $\epsilon$-approximate FOSP in gradient query complexity $\mathcal{O}(\frac{1}{\epsilon^{7/4}})$, which adds a restart mechanism to Nesterov's AGD.

On finding SOSPs, Nesterov's AGD is also proved to be more efficient than GD. Jin et al. (2018b) studied a variant of Nesterov's AGD named perturbed AGD, and proved that it can find an $\epsilon$-approximate SOSP in $\tilde{\mathcal{O}}(\log^6 d/\epsilon^{7/4})$ gradient queries. Their method added two algorithmic features to Nesterov's AGD: random perturbation and negative curvature exploitation, to ensure the monotonic decrease of the Hamiltonian function (see Eq. (4)). Allen-Zhu & Li (2018) proposed a first-order negative curvature finding framework named Neon2 that can find the most negative curvature direction efficiently. Combining Neon2 with CDHS method of Carmon et al. (2018) can find an $\epsilon$-approximate SOSPs in $\tilde{\mathcal{O}}(\log d/\epsilon^{7/4})$ gradient queries, which improved the complexity of perturbed AGD method by a factor of $\text{poly}(\log d)$ due to the use of negative curvature finding subroutine. Recently, Zhang & Li (2021) proposed a single-loop algorithm that also achieves the same function query complexity, which replaced the random perturbation step in perturbed AGD with accelerated negative curvature finding.

Given the advantages of Nesterov's AGD in finding SOSPs in first-order optimization, it is then natural to design AGD based zeroth-order methods for finding SOSPs with smaller function query complexity. To the best of our knowledge, it is still a vacancy in zeroth-order optimization.

**Contributions** The main contributions of this paper are summarized as follows,

- We study the complexity of two AGD based zeroth-order methods for finding $\epsilon$-approximate SOSPs. We first study a zeroth-order version of the perturbed AGD method (Algorithm 1) using the central finite difference version of the coordinate-wise gradient estimator, which can be proved to have a lower approximation error compared to its forward counterpart. The total function query complexity of Algorithm 1 for finding an $\epsilon$-approximate SOSP is $\tilde{\mathcal{O}}(d \log^6 d / \epsilon^{\frac{7}{4}})$.

- Due to the efficiency of the negative curvature finding for finding the most negative curvature direction near a saddle, we further study a zeroth-order version of the perturbed AGD with accelerated negative curvature finding subroutine (Algorithm 3), which uses the finite difference of the two coordinate-wise gradient estimators to approximate the Hessian-vector product. We show that Algorithm 3 can further improve the function query complexity of Algorithm 1 by a factor of $\mathrm{poly}(\log d)$.

- Finally, we conduct several empirical experiments to verify the efficiency and effectiveness of our methods in escaping saddle points.

## 2 PRELIMINARIES

### 2.1 NOTATIONS

Throughout this paper, we use bold uppercase letters $\mathbf{A}, \mathbf{B}$ to denote matrices and bold lowercase letters $\mathbf{x}, \mathbf{y}$ to denote vectors. We use $\| \cdot \|$ to denote the Euclidean norm of a vector and the spectral norm of a matrix. We use $\mathbb{B}_{\mathbf{x}}(r)$ to denote the $\ell_2$ ball with radius $r$ centered at point $\mathbf{x}$. We use $\tilde{\mathcal{O}}(\cdot)$ to hide absolute constants and log factors.

### 2.2 DEFINITIONS

**Definition 1.** *For a differentiable nonconvex function $f : \mathbb{R}^d \to \mathbb{R}$, $f$ is $\ell$-Lipschitz smooth if*
$$\forall \mathbf{x}, \mathbf{y} \in \mathbb{R}^d, \|\nabla f(\mathbf{x}) - \nabla f(\mathbf{y})\| \leq \ell \|\mathbf{x} - \mathbf{y}\|.$$

**Definition 2.** *For a twice differentiable nonconvex function $f : \mathbb{R}^d \to \mathbb{R}$, $f$ is $\rho$-Hessian Lipschitz if*
$$\forall \mathbf{x}, \mathbf{y} \in \mathbb{R}^d, \|\nabla^2 f(\mathbf{x}) - \nabla^2 f(\mathbf{y})\| \leq \rho \|\mathbf{x} - \mathbf{y}\|.$$

**Definition 3.** *For a differentiable function $f$, we say $\mathbf{x}$ is an $\epsilon$-approximate first-order stationary point if $\|\nabla f(\mathbf{x})\| \leq \epsilon$.*

**Definition 4.** *For a twice differentiable function $f$, we say $\mathbf{x}$ is an $\epsilon$-approximate second-order stationary point if*
$$\|\nabla f(\mathbf{x})\| \leq \epsilon \quad and \quad \lambda_{\min}(\nabla^2 f(\mathbf{x})) \geq -\sqrt{\rho \epsilon}.$$

### 2.3 ZEROTH-ORDER GRADIENT ESTIMATOR

In this subsection, we introduce a central difference coordinate-wise gradient estimator, which is widely studied in literature of zeroth-order optimization (Ji et al., 2019; Vlatakis-Gkaragkounis et al., 2019; Lucchi et al., 2021),

$$\hat{\nabla} f(\mathbf{x}) = \sum_{i=1}^{d} \frac{f(\mathbf{x} + \mu \mathbf{e}_i) - f(\mathbf{x} - \mu \mathbf{e}_i)}{2\mu} \mathbf{e}_i, \tag{2}$$

where $\mathbf{e}_i$ is the $i$-th standard basis vector with $1$ at its $i$-th coordinate and $0$ otherwise. When analyzing the approximation error of the above gradient estimator, previous work only exploited the smoothness property of the gradient of $f$, not the property of Hessian Lipschitz (which is a basic assumption for analyzing the second-order convergence properties). To fill this gap, we establish the following lemma,

**Lemma 1.** *For a twice differentiable function $f : \mathbb{R}^d \to \mathbb{R}$, assume that $f$ is $\rho$-Hessian Lipschitz, then for any given smoothing parameter $\mu$ and any $x \in \mathbb{R}^d$, we have*

$$\|\hat{\nabla} f(\mathbf{x}) - \nabla f(\mathbf{x})\|^2 \leq \frac{1}{36} \rho^2 d \mu^4.$$

Note that, under the Hessian Lipschitz assumption, the central difference has a lower approximation error than that of $\mathcal{O}(\ell^2 d \mu^2)$ error under the $\ell$-smooth assumption (Ji et al., 2019).

## 2.4 ZEROTH-ORDER HESSIAN-VECTOR PRODUCT ESTIMATOR

In this subsection, we show how to approximate the Hessian-vector product under the setting that we only have access to the zeroth-order information. By the Hessian Lipschitz property, it is easy to check that the Hessian-vector product $\nabla^2 f(\mathbf{x}) \cdot \mathbf{v}$ can be approximated by the difference of two gradients $\nabla f(\mathbf{x} + \mathbf{v}) - \nabla f(\mathbf{x})$ with approximation error up to $\mathcal{O}(\|\mathbf{v}\|^2)$ for some $\mathbf{v}$ with small magnitude. On the other hand, by Lemma 1, $\nabla f(\mathbf{x} + \mathbf{v}), \nabla f(\mathbf{x})$ can be approximated by the central difference coordinate-wise gradient estimator with high accuracy. Then we define the following zeroth-order Hessian-vector product estimator as follows, which was previously studied in (Ye et al., 2018; Lucchi et al., 2021; Zhang et al., 2022):

$$\mathcal{H}_f(\mathbf{x})\mathbf{v} = \hat{\nabla} f(\mathbf{x} + \mathbf{v}) - \hat{\nabla} f(\mathbf{x}) \tag{3}$$

$$= \sum_{i=1}^{d} \frac{f(\mathbf{x} + \mathbf{v} + \mu\mathbf{e}_i) - f(\mathbf{x} + \mathbf{v} - \mu\mathbf{e}_i)}{2\mu} \mathbf{e}_i - \sum_{i=1}^{d} \frac{f(\mathbf{x} + \mu\mathbf{e}_i) - f(\mathbf{x} - \mu\mathbf{e}_i)}{2\mu} \mathbf{e}_i$$

Above, the notation $\mathcal{H}_f(x)$ can be seen as the Hessian matrix of $f$ at point $\mathbf{x}$ with small perturbations and we don't need to know the explicit expression since we only need to study the approximation error of it, which is established in Lemma 2.

**Lemma 2** (Zhang et al. (2022)). *For a twice differentiable function $f : \mathbb{R}^d \to \mathbb{R}$, assume that $f$ is $\rho$-Hessian Lipschitz, then for any smoothing parameter $\mu$ and $\mathbf{x} \in \mathbb{R}^d$, we have*

$$\|\mathcal{H}_f(\mathbf{x})\mathbf{v} - \nabla^2 f(\mathbf{x})\mathbf{v}\| \leq \rho \left( \frac{\|\mathbf{v}\|^2}{2} + \frac{\sqrt{d}\mu^2}{3} \right).$$

## 2.5 HAMILTONIAN

The following function, which takes the form of Hamiltonian, was proposed by Jin et al. (2018b) to tackle the problem of monotonic decrease of the function value for the momentum-based algorithms in the nonconvex setting,

$$E_t = f(\mathbf{x}_t) + \frac{1}{2\eta}\|\mathbf{v}_t\|^2, \tag{4}$$

where $\mathbf{v}_t = \mathbf{x}_t - \mathbf{x}_{t-1}$ is the momentum.

## 3 ALGORITHM DESCRIPTION

In this section, we propose two novel Nesterov's accelerated method based algorithms that can escape saddle points and converge to an $\epsilon$-approximate SOSP using only zeroth-order oracles.

## 3.1 ZEROTH-ORDER PERTURBED ACCELERATED GRADIENT DESCENT

In this subsection, we introduce the zeroth-order perturbed accelerated gradient descent method in Algorithm 1. The algorithms consist of three parts: the random perturbation steps, the accelerated gradient descent steps and the negative curvature exploitation steps. The random perturbation step is called when the gradient is small and no perturbation is added over the past $\mathscr{T}$ iterations. Let $\kappa = \frac{\ell}{\sqrt{\rho\epsilon}}$, and set the parameters of Algorithm 1 as follows,

$$\eta = \frac{1}{4\ell}, \qquad \theta = \frac{1}{4\sqrt{\kappa}}, \qquad \gamma = \frac{\theta^2}{\eta},$$

$$s = \frac{\gamma}{4\rho}, \qquad \mathscr{T} = \sqrt{\kappa}\chi c, \qquad r = \eta\epsilon\chi^{-5}c^{-8}, \tag{5}$$

where c is constant and $\chi = \max\{1, \log \frac{d\ell\Delta_f}{\rho\epsilon\delta}\}$ with $\Delta_f := f(\mathbf{x}_0) - f(\mathbf{x}^*) < \infty$.

Since we only have access to the zeroth-order information, we can verify if a point $\mathbf{x}$ is an $\epsilon$-approximate FOSP by using the coordinate-wise gradient estimator based on the following fact:

**Algorithm 1** Zeroth-Order Perturbed Accelerated Gradient Descent

1: $\mathbf{v}_0 \leftarrow 0, t_{\text{perturb}} \leftarrow 0$
2: **for** $t = 0, 1, \ldots$ **do**
3:      **if** $\|\hat{\nabla} f(\mathbf{x}_t)\| \leq \frac{3}{4}\epsilon$ and $t - t_{\text{perturb}} > \mathscr{T}$ **then**
4:          $\mathbf{x}_t \leftarrow \mathbf{x}_t + \xi_t, \xi_t \sim \text{Unif}\left(\mathbb{B}_0(r)\right),$ $t_{\text{perturb}} \leftarrow t$
5:      $\mathbf{y}_t \leftarrow \mathbf{x}_t + (1 - \theta)\mathbf{v}_t$
6:      $\mathbf{x}_{t+1} \leftarrow \mathbf{y}_t - \eta\hat{\nabla} f(\mathbf{y}_t)$
7:      $\mathbf{v}_{t+1} \leftarrow \mathbf{x}_{t+1} - \mathbf{x}_t$
8:      **if** Eq. (6) holds **then**
9:          $(\mathbf{x}_{t+1}, \mathbf{v}_{t+1}) \leftarrow \text{NCE}(\mathbf{x}_t, \mathbf{v}_t, s)$

**Algorithm 2** Negative Curvature Exploitation $(\mathbf{x}_t, \mathbf{v}_t, s)$ (Jin et al., 2018b)

1: **if** $\|\mathbf{v}_t\| \geq s$ **then**
2:      $\mathbf{x}_{t+1} \leftarrow \mathbf{x}_t$
3: **else**
4:      $\delta = s \cdot \mathbf{v}_t / \|\mathbf{v}_t\|$
5:      $\mathbf{x}_{t+1} \leftarrow \arg\min_{\mathbf{x} \in \{\mathbf{x}_t + \delta, \mathbf{x}_t - \delta\}} f(\mathbf{x})$
**Return** $(\mathbf{x}_{t+1}, 0)$

**Proposition 1.** *Assume that $f$ is $\rho$-Hessian Lipschitz, with choice of the smoothing parameter $\mu$ in Eq. (2) such that $\mu \leq \sqrt{\frac{3\epsilon}{2\rho\sqrt{d}}}$, we can conclude that if $\|\hat{\nabla} f(\mathbf{x})\| \leq \frac{3\epsilon}{4}$, then we have $\|\nabla f(x)\| \leq \epsilon$, if $\|\hat{\nabla} f(\mathbf{x})\| > \frac{3\epsilon}{4}$, then we have $\|\nabla f(x)\| \geq \frac{\epsilon}{2}$.*

The proof of this proposition directly follows from Lemma 1. The random perturbation is uniformly randomly selected from the $\ell_2$-ball with radius $r$. The second part of the Algorithm 1 is the Nesterov's accelerated gradient descent steps with its gradients estimated by Eq. (2).

The negative curvature exploitation step is called when the following condition holds:

$$f(\mathbf{x}_t) \leq f(\mathbf{y}_t) + \left\langle \hat{\nabla} f(\mathbf{y}_t), \mathbf{x}_t - \mathbf{y}_t \right\rangle - \frac{\gamma}{2}\|\mathbf{y}_t - \mathbf{x}_t\|^2. \tag{6}$$

If this condition hold, then the function have an approximate large negative curvature between $\mathbf{x}_t$ and $\mathbf{y}_t$. In this case, the accelerated gradient step may not decrease the function value of the Hamiltonian. Then we call the negative curvature exploitation step to further decrease the Hamiltonian. Specifically, when Eq. (6) doesn't hold, we have the following lemma:

**Lemma 3.** *Assume that $f(\cdot)$ is $\ell$-smooth, $\rho$-Hessian Lipschitz and set the learning rate $\eta \leq \frac{1}{4\ell}, \theta \in [2\eta\gamma, \frac{1}{2}]$. Then, for each iteration $t$ where Eq. (6) does not holds, we have:*

$$E_{t+1} \leq E_t - \frac{\theta}{2\eta}\|\mathbf{v}_t\|^2 - \frac{\eta}{4}\|\nabla f(\mathbf{y}_t)\|^2 + \eta \cdot \frac{\rho^2 d\mu^4}{48}.$$

On the other hand, when Eq. (6) holds, *i.e.*, a negative curvature direction is observed, then we have the following lemma:

**Lemma 4.** *Assume that $f(\cdot)$ is $\ell$-smooth and $\rho$-Hessian Lipschitz. Then, for each iteration $t$ where Eq. (6) holds, we have:*

$$E_{t+1} \leq E_t - \min\left\{\frac{s^2}{2\eta}, \frac{1}{2}\gamma s^2 - \rho s^3 - \frac{\rho^2 d\mu^4}{9\gamma}\right\}.$$

**Remark 1.** *The results in Lemma 3 and 4 are similar to the ones in Jin et al. (2018b) while with additional system error terms induced by the smoothing parameter $\mu$. Lemma 3 and 4 together ensure the monotonic decrease of the Hamiltonian in each iteration as long as the smoothing parameter $\mu$ is sufficient small.*

Then we set $\mathscr{T} = \sqrt{\kappa}\chi c = \tilde{\Theta}(\sqrt{\kappa})$ and denote $\mathscr{E} := \sqrt{\frac{\epsilon^3}{\rho}}\chi^{-5}c^{-7} = \tilde{\Theta}(\sqrt{\frac{\epsilon^3}{\rho}})$. Based on Lemma 3 and Lemma 4, we can further prove that when the current approximate gradient is large, i.e., $\|\hat{\nabla} f(\mathbf{x}_t)\| \geq \frac{3\epsilon}{4}$ (or equivalently, $\|\nabla f(\mathbf{x}_t)\| \geq \frac{\epsilon}{2}$, according to Lemma 1). We have the following average decrease lemma:

**Lemma 5 (Large gradient).** *If $\|\hat{\nabla} f(\mathbf{x}_\tau)\| \geq \frac{3\epsilon}{4}$ with $\mu \leq \mathcal{O}((\frac{3\epsilon}{2\rho\sqrt{d}})^{1/2})$ in Line 3 of Algorithm 1 for all $\tau \in [0, \mathscr{T}]$, then by running Algorithm 1 with $\mu \leq \tilde{\mathcal{O}}(\frac{\epsilon^{5/8}}{d^{1/4}})$ in Line 6 and $\mu \leq \tilde{\mathcal{O}}(\frac{\epsilon^{1/2}}{d^{1/4}})$ in Line 8, we have $E_{\mathscr{T}} - E_0 \leq -\mathscr{E}$.*

On the other hand, when the current approximate gradient is small and no perturbation is added over the past $\mathscr{T}$ iterations, then we add a uniform random perturbation in $\mathbb{B}_0(r)$. If there exist a large negative curvature direction of the current point, we have

**Lemma 6** (**Negative curvature**). *Suppose* $\|\hat{\nabla}f(\mathbf{x}_t)\| \leq \frac{3\epsilon}{4}$ *( thus* $\|\nabla f(\mathbf{x}_t)\| \leq \epsilon$*),* $\lambda_{\min}(\nabla^2 f(\mathbf{x}_t)) \leq -\sqrt{\rho\epsilon}$ *and no perturbation is added in iterations* $[t - \mathscr{T}, t)$. *Then by running Algorithm 1, we have* $E_{\mathscr{T}} - E_0 \leq -\mathscr{E}$ *with probability at least* $1 - \frac{\delta\mathscr{E}}{2\Delta_f}$.

Utilizing the above lemmas, we finally get the following main result.

**Theorem 1.** *Assume that* $f(\cdot)$ *is* $\ell$-*smooth and* $\rho$-*Hessian Lipschitz. For any* $\delta > 0, \epsilon \leq \frac{\ell^2}{\rho}, f(\mathbf{x}_0) - f^* \leq \Delta_f$, *if we set the hyperparameters as in Eq. (5) and choose* $\mu = \tilde{\mathcal{O}}(\frac{\epsilon^{1/2}}{d^{1/4}})$ *in Line 3 and 8,* $\mu = \tilde{\mathcal{O}}(\frac{\epsilon^{13/8}}{d^{1/2}})$ *in Line 6 of Algorithm 1, respectively, then with probability at least* $1 - \delta$, *one of the iterates of* $\mathbf{x}_t$ *will be an* $\epsilon$-*approximate SOSP. The total number of iterations is no more than* $\mathcal{O}\left(\frac{\Delta_f \ell^{1/2}\rho^{1/4}}{\epsilon^{7/4}} \log^6(\frac{d\ell\Delta_f}{\rho\epsilon\delta})\right)$ *and the total number of function queries (oracle complexity) is no more than*

$$\mathcal{O}\left(\frac{d\Delta_f \ell^{1/2}\rho^{1/4}}{\epsilon^{7/4}} \log^6(\frac{d\ell\Delta_f}{\rho\epsilon\delta})\right).$$

**Proof outline.** We first prove two monotonical descent lemmas (Lemma 3 and Lemma 4) of the Hamiltonian in each iteration and an improve or localize property in Appendix B. Next, in Appendix C, we prove that Hamiltonian will decrease by $\mathcal{E}$ in $\mathscr{T}$ iterations in both large gradient and negative curvature scenarios.

**Remark 2.** *Note that, Theorem 1 only ensures that with high probability, one of the iterates will be an* $\epsilon$-*approximate SOSP. It is then natural to add a termination condition to make the algorithm more practical: Once the pre-condition of random perturbation step is reached, record the current iterate point* $\mathbf{x}_{t_0}$ *and the current function value of the Hamiltonian* $E_{t_0}$ *before adding the random perturbation. If the decrease of the Hamiltonian is less than* $\mathscr{E}$ *after* $\mathscr{T}$ *iterations, then, with high probability* $\mathbf{x}_{t_0}$ *is an* $\epsilon$-*approximate SOSP according to Lemma 6.*

## 3.2 ZEROTH-ORDER PERTURBED ACCELERATED GRADIENT DESCENT WITH ACCELERATED NEGATIVE CURVATURE FINDING

In this subsection, we introduce how to utilize the negative curvature finding to accelerate escaping saddle points. The main task of the negative curvature finding is to find the approximate most negative eigenvector direction near a saddle point. Then adding a perturbation in this direction will obtain a more efficient decrease of the function value.

Classical methods for computing the most negative eigenvector direction like the power method and Lanczos method require the computations of the Hessian-vector products. Since we have only access to the zeroth-order information, an efficient way to approximate the Hessian-vector product is to utilize the zeroth-order Hessian-vector product estimator in Eq. (3). The accelerated negative curvature finding subroutine is self-contained in Line 11-13 of Algorithm 3 when $\zeta \neq \mathbf{0}$. The following lemma states that the accelerated negative curvature finding using zeroth-order Hessian-vector product estimator can find a negative curvature direction in almost the same iteration complexity as the Lanczos method.

**Lemma 7.** *Suppose* $\|\hat{\nabla}f(\mathbf{x}_t)\| \leq \frac{3\epsilon}{4}$, $\lambda_{\min}(\nabla^2 f(\mathbf{x}_t)) \leq -\sqrt{\rho\epsilon}$ *and no perturbation is added in iterations* $[t - \mathscr{T}', t]$. *For any* $0 < \delta_0 < 1$, *let* $\kappa = \frac{\ell}{\sqrt{\rho\epsilon}}$, *and set the parameters as follows,*

$$\eta = \frac{1}{4\ell}, \qquad \theta = \frac{1}{4\sqrt{\kappa}}, \qquad \gamma = \frac{\theta^2}{\eta}, \qquad s = \frac{\gamma}{4\rho},$$

$$\mathscr{T}' = 32\sqrt{\kappa}\log(\frac{\ell\sqrt{d}}{\delta_0\sqrt{\rho\epsilon}}), \qquad r' = \frac{\delta_0\epsilon}{32}\sqrt{\frac{\pi}{\rho d}}. \qquad (7)$$

*Then by running Algorithm 3 for* $\mathscr{T}'$ *iterations after adding the random perturbation in Line 5, with probability at least* $1 - \delta_0$, *we have* $\hat{\mathbf{e}}^{\mathsf{T}}\nabla^2 f(\mathbf{x}_t)\hat{\mathbf{e}} \leq -\frac{\sqrt{\rho\epsilon}}{4}$.

---

**Algorithm 3** Zeroth-Order Perturbed Accelerated Gradient Descent with Accelerated Negative Curvature Finding

---

1: $t_{\text{perturb}} \leftarrow -\mathcal{T}' - 1, \mathbf{y}_0 \leftarrow \mathbf{x}_0, \tilde{\mathbf{x}} \leftarrow \mathbf{x}_0, \zeta \leftarrow \mathbf{0}$
2: **for** $t = 0, 1, \ldots,$ **do**
3:     **if** $\|\hat{\nabla} f(\mathbf{x}_t)\| \leq \frac{3\epsilon}{4}$ and $t - t_{\text{perturb}} > \mathcal{T}'$ **then**
4:         $\tilde{\mathbf{x}} = \mathbf{x}_t$
5:         $\mathbf{x}_t = \tilde{\mathbf{x}} + \xi_t, \xi_t \sim \text{Unif}(\mathbb{B}_0(r'))$
6:         $\mathbf{y}_t = \mathbf{x}_t, \zeta = \hat{\nabla} f(\tilde{\mathbf{x}}), t_{\text{perturb}} \leftarrow t$
7:     **if** $t_{\text{perturb}} \neq -\mathcal{T}' - 1$ and $t - t_{\text{perturb}} = \mathcal{T}'$ **then**
8:         $\hat{\mathbf{e}} \leftarrow \frac{\mathbf{x}_t - \tilde{\mathbf{x}}}{\|\mathbf{x}_t - \tilde{\mathbf{x}}\|}$
9:         $\mathbf{x}_t \leftarrow \arg\min_{\mathbf{x} \in \{\tilde{\mathbf{x}} - \frac{1}{4}\sqrt{\frac{\epsilon}{\rho}}\hat{\mathbf{e}}, \tilde{\mathbf{x}} + \frac{1}{4}\sqrt{\frac{\epsilon}{\rho}}\hat{\mathbf{e}}\}} f(\mathbf{x})$
10:         $\mathbf{y}_t = \mathbf{x}_t, \zeta = \mathbf{0}$
11:     $\mathbf{x}_{t+1} = \mathbf{y}_t - \eta(\hat{\nabla} f(\mathbf{y}_t) - \zeta)$
12:     $\mathbf{v}_{t+1} = \mathbf{x}_{t+1} - \mathbf{x}_t$
13:     $\mathbf{y}_{t+1} = \mathbf{x}_{t+1} + (1 - \theta)\mathbf{v}_{t+1}$
14:     **if** $t_{\text{perturb}} \neq -\mathcal{T}' - 1$ and $t - t_{\text{perturb}} < \mathcal{T}'$ **then**
15:         $(\mathbf{y}_{t+1}, \mathbf{x}_{t+1}) = (\tilde{\mathbf{x}}, \tilde{\mathbf{x}}) + r' \cdot (\frac{\mathbf{y}_{t+1} - \tilde{\mathbf{x}}}{\|\mathbf{y}_{t+1} - \tilde{\mathbf{x}}\|}, \frac{\mathbf{x}_{t+1} - \tilde{\mathbf{x}}}{\|\mathbf{x}_{t+1} - \tilde{\mathbf{x}}\|})$
16:     **else if** $f(\mathbf{x}_{t+1}) \leq f(\mathbf{y}_{t+1}) + \left\langle \hat{\nabla} f(\mathbf{y}_{t+1}), \mathbf{x}_{t+1} - \mathbf{y}_{t+1} \right\rangle - \frac{\gamma}{2}\|\mathbf{y}_{t+1} - \mathbf{x}_{t+1}\|^2$ **then**
17:         $(\mathbf{x}_{t+1}, \mathbf{v}_{t+1}) \leftarrow \text{NCE}(\mathbf{x}_{t+1}, \mathbf{v}_{t+1}, s)$
18:         $\mathbf{y}_{t+1} \leftarrow \mathbf{x}_{t+1} + (1 - \theta)\mathbf{v}_{t+1}$

---

Then moving along the direction of $\hat{\mathbf{e}}$, the function value of $f$ will make further decrease according to the following lemma:

**Lemma 8** (Zhang & Li (2021), Lemma 6). *Suppose the function $f : \mathbb{R}^d \rightarrow \mathbb{R}$ is $\ell$-smooth and $\rho$-Hessian Lipschitz. Then for any point $\mathbf{x}_0 \in \mathbb{R}^d$, if there exist a unit vector $\hat{\mathbf{e}}$ satisfying $\hat{\mathbf{e}}\nabla^2 f(\mathbf{x}_0)\hat{\mathbf{e}} \leq -\frac{\sqrt{\rho\epsilon}}{4}$, then we have $f\left(\mathbf{x}_0 - \frac{f'_{\hat{\mathbf{e}}}(\mathbf{x}_0)}{4|f'_{\hat{\mathbf{e}}}(\mathbf{x}_0)|}\sqrt{\frac{\epsilon}{\rho}}\hat{\mathbf{e}}\right) \leq f(\mathbf{x}_0) - \frac{1}{384}\sqrt{\frac{\epsilon^3}{\rho}}$, where $f'_{\hat{\mathbf{e}}}(\mathbf{x}_0)$ is the directional derivative along the direction $\hat{\mathbf{e}}$.*

**Remark 3.** *In the first-order setting, $f'_{\hat{\mathbf{e}}}(\mathbf{x}_0) = \langle \nabla f(\mathbf{x}_0), \hat{e} \rangle$. However, in the zeroth-order setting, the directional derivative cannot be computed directly. To tackle this problem, one can simply compare the function value of two opposite directions, i.e., Line 9 of Algorithm 3.*

**Theorem 2.** *Assume that $f(\cdot)$ is $\ell$-smooth and $\rho$-Hessian Lipschitz. For any $\delta > 0, \epsilon \leq \frac{\ell^2}{\rho}, f(\mathbf{x}_0) - f^* \leq \Delta_f$, if we set the hyperparameters as in Eq. (7) with $\delta_0 = \frac{\delta}{384\Delta_f}\sqrt{\frac{\epsilon^3}{\rho}}$ and choose $\mu = \tilde{\mathcal{O}}(\frac{\epsilon^{1/2}}{d^{1/4}})$ in Line 3 and 16, $\mu = \tilde{\mathcal{O}}(\frac{\epsilon^{13/8}}{d^{1/2}})$ in Line 11 of Algorithm 3. Then with probability at least $1 - \delta$, one of the iterates of $\mathbf{x}_t$ in Algorithm 3 will be an $\epsilon$-approximate SOSP. The total number of iterations is no more than $\mathcal{O}\left(\frac{\Delta_f \ell^{1/2}\rho^{1/4}}{\epsilon^{7/4}}\log(\frac{\ell\sqrt{d}\Delta_f}{\delta\epsilon^2})\right)$ and the total number of function queries (oracle complexity) is no more than*

$$\mathcal{O}\left(\frac{d\Delta_f \ell^{1/2}\rho^{1/4}}{\epsilon^{7/4}}\log(\frac{\ell\sqrt{d}\Delta_f}{\delta\epsilon^2})\right).$$

**Remark 4.** *Similar to Algorithm 1, we can also add an termination condition for Algorithm 3: Once the pre-condition of random perturbation step is reached, record the current iterate point $\mathbf{x}_{t_0}$ and the current function value $f(\mathbf{x}_{t_0})$ before adding the random perturbation. If the decrease of the function is less than $\frac{1}{384}\sqrt{\frac{\epsilon^3}{\rho}}$ after $\mathcal{T}'$ iterations, then, with high probability $\mathbf{x}_{t_0}$ is an $\epsilon$-approximate SOSP according to Lemma 8.*

**Remark 5** (Proof outline). *The main difference between Algorithm 1 and Algorithm 3 is the way in which random perturbations are added. Specifically, in Algorithm 1, we add a uniform random perturbation nearby a first-order stationary point. If it is a saddle point, then by running the zeroth-order accelerated gradient descent for $\mathcal{T} = \sqrt{\kappa}\chi c = \tilde{\Theta}(\sqrt{\kappa})$ steps, the value of Hamiltonian function will decrease by $\mathscr{E} := \sqrt{\frac{\epsilon^3}{\rho}}\chi^{-5}c^{-7} = \tilde{\Theta}(\sqrt{\frac{\epsilon^3}{\rho}})$. In Algorithm 3, the perturbation is*

*added along an approximate negative curvature direction, which is obtained by running $\mathscr{T}' = 32\sqrt{\kappa}\log(\frac{\ell\sqrt{d}}{\delta_0\sqrt{\rho\epsilon}})$ steps of zeroth-order accelerated negative curvature finding (Line 11-13). Then moving along the negative curvature direction, the value of Hamiltonian function will decrease by $\frac{1}{384}\sqrt{\frac{\epsilon^3}{\rho}}$ (no more log term as in $\mathscr{E}$). Thus, the total function query complexity induced by Algorithm 3 is $\mathcal{O}(d \cdot \sqrt{\kappa}\log(\frac{\ell\sqrt{d}}{\delta_0\sqrt{\rho\epsilon}}) \cdot \sqrt{\frac{\rho}{\epsilon^3}}) = \mathcal{O}\left(\frac{d\Delta_f\ell^{1/2}\rho^{1/4}}{\epsilon^{7/4}}\log(\frac{\ell\sqrt{d}\Delta_f}{\delta\epsilon^2})\right).$*

## 4 NUMERICAL EXPERIMENTS

In this section, we conduct several numerical experiments to verify the effectiveness of the proposed methods for escaping saddle points and the efficiency compared with the existing methods. Specially, we run zeroth-order perturbed accelerated gradient descent (Algorithm 1) and zeroth-order perturbed accelerated gradient descent with accelerated negative curvature finding (Algorithm 3) against the perturbed approximate gradient descent (PAGD) and the random search power iteration (RSPI) method. All experiments are performed on a computer with a six-core Intel Core i5-10500 CPU.

### 4.1 CUBIC REGULARIZATION PROBLEM

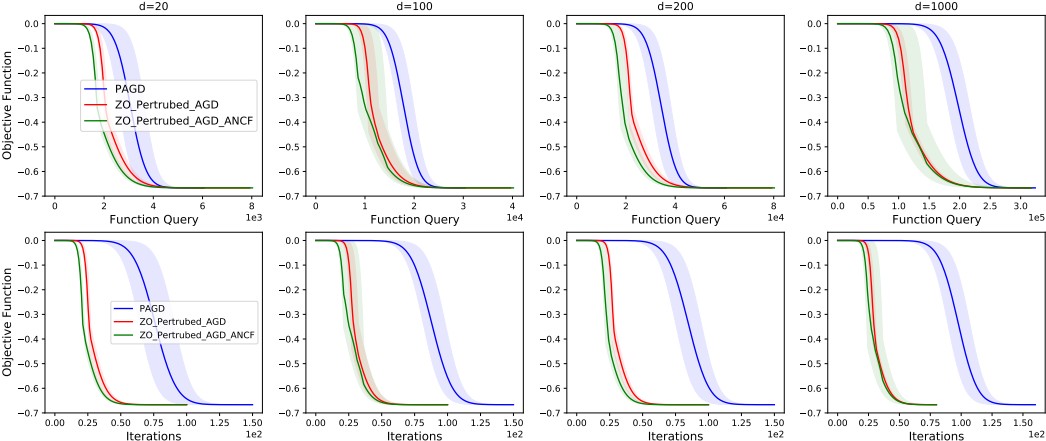

Figure 1: Performance of different algorithms to minimize the cubic regularization problem with growing dimensions. Confidence intervals show mini-max intervals over ten runs

We first consider the cubic regularization problem (Liu et al., 2018), which is defined as:

$$\min_{\mathbf{x}\in\mathbb{R}^d} f(\mathbf{x}) := \frac{1}{2}\mathbf{x}^\mathsf{T}\mathbf{A}\mathbf{x} + \frac{1}{6}\|\mathbf{x}\|^3. \tag{8}$$

Above, $\mathbf{A}$ is a randomly generated diagonal matrix with only one diagonal entry is -1 and the rest diagonal entries are uniformly distributed between $[1, 2]$. So that with increase of the dimension, the negative curvature directions that can escape from the saddle point will be more difficult to explore. In this experiment, we set $\epsilon = 10^{-2}$. To test the ability of different algorithms to escape from saddle points, we initialize all algorithms at a strict saddle point $\mathbf{x}_0 = (0, \ldots, 0)^\mathsf{T}$.

In this experiment, we run Algorithm 1, 3, PAGD on the above cubic regularization problem from a strict saddle point. For Algorithm 1 and 3, the parameter settings basically follow Eq. (5) and Eq. (7). Specifically, we choose $\epsilon = 0.001$ and the perturbation radius $r$ and $r'$ are set to 0.001. The Lipschitz constants $\ell$ and $\rho$ are selected based on a coarse grid search of the region $\{0.1, 1, 10, 100\} \times \{0.1, 1, 10, 100\}$. Since all algorithms have certain randomness, we repeatedly run each algorithm multiple times and report the averaged function value versus the averaged number of function queries and the number of iterations in Figure 2.

The results in Fig. 1 illustrate that Algorithm 1, 3 can escape saddle points using less iterations than PAGD and converge faster than PAGD. On the other hand, in all dimensions, the number of

iterations for escaping saddle points are almost the same. This verifies the result in Lemma 6 and 7 that the number of iterations of Algorithm 1, 3 are only log dependent on the dimension $d$.

## 4.2 QUARTIC FUNCTION

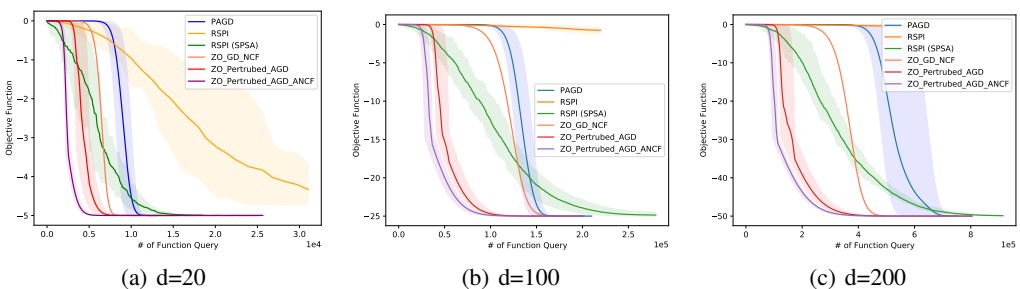

|  (a) d=20 | (b) d=100 | (c) d=200 |

Figure 2: Performance of different algorithms to minimize the quartic function with growing dimensions. Confidence intervals show mini-max intervals over ten runs.

Then we consider the following quartic function (Lucchi et al., 2021),

$$f(x_1, x_2, \ldots, x_d, y) = \frac{1}{4} \sum_{i=1}^{d} x_i^4 - y \sum_{i=1}^{d} x_i + \frac{d}{2} y^2 \tag{9}$$

which has a strict saddle point at $\mathbf{x}_0 = (0, \ldots, 0)^\mathsf{T}$ and two global minima at $(1, \ldots, 1)^\mathsf{T}$ and $(-1, \ldots, -1)^\mathsf{T}$.

In this experiment, we run Algorithm 1, 3, perturbed approximate gradient descent (PAGD), Random Search Power Iteration (RSPI) and ZO-GD-NCF on the above quartic function staring from its saddle point. Especially, we also run an acceleration version of RSPI, which replaces the finite difference gradient estimator in RSPI by the SPSA estimator (Spall et al., 1992). The parameter settings of PAGD are taken from Vlatakis-Gkaragkounis et al. (2019) and the parameters of RSPI are taken from the appendix of Lucchi et al. (2021). For Algorithm 1 and 3, the parameter settings basically follow Eq. (5) and Eq. (7). Specifically, we choose $\epsilon = 10^{-4}$ and the perturbation radius $r$ and $r'$ are set to 0.01. The Lipschitz constants $\ell$ and $\rho$ are selected based on a coarse grid search of the region $\{10, 20, 100, 150, 200\} \times \{0.1, 1, 10\}$. Since all algorithms have certain randomness, we repeatedly run each algorithm multiple times and report the averaged function value versus the averaged number of function queries in Figure 2.

The results in Fig.2 illustrate that both Algorithms 1 and 3 can efficiently escape saddle points and converge quickly to the global minimum. Note that, for all dimensions, Algorithms 1 and 3 escape saddle points with fewer function queries than PAGD. This verifies the theoretical result that algorithms 1 and 3 take $\tilde{\Theta}(\sqrt{\kappa})$ iterations for escaping saddle points when the initial point is a saddle point, while PAGD takes $\tilde{\Theta}(\kappa)$ iterations. For high dimensional problems, the computational cost of RSPI for escaping saddle points is expensive. In contrast, RSPI with SPSA estimator is much more efficient.

## 5 CONCLUSION

In this paper, we study the complexity of two zeroth-order AGD based algorithms for escaping saddle points and converging to SOSPs. The first method is a zeroth-order version of the perturbed AGD which uses the central finite difference version of the coordinate-wise gradient estimator. The second method extracts accelerated negative curvature findings by using the finite difference of two coordinate-wise gradient estimators. Both methods improve the function query complexity of prior zeroth-order methods for converging to SOSPs.

ACKNOWLEDGMENT

The authors thank four anonymous reviewers for their helpful comments and suggestions. Bin Gu was partially supported by the National Natural Science Foundation of China under Grant 62076138.

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

## APPENDIX

## A    AUXILIARY LEMMAS

**Lemma 9** (Nesterov et al. (2018), Lemma 1.2.3 & 1.2.4). *If $f$ is $\ell$-Lipschitz smooth, then for all* $x, y \in \mathbb{R}^d$,

$$|f(\mathbf{y}) - f(\mathbf{x}) - \nabla f(\mathbf{x})^T(\mathbf{y} - \mathbf{x})| \le \frac{\ell}{2}\|\mathbf{y} - \mathbf{x}\|^2.$$

*If $f$ is $\rho$-Hessian Lipschitz, then for all* $x, y \in \mathbb{R}^d$,

$$\left\|\nabla f(\mathbf{y}) - \nabla f(\mathbf{x}) - \nabla^2 f(\mathbf{x})(\mathbf{y} - \mathbf{x})\right\| \le \frac{\rho}{2}\|\mathbf{y} - \mathbf{x}\|^2,$$

*and*

$$\left| f(\mathbf{y}) - f(\mathbf{x}) - \nabla f(\mathbf{x})^T (\mathbf{y} - \mathbf{x}) - \frac{1}{2}(\mathbf{y} - \mathbf{x})^T \nabla^2 f(\mathbf{x})(\mathbf{y} - \mathbf{x}) \right| \le \frac{\rho}{6} \|\mathbf{y} - \mathbf{x}\|^3.$$

**Lemma 1.** *If $f$ is $\rho$-Hessian Lipschitz, then for any given smoothing parameter $\mu$ and any $x \in \mathbb{R}^d$, if $f$ is $\ell$-Lipschitz smooth, we have*

$$\|\hat{\nabla} f(\mathbf{x}) - \nabla f(\mathbf{x})\|^2 \le \frac{1}{36} \rho^2 d \mu^4 \tag{10}$$

*Proof.*

$$\left\| \nabla f(x) - \hat{\nabla}_{coord} f(x) \right\| = \left\| \sum_{i=1}^d \frac{f(x + \mu e_i) - f(x - \mu e_i)}{2\mu} e_i - \nabla f(x) \right\|$$

$$= \frac{1}{2\mu} \left\| \sum_{i=1}^d (f(x + \mu e_i) - f(x - \mu e_i) - 2\mu \nabla_i f(x)) e_i \right\|$$

Since $f$ is $\rho$-Hessian Lipschitz, for all $i \in [d]$, we have

$$f(x + \mu e_i) - f(x - \mu e_i) - 2\mu \nabla_i f(x)$$

$$= \left[ f(x + \mu e_i) - f(x) - \mu \nabla_i f(x) - \frac{\mu^2}{2} \nabla_{ii}^2 f(x) \right] - \left[ f(x - \mu e_i) - f(x) + \mu \nabla_i f(x) - \frac{\mu^2}{2} \nabla_{ii}^2 f(x) \right]$$

$$\le \left| f(x + \mu e_i) - f(x) - \mu \nabla_i f(x) - \frac{\mu^2}{2} \nabla_{ii}^2 f(x) \right| + \left| f(x - \mu e_i) - f(x) + \mu \nabla_i f(x) - \frac{\mu^2}{2} \nabla_{ii}^2 f(x) \right|$$

$$\overset{\text{①}}{\le} 2 \cdot \frac{\rho}{6} \mu^3 = \frac{\rho}{3} \mu^3$$

where ① is due to Lemma 9.

$$\left\| \nabla f(x) - \hat{\nabla}_{coord} f(x) \right\|$$

$$= \frac{1}{2\mu} \left\| \sum_{i=1}^d (f(x + \mu e_i) - f(x - \mu e_i) - 2\mu \nabla_i f(x)) e_i \right\|$$

$$= \frac{1}{2\mu} \sqrt{\sum_{i=1}^d (f(x + \mu e_i) - f(x - \mu e_i) - 2\mu \nabla_i f(x))^2}$$

$$\le \frac{1}{2\mu} \sqrt{d \left( \frac{\rho \mu^3}{3} \right)^2} = \frac{\sqrt{d} \rho \mu^2}{6}$$

$\square$

**Lemma 10** (Jin et al. (2018b), Lemma 24 & 25). *Define*

$$\mathbf{A} = \begin{pmatrix} a & b \\ 1 & 0 \end{pmatrix}$$

*Let $\mu_1, \mu_2$ denote the two eigenvalues of $\mathbf{A}$, then $\mathbf{A}$ can be rewritten as $\mathbf{A} = \begin{pmatrix} \mu_1 + \mu_2 & -\mu_1 \mu_2 \\ 1 & 0 \end{pmatrix}$ and for any $t \in \mathbb{N}$:*

$$(0 \quad 1) \mathbf{A}^t = (1 \quad 0) \mathbf{A}^{t-1}$$

$$(\mu_1 - 1)(\mu_2 - 1)(1 \quad 0) \sum_{\tau=0}^{t-1} \mathbf{A}^\tau \begin{pmatrix} 1 \\ 0 \end{pmatrix} = 1 - (1 \quad 0) \mathbf{A}^t \begin{pmatrix} 1 \\ 1 \end{pmatrix}.$$

**Lemma 11** (Jin et al. (2018b), Lemma 30). *Let $\theta \in (0, 1/4]$, define*

$$\mathbf{A} = \begin{pmatrix} (2-\theta)(1-x) & -(1-\theta)(1-x) \\ 1 & 0 \end{pmatrix},$$

*and let $x \in [-\frac{1}{4}, \frac{\theta^2}{(2-\theta)^2}]$. Denote $(a_t \quad -b_t) = (1 \quad 0)\mathbf{A}^t$, then for any $t \geq \frac{2}{\theta} + 1$, we have*

$$\sum_{\tau=0}^{t-1} a_\tau \geq \Omega(\frac{1}{\theta^2}), \qquad \frac{1}{b_t}(\sum_{\tau=0}^{t-1} a_\tau) \geq \Omega(1)\min\{\frac{1}{\theta}, \frac{1}{\sqrt{|x|}}\}.$$

**Lemma 12** (Jin et al. (2018b), Lemma 32). *Let $\theta \in (0, 1/4]$, define*

$$\mathbf{A} = \begin{pmatrix} (2-\theta)(1-x) & -(1-\theta)(1-x) \\ 1 & 0 \end{pmatrix},$$

*and let $x \in [\frac{\theta^2}{(2-\theta)^2}, \frac{1}{4}]$. Denote $(a_t \quad -b_t) = (1 \quad 0)\mathbf{A}^t$, then for any $t \geq 0$, we have*

$$\max\{|a_t|, |b_t|\} \leq (t+1)(1-\theta)^{1/2}.$$

**Lemma 13** (Jin et al. (2018b), Lemma 34). *Under the same setting as in Lemma 12, for any sequence $\epsilon_\tau$, any $t \geq \Omega(1/\theta)$, we have:*

$$\sum_{\tau=0}^{t-1} a_\tau \epsilon_\tau \leq \mathcal{O}(1/x)\left(|\epsilon_0| + \sum_{\tau=1}^{t-1} |\epsilon_\tau - \epsilon_{\tau-1}|\right)$$

$$\sum_{\tau=0}^{t-1} (a_\tau - a_{\tau-1})\epsilon_\tau \leq \mathcal{O}(1/\sqrt{x})\left(|\epsilon_0| + \sum_{\tau=1}^{t-1} |\epsilon_\tau - \epsilon_{\tau-1}|\right)$$

**Lemma 14** (Jin et al. (2018b), Lemma 36). *Let $\theta \in (0, 1/4]$, define*

$$\mathbf{A} = \begin{pmatrix} (2-\theta)(1-x) & -(1-\theta)(1-x) \\ 1 & 0 \end{pmatrix},$$

*and let $x \in [-1/4, 0]$, denote $(a_t \quad -b_t) = (1 \quad 0)\mathbf{A}^t$. Then for any $0 \leq \tau \leq t$, we have*

$$|a_{t-\tau}||a_\tau - b_\tau| \leq [\frac{2}{\theta} + t + 1]|a_{t+1} - b_{t+1}|.$$

**Lemma 15** (Jin et al. (2018b), Lemma 37). *Under the same setting as in Lemma 14, let $\mathbf{A}(x) = \mathbf{A}$ and $g(x) = |(1 \quad 0)[\mathbf{A}(x)]^t \begin{pmatrix} 1 \\ 0 \end{pmatrix}|$, then we have*

1. *$g(x)$ is a monotonically decreasing function for $x \in [-1, \theta^2/(2-\theta)^2]$.*

2. *For any $x \in [\theta^2/(2-\theta^2), 1]$, we have $g(x) \leq g(\theta^2/(2-\theta^2))$.*

**Lemma 16** (Jin et al. (2018b), Lemma 38). *Under the same setting as in Lemma 14, we have*

$$|a_{t+1} - b_{t+1}| \geq |a_t - b_t| = (a_t, \quad -b_t) = (1 \quad 0)\mathbf{A}^t \begin{pmatrix} 1 \\ 1 \end{pmatrix} \qquad and \qquad |a_t - b_t| \geq \frac{\theta}{2}(1 + \frac{1}{2}\min\{\frac{|x|}{\theta}, \sqrt{|x|}\})^t.$$

**Lemma 17** (Zhang & Li (2021), Lemma 21). *Consider the sequence with recurrence:*

$$\xi_{t+2} = (1+\kappa)((2-\theta)\xi_{t+1} - (1-\theta)\xi_t),$$

*for some $\kappa > 0$. Then we have*

$$\xi_t = (\frac{1+\kappa}{2})^t(C_1(2-\theta-\mu)^t + C_2(2-\theta-\mu)^t),$$

*where $\mu = \sqrt{(2-\theta)^2 - \frac{4(1-\theta)}{1+\kappa}}$, $C_1 = -\frac{2-\theta-\mu}{2\mu}\xi_0 + \frac{1}{(1+\kappa)\mu}\xi_1$, $C_2 = \frac{1-\theta+\mu}{2\mu}\xi_0 - \frac{1}{(1+\kappa)\mu}\xi_1$.*

## B    PROOF OF HAMILTONIAN LEMMAS IN THE ZEROTH-ORDER SETTING

**Lemma 3.** *Assume that $f(\cdot)$ is $\ell$-smooth and set the learning rate $\eta \leq \frac{1}{4\ell}, \theta \in [2\eta\gamma, \frac{1}{2}]$. Then, for each iteration $t$ where Eq. (6) does not hold, we have:*

$$E_{t+1} \leq E_t - \frac{\theta}{2\eta}\|\mathbf{v}_t\|^2 - \frac{\eta}{4}\|\nabla f(\mathbf{y}_t)\|^2 + \eta \cdot \frac{\rho^2 d\mu^4}{48}$$

*Proof.*

$$\mathbf{x}_{t+1} \leftarrow \mathbf{y}_t - \eta\hat{\nabla}f(\mathbf{y}_t)$$
$$\mathbf{y}_{t+1} \leftarrow \mathbf{x}_{t+1} + (1-\theta)(\mathbf{x}_{t+1} - \mathbf{x}_t)$$

By smoothness, with $\eta \leq \frac{1}{4\ell}$, we have

$$f(\mathbf{x}_{t+1}) \leq f(\mathbf{y}_t) + \langle \nabla f(\mathbf{y}_t), \mathbf{x}_{t+1} - \mathbf{y}_t \rangle + \frac{\ell}{2}\|\mathbf{x}_{t+1} - \mathbf{y}_t\|^2$$

$$= f(\mathbf{y}_t) - \eta\left\langle \nabla f(\mathbf{y}_t), \hat{\nabla}f(\mathbf{y}_t) \right\rangle + \frac{\ell\eta^2}{2}\|\hat{\nabla}f(\mathbf{y}_t)\|^2$$

According to the update rule of the accelerated gradient descent, we have

$$\|\mathbf{x}_{t+1} - \mathbf{x}_t\|^2 = \|\mathbf{y}_t - \eta\hat{\nabla}f(\mathbf{y}_t) - \mathbf{x}_t\|^2$$

$$= \|\mathbf{y}_t - \mathbf{x}_t\|^2 - 2\eta\left\langle \hat{\nabla}f(\mathbf{y}_t), \mathbf{y}_t - \mathbf{x}_t \right\rangle + \eta^2\|\hat{\nabla}f(\mathbf{y}_t)\|^2.$$

Dividing both sides by $2\eta$, we have

$$\frac{1}{2\eta}\|\mathbf{x}_{t+1} - \mathbf{x}_t\|^2 = \frac{1}{2\eta}\|\mathbf{y}_t - \mathbf{x}_t\|^2 + \left\langle \hat{\nabla}f(\mathbf{y}_t), \mathbf{x}_t - \mathbf{y}_t \right\rangle + \frac{\eta}{2}\|\hat{\nabla}f(\mathbf{y}_t)\|^2$$

Then we have

$$f(\mathbf{x}_{t+1}) + \frac{1}{2\eta}\|\mathbf{x}_{t+1} - \mathbf{x}_t\|^2$$

$$\leq f(\mathbf{y}_t) + \frac{1}{2\eta}\|\mathbf{y}_t - \mathbf{x}_t\|^2 + \left\langle \hat{\nabla}f(\mathbf{y}_t), \mathbf{x}_t - \mathbf{y}_t \right\rangle + \frac{\eta}{2}\|\hat{\nabla}f(\mathbf{y}_t)\|^2 - \eta\left\langle \nabla f(\mathbf{y}_t), \hat{\nabla}f(\mathbf{y}_t) \right\rangle + \frac{\ell\eta^2}{2}\|\hat{\nabla}f(\mathbf{y}_t)\|^2$$

$$= f(\mathbf{y}_t) + \frac{1}{2\eta}\|\mathbf{y}_t - \mathbf{x}_t\|^2 + \left\langle \hat{\nabla}f(\mathbf{y}_t), \mathbf{x}_t - \mathbf{y}_t \right\rangle - \eta\left\langle \nabla f(\mathbf{y}_t), \hat{\nabla}f(\mathbf{y}_t) \right\rangle + \frac{\eta}{2}(1 + \ell\eta)\|\hat{\nabla}f(\mathbf{y}_t)\|^2$$

As long as the following condition holds:

$$f(\mathbf{x}_t) \geq f(\mathbf{y}_t) + \left\langle \hat{\nabla}f(\mathbf{y}_t), \mathbf{x}_t - \mathbf{y}_t \right\rangle - \frac{\gamma}{2}\|\mathbf{x}_t - \mathbf{y}_t\|^2,$$

we have

$$f(\mathbf{x}_{t+1}) + \frac{1}{2\eta}\|\mathbf{x}_{t+1} - \mathbf{x}_t\|^2$$

$$\leq f(\mathbf{x}_t) + \frac{1 + \eta\gamma}{2\eta}\|\mathbf{y}_t - \mathbf{x}_t\|^2 - \eta\left\langle \nabla f(\mathbf{y}_t), \hat{\nabla}f(\mathbf{y}_t) \right\rangle + \frac{\eta}{2}(1 + \ell\eta)\|\hat{\nabla}f(\mathbf{y}_t)\|^2$$

Note that

$$-\left\langle \nabla f(\mathbf{y}_t), \hat{\nabla}f(\mathbf{y}_t) \right\rangle = -\|\nabla f(\mathbf{y}_t)\|^2 - \left\langle \nabla f(\mathbf{y}_t), \hat{\nabla}f(\mathbf{y}_t) - \nabla f(\mathbf{y}_t) \right\rangle,$$

and

$$\|\hat{\nabla}f(\mathbf{y}_t)\|^2 = \|\nabla f(\mathbf{y}_t) + \hat{\nabla}f(\mathbf{y}_t) - \nabla f(\mathbf{y}_t)\|^2$$

$$= \|\nabla f(\mathbf{y}_t)\|^2 + 2\left\langle \nabla f(\mathbf{y}_t), \hat{\nabla}f(\mathbf{y}_t) - \nabla f(\mathbf{y}_t) \right\rangle + \|\hat{\nabla}f(\mathbf{y}_t) - \nabla f(\mathbf{y}_t)\|^2$$

Combine the two equations with the above inequality, we have

$$f(\mathbf{x}_{t+1}) + \frac{1}{2\eta}\|\mathbf{x}_{t+1} - \mathbf{x}_t\|^2$$

$$\leq f(\mathbf{x}_t) + \frac{1+\eta\gamma}{2\eta}\|\mathbf{y}_t - \mathbf{x}_t\|^2 - \frac{\eta(1-\ell\eta)}{2}\|\nabla f(\mathbf{y}_t)\|^2 + \ell\eta^2 \left\langle \nabla f(\mathbf{y}_t), \hat{\nabla} f(\mathbf{y}_t) - \nabla f(\mathbf{y}_t)\right\rangle$$

$$+ \frac{\eta(1+\ell\eta)}{2}\|\hat{\nabla} f(\mathbf{y}_t) - \nabla f(\mathbf{y}_t)\|^2$$

$$\leq f(\mathbf{x}_t) + \frac{1+\eta\gamma}{2\eta}\|\mathbf{y}_t - \mathbf{x}_t\|^2 - \frac{\eta(1-\ell\eta)}{2}\|\nabla f(\mathbf{y}_t)\|^2 + \ell\eta^2 \left( \frac{\beta}{2}\|\nabla f(\mathbf{y}_t)\|^2 + \frac{1}{2\beta}\|\hat{\nabla} f(\mathbf{y}_t) - \nabla f(\mathbf{y}_t)\|^2 \right)$$

$$+ \frac{\eta(1+\ell\eta)}{2}\|\hat{\nabla} f(\mathbf{y}_t) - \nabla f(\mathbf{y}_t)\|^2$$

$$= f(\mathbf{x}_t) + \frac{1+\eta\gamma}{2\eta}\|\mathbf{y}_t - \mathbf{x}_t\|^2 - \eta \cdot \frac{1-\ell\eta - \beta\ell\eta}{2}\|\nabla f(\mathbf{y}_t)\|^2 + \eta(\frac{\ell\eta}{2\beta} + \frac{1+\ell\eta}{2})\|\hat{\nabla} f(\mathbf{y}_t) - \nabla f(\mathbf{y}_t)\|^2.$$

Take $\beta = 1$ and $\eta \leq \frac{1}{4\ell}$, we have

$$f(\mathbf{x}_{t+1}) + \frac{1}{2\eta}\|\mathbf{x}_{t+1} - \mathbf{x}_t\|^2$$

$$\leq f(\mathbf{x}_t) + \frac{1+\eta\gamma}{2\eta}\|\mathbf{y}_t - \mathbf{x}_t\|^2 - \frac{\eta}{4}\|\nabla f(\mathbf{y}_t)\|^2 + \frac{3\eta}{4}\|\hat{\nabla} f(\mathbf{y}_t) - \nabla f(\mathbf{y}_t)\|^2$$

$$\leq f(\mathbf{x}_t) + \frac{1+\eta\gamma}{2\eta}\|\mathbf{y}_t - \mathbf{x}_t\|^2 - \frac{\eta}{4}\|\nabla f(\mathbf{y}_t)\|^2 + \eta \cdot \frac{\rho^2 d\mu^4}{48}$$

Using the fact that $\|\mathbf{y}_t - \mathbf{x}_t\| = (1-\theta)\|\mathbf{x}_t - \mathbf{x}_{t-1}\|$, we have

$$f(\mathbf{x}_{t+1}) + \frac{1}{2\eta}\|\mathbf{x}_{t+1} - \mathbf{x}_t\|^2$$

$$\leq f(\mathbf{x}_t) + \frac{1+\eta\gamma}{2\eta}(1-\theta)^2\|\mathbf{x}_t - \mathbf{x}_{t-1}\|^2 - \frac{\eta}{4}\|\nabla f(\mathbf{y}_t)\|^2 + \eta \cdot \frac{\rho^2 d\mu^4}{48}$$

$$= f(\mathbf{x}_t) + \frac{1}{2\eta}\|\mathbf{x}_t - \mathbf{x}_{t-1}\|^2 - \frac{2\theta - \theta^2 - \eta\gamma(1-\theta)^2}{2\eta}\|\mathbf{v}_t\|^2 - \frac{\eta}{4}\|\nabla f(\mathbf{y}_t)\|^2 + \eta \cdot \frac{\rho^2 d\mu^4}{48}$$

$$\leq f(\mathbf{x}_t) + \frac{1}{2\eta}\|\mathbf{x}_t - \mathbf{x}_{t-1}\|^2 - \frac{\theta}{2\eta}\|\mathbf{v}_t\|^2 - \frac{\eta}{4}\|\nabla f(\mathbf{y}_t)\|^2 + \eta \cdot \frac{\rho^2 d\mu^4}{48}$$

$\square$

**Lemma 4.** *Assume that $f(\cdot)$ is $\ell$-smooth and $\rho$-Hessian Lipschitz. Then, for each iteration $t$ where Eq. (6) holds, we have:*

$$E_{t+1} \leq E_t - \min\{\frac{s^2}{2\eta}, \frac{1}{2}\gamma s^2 - \rho s^3 - \frac{\rho^2 d\mu^4}{9\gamma}\}$$

*Proof.* When $\|\mathbf{v}_t\| \geq s$, then $\mathbf{x}_{t+1} = \mathbf{x}_t$, so we have

$$E_{t+1} = f(\mathbf{x}_{t+1}) = f(\mathbf{x}_t) = E_t - \frac{1}{2\eta}\|\mathbf{v}_t\|^2 \leq E_t - \frac{s^2}{2\eta}.$$

When $\|\mathbf{v}_t\| \leq s$ happens, we have

$$f(\mathbf{x}_t) = f(\mathbf{y}_t) + \langle \nabla f(\mathbf{y}_t), \mathbf{x}_t - \mathbf{y}_t\rangle + \frac{1}{2}(\mathbf{x}_t - \mathbf{y}_t)^{\mathsf{T}}\nabla^2 f(\zeta_t)(\mathbf{x}_t - \mathbf{y}_t),$$

where $\zeta_t = \mathbf{y}_t + \alpha(\mathbf{x}_t - \mathbf{y}_t)$, and $\alpha \in [0, 1]$. When the following condition holds:

$$f(\mathbf{x}_t) \leq f(\mathbf{y}_t) + \left\langle \hat{\nabla} f(\mathbf{y}_t), \mathbf{x}_t - \mathbf{y}_t\right\rangle - \frac{\gamma}{2}\|\mathbf{x}_t - \mathbf{y}_t\|^2$$

$$= f(\mathbf{y}_t) + \langle \nabla f(\mathbf{y}_t), \mathbf{x}_t - \mathbf{y}_t\rangle + \left\langle \hat{\nabla} f(\mathbf{y}_t) - \nabla f(\mathbf{y}_t), \mathbf{x}_t - \mathbf{y}_t\right\rangle - \frac{\gamma}{2}\|\mathbf{x}_t - \mathbf{y}_t\|^2,$$

we have

$$\frac{1}{2}(\mathbf{x}_t - \mathbf{y}_t)^{\mathsf{T}}\nabla^2 f(\zeta_t)(\mathbf{x}_t - \mathbf{y}_t) \leq \left\langle \hat{\nabla} f(\mathbf{y}_t) - \nabla f(\mathbf{y}_t), \mathbf{x}_t - \mathbf{y}_t\right\rangle - \frac{\gamma}{2}\|\mathbf{x}_t - \mathbf{y}_t\|^2$$

$$\leq \frac{1}{2} \left( \frac{1}{\beta} \|\hat{\nabla} f(\mathbf{y}_t) - \nabla f(\mathbf{y}_t)\|^2 + \beta \|\mathbf{x}_t - \mathbf{y}_t\|^2 \right) - \frac{\gamma}{2} \|\mathbf{x}_t - \mathbf{y}_t\|^2$$

$$= -\frac{\gamma - \beta}{2} \|\mathbf{x}_t - \mathbf{y}_t\|^2 + \frac{1}{2\beta} \|\hat{\nabla} f(\mathbf{y}_t) - \nabla f(\mathbf{y}_t)\|^2$$

$$\leq -\frac{\gamma - \beta}{2} \|\mathbf{x}_t - \mathbf{y}_t\|^2 + \frac{\rho^2 d\mu^4}{18\beta}.$$

Take $\beta = \frac{\gamma}{2}$ we have

$$\frac{1}{2} (\mathbf{x}_t - \mathbf{y}_t)^\mathsf{T} \nabla^2 f(\zeta_t)(\mathbf{x}_t - \mathbf{y}_t) \leq -\frac{\gamma}{4} \|\mathbf{x}_t - \mathbf{y}_t\|^2 + \frac{\rho^2 d\mu^4}{9\gamma}$$

Note that $\min\{\langle \nabla f(\mathbf{x}_t), \delta \rangle, \langle \nabla f(\mathbf{x}_t), -\delta \rangle\} \leq 0$. Without loss of generality, we assume that $\langle \nabla f(\mathbf{x}_t), \delta \rangle \leq 0$. Since $\mathbf{x}_{t+1} = \arg\min_{\mathbf{x} \in \{\mathbf{x}_t + \delta, \mathbf{x}_t - \delta\}} f(\mathbf{x})$, we have

$$f(\mathbf{x}_{t+1}) \leq f(\mathbf{x}_t + \delta) = f(\mathbf{x}_t) + \langle \nabla f(\mathbf{x}_t), \delta \rangle + \frac{1}{2} \delta^\mathsf{T} \nabla^2 f(\zeta_t') \delta \leq f(\mathbf{x}_t) + \frac{1}{2} \delta^\mathsf{T} \nabla^2 f(\zeta_t') \delta,$$

where $\zeta_t' = \mathbf{x}_t + \alpha' \delta$ and $\alpha' \in [0, 1]$. Since $\|\zeta_t - \zeta_t'\| \leq 2s$ and $\delta$ lines up with $\mathbf{y}_t - \mathbf{x}_t$, we have

$$\delta^\mathsf{T} \nabla^2 f(\zeta_t') \delta \leq \delta^\mathsf{T} \nabla^2 f(\zeta_t) \delta + \|\nabla^2 f(\zeta_t') - \nabla^2 f(\zeta_t)\| \|\delta\|^2 \leq -\frac{\gamma}{2} \|\delta\|^2 + 2\rho s \|\delta\|^2 + \frac{2\rho^2 d\mu^4}{9\gamma}$$

$$= -\frac{\gamma}{2} s^2 + 2\rho s^3 + \frac{2\rho^2 d\mu^4}{9\gamma}$$

Finally we get

$$E_{t+1} = f(\mathbf{x}_{t+1}) \leq f(\mathbf{x}_t) - \left( \frac{1}{4} \gamma s^2 - \rho s^3 - \frac{\rho^2 d\mu^4}{9\gamma} \right) \leq E_t - \left( \frac{1}{4} \gamma s^2 - \rho s^3 - \frac{\rho^2 d\mu^4}{9\gamma} \right).$$

□

**Lemma 18.** *If the Eq. (6) does not holds, then for all steps in $[t, t+T]$, we have:*

$$\sum_{\tau=t+1}^{t+T} \|\mathbf{x}_\tau - \mathbf{x}_{\tau-1}\|^2 \leq \frac{2\eta}{\theta} (E_t - E_{t+T}) + \frac{\eta \rho^2 d\mu^4}{48} T$$

*Proof.* The proof directly follows from the results of Lemma 3.

□

## C   PROOF OF MAIN RESULTS OF ALGORITHM 1

Recall the parameter settings in Algorithm 4,

$$\eta = \frac{1}{4\ell}, \quad \theta = \frac{1}{4\sqrt{\kappa}}, \quad \gamma = \frac{\theta^2}{\eta} = \frac{\sqrt{\rho\epsilon}}{4}, \quad s = \frac{\gamma}{4\rho} = \frac{1}{16} \sqrt{\frac{\epsilon}{\rho}}, \quad r = \eta \epsilon \chi^{-5} c^{-8},$$

where $\kappa = \frac{\ell}{\sqrt{\rho\epsilon}}$. Denote

$$\mathscr{T} = \sqrt{\kappa} \cdot \chi c, \quad \mathscr{E} = \sqrt{\frac{\epsilon^3}{\rho}} \cdot \chi^{-5} c^{-7}, \quad \mathscr{S} = \sqrt{\frac{2\epsilon}{\rho}} \chi^{-2} c^{-3}, \quad \mathscr{M} = \frac{\epsilon\sqrt{\kappa}}{\ell} c^{-1}$$

**Lemma 19.** *After running the NCE with $\mu \leq \tilde{\mathcal{O}}(\frac{\epsilon^{1/2}}{d^{1/4}})$ for one step, we have*

$$E_{t+1} - E_t \leq -2\mathscr{E}.$$

*Proof.* According to Lemma 4, with the choice of the smoothing parameter such that $\mu \leq \tilde{\mathcal{O}}(\frac{\epsilon^{1/2}}{d^{1/4}})$, we have

$$E_{t+1} - E_t \leq -\min\left\{ \frac{s^2}{2\eta}, \frac{1}{2} \gamma s^2 - \rho s^3 - \frac{\rho^2 d\mu^4}{9\gamma} \right\} \leq -\Omega(\mathscr{E} c^7) \geq 2\mathscr{E}.$$

□

**Lemma 20.** *Let $\mathbf{0}$ be an origin point. Denote*

$$\delta_\tau = \hat{\nabla} f(\mathbf{y}_\tau) - \hat{\nabla} f(\mathbf{0}) - \nabla^2 f(\mathbf{0}) \mathbf{y}_\tau$$

*Then the zeroth-order AGD update can be rewritten as:*

$$\begin{pmatrix} \mathbf{x}_{t+1} \\ \mathbf{x}_t \end{pmatrix} = \mathbf{A}^t \begin{pmatrix} \mathbf{x}_1 \\ \mathbf{x}_0 \end{pmatrix} - \eta \sum_{\tau=1}^{t} \mathbf{A}^{t-\tau} \begin{pmatrix} \hat{\nabla} f(\mathbf{0}) + \delta_\tau \\ 0 \end{pmatrix} \quad (11)$$

*, where $\mathbf{A} = \begin{pmatrix} (2-\theta)(\mathbf{I} - \eta\nabla^2 f(\mathbf{0})) & -(1-\theta)(\mathbf{I} - \eta\nabla^2 f(\mathbf{0})) \\ \mathbf{I} & 0 \end{pmatrix}$.*

*Proof.*

$$\mathbf{x}_{t+1} = (2-\theta)\mathbf{x}_t - (1-\theta)\mathbf{x}_{t-1} - \eta\hat{\nabla} f((2-\theta)\mathbf{x}_t - (1-\theta)\mathbf{x}_{t-1})$$

Then we have

$$\begin{pmatrix} \mathbf{x}_{t+1} \\ \mathbf{x}_t \end{pmatrix} = \begin{pmatrix} (2-\theta)(\mathbf{I} - \eta\nabla^2 f(\mathbf{0})) & -(1-\theta)(\mathbf{I} - \eta\nabla^2 f(\mathbf{0})) \\ \mathbf{I} & 0 \end{pmatrix} \begin{pmatrix} \mathbf{x}_t \\ \mathbf{x}_{t-1} \end{pmatrix} - \eta \begin{pmatrix} \hat{\nabla} f(\mathbf{0}) + \delta_t \\ 0 \end{pmatrix}$$

$$= \mathbf{A}^t \begin{pmatrix} \mathbf{x}_1 \\ \mathbf{x}_0 \end{pmatrix} - \eta \sum_{\tau=1}^{t} \mathbf{A}^{t-\tau} \begin{pmatrix} \hat{\nabla} f(\mathbf{0}) + \delta_\tau \\ 0 \end{pmatrix}$$

$\square$

**Lemma 21.** *If for any $\tau \leq t$, we have $\|\mathbf{x}_\tau\| \leq R$, then for any $\tau \leq t$, we have*

1. $\|\delta_\tau\| \leq \rho\mathcal{O}(R^2 + \sqrt{d}\mu^2)$

2. $\|\delta_\tau - \delta_{\tau-1}\| \leq \rho\mathcal{O}(R\|\mathbf{x}_\tau - \mathbf{x}_{\tau-1}\| + R\|\mathbf{x}_{\tau-1} - \mathbf{x}_{\tau-1}\| + \sqrt{d}\mu^2)$

3. $\sum_{\tau=1}^{t} \|\delta_\tau - \delta_{\tau-1}\|^2 \leq \mathcal{O}(\rho^2 R^2 \sum_{\tau-1}^{t} \|\mathbf{x}_\tau - \mathbf{x}_{\tau-1}\|^2 + t\rho^2 d\mu^4)$

*Proof.* For the first inequality, by using the second inequality of Lemma 9, we have

$$\|\nabla f(\mathbf{y}_\tau) - \nabla f(\mathbf{0}) - \nabla^2 f(\mathbf{0})\mathbf{y}_\tau\| \leq \frac{\rho}{2}\|\mathbf{y}_\tau\|^2 = \frac{\rho}{2}\|(2-\theta)\mathbf{x}_\tau - (1-\theta)\mathbf{x}_{\tau-1}\|^2 \leq \mathcal{O}(\rho R^2).$$

Using Lemma 1, we have

$$\begin{aligned}
\|\delta_\tau\| =& \|\hat{\nabla} f(\mathbf{y}_\tau) - \hat{\nabla} f(\mathbf{0}) - \nabla^2 f(\mathbf{0})\mathbf{y}_\tau\| \\
\leq& \|\nabla f(\mathbf{y}_\tau) - \nabla f(\mathbf{0}) - \nabla^2 f(\mathbf{0})\mathbf{y}_\tau\| + \|\hat{\nabla} f(\mathbf{y}_\tau) - \hat{\nabla} f(\mathbf{0}) - (\nabla f(\mathbf{y}_\tau) - \nabla f(\mathbf{0}))\| \\
\leq& \mathcal{O}(\rho R^2 + \sqrt{d}\rho\mu^2).
\end{aligned}$$

For the second inequality, we have

$$\delta_\tau - \delta_{\tau-1} = \hat{\nabla} f(\mathbf{y}_\tau) - \hat{\nabla} f(\mathbf{y}_{\tau-1}) - \nabla^2 f(\mathbf{0})(\mathbf{y}_\tau - \mathbf{y}_{\tau-1}).$$

Then we have

$$\begin{aligned}
& \|\nabla f(\mathbf{y}_\tau) - \nabla f(\mathbf{y}_{\tau-1}) - \nabla^2 f(\mathbf{0})(\mathbf{y}_\tau - \mathbf{y}_{\tau-1})\| \\
=& \|\int_0^1 (\nabla^2 f(\mathbf{x}_{\tau-1} + \theta(\mathbf{y}_\tau - \mathbf{y}_{\tau-1})) - \nabla^2 f(\mathbf{0}))d\theta(\mathbf{y}_\tau - \mathbf{y}_{\tau-1})\| \\
\leq& \|\int_0^1 (\nabla^2 f(\mathbf{y}_{\tau-1} + \theta(\mathbf{y}_\tau - \mathbf{y}_{\tau-1})) - \nabla^2 f(\mathbf{0}))d\theta\| \cdot \|\mathbf{y}_\tau - \mathbf{y}_{\tau-1}\| \leq \rho\max\{\|\mathbf{y}_\tau\|, \|\mathbf{y}_{\tau-1}\|\}\|\mathbf{y}_\tau - \mathbf{y}_{\tau-1}\| \\
\leq& \mathcal{O}(\rho R)(\|\mathbf{x}_\tau - \mathbf{x}_{\tau-1}\| + \|\mathbf{x}_{\tau-1} - \mathbf{x}_{\tau-2}\|).
\end{aligned}$$

Thus,

$$\|\delta_\tau - \delta_{\tau-1}\|$$

$$\leq \|\hat{\nabla} f(\mathbf{y}_\tau) - \hat{\nabla} f(\mathbf{y}_{\tau-1}) - (\nabla f(\mathbf{y}_\tau) - \nabla f(\mathbf{y}_{\tau-1}))\| + \|\nabla f(\mathbf{y}_\tau) - \nabla f(\mathbf{y}_{\tau-1}) - \nabla^2 f(\mathbf{0})(\mathbf{y}_\tau - \mathbf{y}_{\tau-1})\|$$
$$\leq \mathcal{O}(\rho R)(\|\mathbf{x}_\tau - \mathbf{x}_{\tau-1}\| + \|\mathbf{x}_{\tau-1} - \mathbf{x}_{\tau-2}\|) + \mathcal{O}(\rho\sqrt{d}\mu^2)$$

Then we have

$$\sum_{\tau=1}^{t} \|\delta_\tau - \delta_{\tau-1}\|^2 \leq \mathcal{O}(\rho^2 R^2 \sum_{\tau-1}^{t} \|\mathbf{x}_\tau - \mathbf{x}_{\tau-1}\|^2 + \sum_{\tau=1}^{t} \rho^2 d\mu^4)$$

$\square$

## C.1 LARGE GRADIENT

Let $\mathcal{S}$ be the subspace with eigenvalues in $(\frac{\theta^2}{\eta(2-\theta)^2}, \ell]$ and $\mathcal{S}^c$ be the complementary subspace.

**Lemma 22 (Large momentum or large gradient).** *If $\|\mathbf{v}_t\| \geq \mathcal{M}$ or $\|\nabla f(\mathbf{x}_t)\| \geq 2\ell\mathcal{M}$, and at iteration $t$ only AGD is used with smoothing parameter $\mu \leq \mathcal{O}(\frac{\epsilon^{1/2}\kappa^{1/8}}{\rho^{1/2}d^{1/4}}c^{-1/2})$ and without NCE or perturbation, we have:*

$$E_{t+1} - E_t \leq -\frac{4\mathscr{E}}{\mathscr{T}}.$$

*Proof.* When $\|\mathbf{v}_t\| \geq \frac{\epsilon\sqrt{\kappa}}{\ell}$ and $\mu \leq \mathcal{O}(\frac{\epsilon^{1/2}\kappa^{1/8}}{\rho^{1/2}d^{1/4}}c^{-1/2})$, using Lemma 3, we have

$$E_{t+1} - E_t \leq -\frac{\theta}{2\eta}\|\mathbf{v}_t\|^2 + \frac{\eta\rho^2 d\mu^4}{48} \leq -\Omega(\frac{\ell}{\sqrt{\kappa}}\frac{\epsilon^2\kappa}{\ell^2}c^{-2} - \frac{\rho^2 d\mu^4}{\ell}) = -\Omega(\frac{\epsilon^2\sqrt{\kappa}c^{-2} - \rho^2 d\mu^4}{\ell}) \leq -\Omega(\frac{\epsilon^2\sqrt{\kappa}c^{-2}}{\ell})$$
$$\leq -\Omega(\frac{\mathscr{E}}{\mathscr{T}}c^6) \leq -\frac{4\mathscr{E}}{\mathscr{T}},$$

holds for large enough constant $c$. When $\|\mathbf{v}_t\| \leq \mathcal{M}$ and $\|\nabla f(\mathbf{x}_t)\| \geq 2\ell\mathcal{M}$, then by gradient Lipschitz assumption, we have

$$\|\nabla f(\mathbf{y}_t)\| \geq \|\nabla f(\mathbf{x}_t)\| - \|\nabla f(\mathbf{y}_t) - f(\mathbf{x}_t)\| \geq \|\nabla f(\mathbf{x}_t)\| - \ell(1-\theta)\|\mathbf{v}_t\| \geq \ell\mathcal{M}.$$

Using Lemma 3, with $\mu \leq \mathcal{O}(\frac{\epsilon^{1/2}\kappa^{1/8}}{\rho^{1/2}d^{1/4}}c^{-1/2})$, we have

$$E_{t+1} - E_t \leq -\frac{\eta}{4}\|\nabla f(\mathbf{y}_t)\|^2 + \frac{\eta\rho^2 d\mu^4}{48} \leq -\Omega(\frac{\epsilon^2\kappa c^{-2} - \rho^2 d\mu^4}{\ell}) \leq -\Omega(\frac{\epsilon^2\sqrt{\kappa}c^{-2} - \rho^2 d\mu^4}{\ell})$$
$$\leq -\Omega(\frac{\epsilon^2\sqrt{\kappa}c^{-2}}{\ell}) \leq -\Omega(\frac{\mathscr{E}}{\mathscr{T}}c^6) \leq -\frac{4\mathscr{E}}{\mathscr{T}},$$

holds for large enough constant $c$.

$\square$

**Lemma 23.** *If $\|\mathcal{P}_{\mathcal{S}^c}\nabla f(\mathbf{x}_0)\| \geq \frac{\epsilon}{4}$, $\|\mathbf{v}_0\| \leq \mathcal{M}$, $\mathbf{v}_0^{\mathsf{T}}[\mathcal{P}_{\mathcal{S}}^{\mathsf{T}}\nabla^2 f(\mathbf{x}_0)\mathcal{P}_{\mathcal{S}}]\mathbf{v}_0 \leq 2\sqrt{\rho\epsilon}\mathcal{M}^2$, $\mu \leq \tilde{\mathcal{O}}(\frac{\epsilon^{5/8}}{d^{1/4}})$, and for $t \in [0, \mathscr{T}/4]$ only AGD steps are used, then we have*

$$E_{\mathscr{T}/4} - E_0 \leq -\mathscr{E}.$$

*Proof.* Define $\mathbf{x}_{-1} = \mathbf{x}_0 - \mathbf{v}_0$. Without loss of generality, set $\mathbf{x}_0 = \mathbf{0}$. Using Lemma 20, we have

$$\begin{pmatrix} \mathbf{x}_t \\ \mathbf{x}_{t-1} \end{pmatrix} = \mathbf{A}^{t-1}\begin{pmatrix} \mathbf{0} \\ -\mathbf{v}_0 \end{pmatrix} - \eta\sum_{\tau=0}^{t-1}\mathbf{A}^{t-1-\tau}\begin{pmatrix} \hat{\nabla} f(\mathbf{0}) + \delta_\tau \\ 0 \end{pmatrix}$$

Denote

$$\mathbf{A}_j = \begin{pmatrix} (2-\theta)(1-\eta\lambda_j) & -(1-\theta)(1-\eta\lambda_j) \\ 1 & 0 \end{pmatrix},$$

where $\lambda_j$ is the $j$-th eigenvalue of $\nabla^2 f(\mathbf{0})$. Denote

$$\begin{pmatrix} a_t^{(j)} & -b_t^{(j)} \end{pmatrix} = (1 \quad 0) \, \mathbf{A}_j^t.$$

Then we have for the $j$-th eigen-direction

$$x_t^{(j)} = b_t^{(j)} v_0^{(j)} - \eta \sum_{\tau=0}^{t-1} (\hat{\nabla} f(\mathbf{0}) + \delta_\tau^{(j)}) = -\eta \left[ \sum_{\tau=0}^{t-1} a_\tau^{(j)} \right] \left( \hat{\nabla} f(\mathbf{0})^{(j)} + \sum_{\tau=0}^{t-1} p_\tau^{(j)} \delta_\tau^{(j)} + q_t^{(j)} v_0^{(j)} \right),$$

where

$$p_\tau^{(j)} = \frac{a_{t-1}^{(j)}}{\sum_{\tau=0}^{t-1-\tau} a_\tau^{(j)}}, \qquad q_t^{(j)} = -\frac{b_t^{(j)}}{\eta \sum_{\tau=0}^{t-1-\tau} a_\tau^{(j)}}$$

For $j \in \mathcal{S}^c$, using Lemma , we have $\sum_{\tau=0}^{t-1} a_\tau^{(j)} \geq \Omega(\frac{1}{\theta^2})$. Then rewrite the above equation as

$$x_t^{(j)} = -\eta \left[ \sum_{\tau=0}^{t-1} a_\tau^{(j)} \right] \left( \hat{\nabla} f(\mathbf{0})^{(j)} + \tilde{\delta}^{(j)} + \tilde{v}^{(j)} \right),$$

where $\tilde{\delta}^{(j)} = \sum_{\tau=0}^{t-1} p_\tau^{(j)} \delta_\tau^{(j)}, \tilde{v}^{(j)} = q_t^{(j)} v_0^{(j)}$.

For all $j \in \mathcal{S}^c$,

$$|\tilde{\delta}^{(j)}| = |\sum_{\tau=0}^{t-1} p_\tau^{(j)} \delta_\tau^{(j)}| \leq \sum_{\tau=0}^{t-1} p_\tau^{(j)} (|\delta_0^{(j)}| + |\delta_\tau^{(j)} - \delta_0^{(j)}|) = |\delta_0^{(j)}| + \sum_{\tau=0}^{t-1} p_\tau^{(j)} |\delta_\tau^{(j)} - \delta_0^{(j)}| \leq |\delta_0^{(j)}| + \sum_{\tau=1}^{t-1} |\delta_\tau^{(j)} - \delta_{\tau-1}^{(j)}|$$

Then by Cauchy-Swartz inequality,

$$\|\mathcal{P}_{\mathcal{S}^c} \tilde{\delta}\|^2 = \sum_{j \in \mathcal{S}^c} |\tilde{\delta}^{(j)}|^2 \leq \sum_{j \in \mathcal{S}^c} (|\delta_0^{(j)}| + \sum_{\tau=1}^{t-1} |\delta_\tau^{(j)} - \delta_{\tau-1}^{(j)}|)^2 \leq 2 \left[ \sum_{j \in \mathcal{S}^c} |\delta_0^{(j)}|^2 + \sum_{j \in \mathcal{S}^c} (\sum_{\tau=1}^{t-1} |\delta_\tau^{(j)} - \delta_{\tau-1}^{(j)}|)^2 \right]$$

$$\leq 2 \left[ \sum_{j \in \mathcal{S}^c} |\delta_0^{(j)}|^2 + t \sum_{j \in \mathcal{S}^c} \sum_{\tau=1}^{t-1} (|\delta_\tau^{(j)} - \delta_{\tau-1}^{(j)}|)^2 \right] \leq 2 \|\delta_0\|^2 + 2t \sum_{\tau=1}^{t-1} \|\delta_\tau - \delta_{\tau-1}\|^2$$

Assume that $E_{\mathcal{T}/4} - E_0 \geq -\mathscr{E}$. By Lemma 18 and choose $\mu \leq \mathcal{O}((\sqrt{\frac{\epsilon^3}{\rho^5}} \frac{\chi^{-6} c^{-8}}{d})^{1/4}) = \tilde{O}(\frac{\epsilon^{3/8}}{(d)^{1/4}})$, we have $\|\mathbf{x}_t - \mathbf{x}_0\| \leq \sqrt{t \sum_{\tau=1}^{t} \|\mathbf{x}_\tau - \mathbf{x}_{\tau-1}\|^2} \leq \sqrt{\frac{2\eta\mathscr{E}}{\theta} \cdot \frac{\mathcal{T}}{4} + \frac{\mathcal{T}^2}{16} \frac{\eta\rho^2 d\mu^4}{48}} \leq \mathscr{S}$. With $\mu \leq \mathcal{O}((\sqrt{\frac{\epsilon^5}{\rho^3}} \frac{\chi^{-10} c^{-14}}{d\ell})^{1/4}) = \tilde{O}(\frac{\epsilon^{5/8}}{(d)^{1/4}})$, by Lemma 21 we have $\|\delta_0\| \leq \mathcal{O}(\rho\mathscr{S}^2)$. By Lemma 18 and Lemma 21, we have

$$t \sum_{\tau=1}^{t-1} \|\delta_\tau - \delta_{\tau-1}\|^2 \leq \mathcal{O}(\rho^2 \mathscr{S}^2 t \sum_{\tau=1}^{t-1} \|\mathbf{x}_\tau - \mathbf{x}_{\tau-1}\|^2 + t^2 \rho^2 d\mu^4) \leq \mathcal{O}(\rho^2 \mathscr{S}^4)$$

So we have $\|\mathcal{P}_{\mathcal{S}^c} \tilde{\delta}\| \leq \mathcal{O}(\rho\mathscr{S}^2) \leq \mathcal{O}(\epsilon c^{-6})$.

By Lemma 11, $-\eta q_t^{(j)} = \frac{b_t}{\sum_{\tau=0}^{t-1} a_\tau} \leq \mathcal{O}(1) \max\{\theta, \sqrt{\eta|\lambda_j|}\}$, then

$$\|\mathcal{P}_{\mathcal{S}^c} \tilde{v}\|^2 = \sum_{j \in \mathcal{S}^c} [q_t^{(j)} v_0^{(j)}]^2 \leq \mathcal{O}(1) \sum_{j \in \mathcal{S}^c} \frac{\max\{\theta^2, \eta|\lambda_j|\}}{\eta^2} [v_0^{(j)}]^2$$

Since the NCE step is not reached, then we have:

$$f(\mathbf{x}_0) \geq f(\mathbf{y}_0) + \left\langle \hat{\nabla} f(\mathbf{y}_0), \mathbf{x}_0 - \mathbf{y}_0 \right\rangle - \frac{\gamma}{2} \|\mathbf{x}_0 - \mathbf{y}_0\|^2$$

$$= f(\mathbf{y}_0) + \langle \nabla f(\mathbf{y}_0), \mathbf{x}_0 - \mathbf{y}_0 \rangle + \left\langle \hat{\nabla} f(\mathbf{y}_0) - \nabla f(\mathbf{y}_0), \mathbf{x}_0 - \mathbf{y}_0 \right\rangle - \frac{\gamma}{2} \|\mathbf{x}_0 - \mathbf{y}_0\|^2$$

$$\geq f(\mathbf{y}_0) + \langle \nabla f(\mathbf{y}_0), \mathbf{x}_0 - \mathbf{y}_0 \rangle - \frac{1}{2\beta} \|\hat{\nabla} f(\mathbf{y}_0) - \nabla f(\mathbf{y}_0)\|^2 - \frac{\gamma + \beta}{2} \|\mathbf{x}_0 - \mathbf{y}_0\|^2$$

$$\geq f(\mathbf{y}_0) + \langle \nabla f(\mathbf{y}_0), \mathbf{x}_0 - \mathbf{y}_0 \rangle - \frac{\rho^2 d\mu^4}{72\beta} - \frac{\gamma + \beta}{2}\|\mathbf{x}_0 - \mathbf{y}_0\|^2$$

$$= f(\mathbf{y}_0) + \langle \nabla f(\mathbf{y}_0), \mathbf{x}_0 - \mathbf{y}_0 \rangle - \frac{\rho^2 d\mu^4}{72\gamma} - \gamma\|\mathbf{x}_0 - \mathbf{y}_0\|^2,$$

where the last step is by taking $\beta = \gamma$. Then we have

$$\frac{1}{2}(\mathbf{x}_0 - \mathbf{y}_0)^\mathsf{T}\nabla^2 f(\zeta_0)(\mathbf{x}_0 - \mathbf{y}_0) \geq -\frac{\rho^2 d\mu^4}{72\gamma} - \gamma\|\mathbf{x}_0 - \mathbf{y}_0\|^2,$$

where $\zeta_0 = \phi\mathbf{x}_0 + (1-\phi)\mathbf{y}_0$ and $\phi \in [0,1]$. Note that $(1-\theta)\mathbf{v}_0 = \mathbf{y}_0 - \mathbf{x}_0$, we have

$$\frac{1}{2}\mathbf{v}_0^\mathsf{T}\nabla^2 f(\zeta_0)\mathbf{v}_0 \geq -\frac{\rho^2 d\mu^4}{72(1-\theta)^2\gamma} - \gamma\|\mathbf{v}_0\|^2 \geq -\frac{\rho^2 d\mu^4}{18\gamma} - \gamma\|\mathbf{v}_0\|^2,$$

where the last inequality uses the fact that $\theta \leq \frac{1}{2}$. Using the Hessian Lipschitz property, we have

$$\|\nabla^2 f(\zeta_0) - \nabla^2 f(\mathbf{x}_0)\| \leq \rho\|\mathbf{y}_0\| \leq \rho\|\mathbf{v}_0\| \leq \rho\mathscr{M} = \frac{(\rho\epsilon)^{3/4}}{\sqrt{\ell}}c^{-1} \leq \frac{\sqrt{\rho\epsilon}}{2} = 2\gamma.$$

Then we have

$$\mathbf{v}_0^\mathsf{T}\nabla^2 f(\mathbf{x}_0)\mathbf{v}_0 \geq -\frac{\rho^2 d\mu^4}{9\gamma} - 4\gamma\|\mathbf{v}_0\|^2 \geq -\frac{\rho^2 d\mu^4}{\sqrt{\rho\epsilon}} - \sqrt{\rho\epsilon}\|\mathbf{v}_0\|^2.$$

Since $\frac{\theta^2}{\eta(1-\theta)^2} = \Theta(\sqrt{\rho\epsilon})$, we have

$$\sum_{j\in\mathcal{S}^c}|\lambda_j|[v_0^{(j)}]^2 \leq \sqrt{\rho\epsilon}\|\mathbf{v}_0\|^2 + \frac{\rho^2 d\mu^4}{\sqrt{\rho\epsilon}} + \sum_{j:0<\lambda_j\leq\frac{\theta^2}{\eta(1-\theta)^2}}\lambda_j[v_0^{(j)}]^2 + \sum_{j:\lambda_j>\frac{\theta^2}{\eta(1-\theta)^2}}\lambda_j[v_0^{(j)}]^2$$

$$\leq \mathcal{O}(\sqrt{\rho\epsilon})\|\mathbf{v}_0\|^2 + \frac{\rho^2 d\mu^4}{\sqrt{\rho\epsilon}} + \mathbf{v}_0^\mathsf{T}[\mathcal{P}_\mathcal{S}^\mathsf{T}\nabla^2 f(\mathbf{0})\mathcal{P}_\mathcal{S}]\mathbf{v}_0.$$

With $\mu \leq \mathcal{O}(\frac{\epsilon^2\sqrt{\rho\epsilon}}{\ell\rho^2 d})^{1/4} = \tilde{\mathcal{O}}(\frac{\epsilon^{5/8}}{d^{1/4}})$, then we have

$$\|\mathcal{P}_{\mathcal{S}^c}\tilde{\mathbf{v}}\|^2 \leq \mathcal{O}(\frac{1}{\eta})\left[\sqrt{\rho\epsilon}\|\mathbf{v}_0\|^2 + \frac{\rho^2 d\mu^4}{\sqrt{\rho\epsilon}} + \mathbf{v}_0^\mathsf{T}[\mathcal{P}_\mathcal{S}^\mathsf{T}\nabla^2 f(\mathbf{0})\mathcal{P}_\mathcal{S}]\mathbf{v}_0\right] \leq \mathcal{O}(\ell\sqrt{\rho\epsilon}\mathscr{M}^2) = \mathcal{O}(\epsilon^2 c^{-2})$$

Then we have

$$\|\mathbf{x}_t\| \geq \|\mathcal{P}_{\mathcal{S}^c}\mathbf{x}_t\| \geq \eta\left[\min_{j\in\mathcal{S}^c}\sum_{\tau=0}^{t-1}a_\tau^{(j)}\right]\|\mathcal{P}_{\mathcal{S}^c}(\hat{\nabla}f(\mathbf{0}) + \tilde{\delta} + \tilde{\mathbf{v}})\|$$

$$\geq \Omega(\frac{\eta}{\theta^2})\left[\|\mathcal{P}_{\mathcal{S}^c}\nabla f(\mathbf{0})\| - \frac{\rho\sqrt{d}\mu^2}{6} - \|\mathcal{P}_{\mathcal{S}^c}\tilde{\delta}\| - \|\mathcal{P}_{\mathcal{S}^c}\tilde{\mathbf{v}}\|\right]$$

$$\geq \Omega(\frac{\eta\epsilon}{\theta^2}) \geq \mathscr{S},$$

which contradicts with $\|\mathbf{x}_t - \mathbf{x}_0\| = \|\mathbf{x}_t\| \leq \mathscr{S}$. So we have

$$E_{\mathscr{T}/4} - E_0 \leq -\mathscr{E}.$$

$\square$

**Lemma 24.** *If $\|\mathbf{v}_0\| \leq \mathscr{M}$ and $\|\nabla f(\mathbf{x}_0)\| \leq 2\ell\mathscr{M}, E_{\mathscr{T}/2} - E_0 \geq -\mathscr{E}, \mu \leq \tilde{\mathcal{O}}(\frac{\epsilon^{5/8}}{d^{1/4}})$ and for any $t \in [0, \mathscr{T}/2]$ only SGD steps are used. Then $\forall t \in [\mathscr{T}/4, \mathscr{T}/2]$:*

$$\|\mathcal{P}_\mathcal{S}\nabla f(\mathbf{x}_t)\| \leq \frac{\epsilon}{4} \quad and \quad \mathbf{v}_t^\mathsf{T}[\mathcal{P}_\mathcal{S}^\mathsf{T}\nabla^2 f(\mathbf{x}_0)\mathcal{P}_\mathcal{S}]\mathbf{v}_t \leq \sqrt{\rho\epsilon}\mathscr{M}^2.$$

*Proof.* Since $E_{\mathscr{T}/4} - E_0 \geq -\mathscr{E}$. By Lemma 18 and choose $\mu \leq \mathcal{O}((\sqrt{\frac{\epsilon^3}{\rho^5}} \frac{\chi^{-6} c^{-8}}{d})^{1/4}) = \tilde{O}(\frac{\epsilon^{3/8}}{(d)^{1/4}})$, we have $\|\mathbf{x}_t - \mathbf{x}_0\| \leq \sqrt{t \sum_{\tau=1}^{t} \|\mathbf{x}_\tau - \mathbf{x}_{\tau-1}\|^2} \leq \sqrt{\frac{2\eta\mathscr{E}}{\theta} \cdot \frac{\mathscr{T}}{4} + \frac{\mathscr{T}^2}{16} \frac{\eta\rho^2 d\mu^4}{48}} \leq \mathscr{S}$. Define $\mathbf{x}_{-1} = \mathbf{x}_0 - \mathbf{v}_0$. Without loss of generality, set $\mathbf{x}_0 = \mathbf{0}$. Using Lemma 20, we have

$$\begin{pmatrix} \mathbf{x}_t \\ \mathbf{x}_{t-1} \end{pmatrix} = \mathbf{A}^{t-1} \begin{pmatrix} \mathbf{0} \\ -\mathbf{v}_0 \end{pmatrix} - \eta \sum_{\tau=0}^{t-1} \mathbf{A}^{t-1-\tau} \begin{pmatrix} \hat{\nabla} f(\mathbf{0}) + \delta_\tau \\ 0 \end{pmatrix}$$

Define $\Delta_t = \int_0^1 (\nabla^2 f(\phi \mathbf{x}_t) - \nabla^2 f(\mathbf{0})) d\phi$. Then we have

$$\nabla f(\mathbf{x}_t) = \nabla f(\mathbf{0}) + (\nabla^2 f(\mathbf{0}) + \Delta_t)\mathbf{x}_t = \hat{\nabla} f(\mathbf{0}) + (\nabla^2 f(\mathbf{0}) + \Delta_t)\mathbf{x}_t + \nabla f(\mathbf{0}) - \hat{\nabla} f(\mathbf{0})$$

$$= \left( \mathbf{I} - \eta \nabla^2 f(\mathbf{0}) \left(\mathbf{I} \quad \mathbf{0}\right) \sum_{\tau=0}^{t-1} \mathbf{A}^{t-1-\tau} \begin{pmatrix} \mathbf{I} \\ \mathbf{0} \end{pmatrix} \right) \hat{\nabla} f(\mathbf{0}) + \nabla^2 f(\mathbf{0}) \left(\mathbf{I} \quad \mathbf{0}\right) \mathbf{A}^t \begin{pmatrix} \mathbf{0} \\ -\mathbf{v}_0 \end{pmatrix}$$

$$- \eta \nabla^2 f(\mathbf{0}) \left(\mathbf{I} \quad \mathbf{0}\right) \sum_{\tau=0}^{t-1} \mathbf{A}^{t-1-\tau} \begin{pmatrix} \delta_t \\ \mathbf{0} \end{pmatrix} + \Delta_t \mathbf{x}_t + \nabla f(\mathbf{0}) - \hat{\nabla} f(\mathbf{0}).$$

If we choose $\mu \leq \tilde{\mathcal{O}}(\frac{\epsilon^{1/2}}{d^{1/4}})$, we have

$$\|\Delta_t \mathbf{x}_t\| \leq \rho \|\mathbf{x}_t\|^2 \leq \mathcal{O}(\rho \mathscr{S}^2) \leq \mathcal{O}(\epsilon c^{-6)} \leq \epsilon/20$$

$$\|\nabla f(\mathbf{0}) - \hat{\nabla} f(\mathbf{0})\| \leq \frac{\rho\sqrt{d}\mu^2}{6} \leq \epsilon/20$$

By Lemma 11, we have

$$1 - \eta\lambda_j \left(1 \quad 0\right) \sum_{\tau=0}^{t-1} \mathbf{A}_j^{t-1-\tau} \begin{pmatrix} 1 \\ 0 \end{pmatrix} = \left(1 \quad 0\right) \mathbf{A}_j^t \begin{pmatrix} 1 \\ 1 \end{pmatrix}.$$

Denote

$$\left(a_t^{(j)}, \quad -b_t^{(j)}\right) = \left(1 \quad 0\right) \mathbf{A}_j^t.$$

By Lemma 12, $\max_{j \in \mathcal{S}} \left\{|a_t^{(j)}|, |b_t^{(j)}|\right\} \leq (t+1)(1-\theta)^{t/2}$, then we have when $t \geq \mathscr{T}/4 = \Omega(\frac{2}{\theta} \log \frac{1}{\theta})$, $\mu \leq \tilde{\mathcal{O}}(\frac{\epsilon^{1/2}}{d^{1/4}})$,

$$\|\mathcal{P}_{\mathcal{S}} \left( \left( \mathbf{I} - \eta \nabla^2 f(\mathbf{0}) \left(\mathbf{I} \quad \mathbf{0}\right) \sum_{\tau=0}^{t-1} \mathbf{A}^{t-1-\tau} \begin{pmatrix} \mathbf{I} \\ \mathbf{0} \end{pmatrix} \right) \hat{\nabla} f(\mathbf{0}) \right) \|^2 = \sum_{j \in \mathcal{S}} |(a_t^{(j)} - b_t^{(j)}) \hat{\nabla} f(\mathbf{0})^{(j)}|^2$$

$$\leq (t+1)^2 (1-\theta)^t \|\hat{\nabla} f(\mathbf{0})\|^2 \leq (t+1)^2 (1-\theta)^t 2(\|\nabla f(\mathbf{0})\|^2 + \frac{\rho^2 d\mu^4}{36}) \leq \epsilon^2/400$$

$$\|\mathcal{P}_{\mathcal{S}} \left( \nabla^2 f(\mathbf{0}) \left(\mathbf{I} \quad \mathbf{0}\right) \mathbf{A}^t \begin{pmatrix} \mathbf{0} \\ -\mathbf{v}_0 \end{pmatrix} \right) \|^2 \leq \sum_{j \in \mathcal{S}} |\lambda_j b_t^{(j)} \mathbf{v}_0^{(j)}|^2 \ell^2 (t+1)^2 (1-\theta)^t \|\mathbf{v}_0\|^2 \leq \epsilon^2/400.$$

Using Lemma 13, for all $j \in \mathcal{S}$, we have

$$\left| \left( \eta \nabla^2 f(\mathbf{0}) \left(\mathbf{I} \quad \mathbf{0}\right) \sum_{\tau=0}^{t-1} \mathbf{A}^{t-1-\tau} \begin{pmatrix} \delta_t \\ \mathbf{0} \end{pmatrix} \right)^{(j)} \right| = |\eta\lambda_j \sum_{\tau=0}^{t-1} a_\tau^{(j)} \delta_{t-1-\tau}| \leq |\delta_{t-1}^{(j)}| + \sum_{\tau=1}^{t-1} |\delta_\tau^{(j)} - \delta_{\tau-1}^{(j)}|$$

Using Lemma 21 and choose $\mu \leq \tilde{O}(\frac{\epsilon^{5/8}}{d^{1/4}})$, we have

$$\|\mathcal{P}_{\mathcal{S}} \left( \eta \nabla^2 f(\mathbf{0}) \left(\mathbf{I} \quad \mathbf{0}\right) \sum_{\tau=0}^{t-1} \mathbf{A}^{t-1-\tau} \begin{pmatrix} \delta_t \\ \mathbf{0} \end{pmatrix} \right) \| \leq 2\|\delta_{t-1}\|^2 + 2t \sum_{\tau=1}^{t-1} \|\delta_\tau - \delta_{\tau-1}\|^2 \leq \mathcal{O}(\rho^2 \mathscr{S}^4) \leq \mathcal{O}(\epsilon^2 c^{-12}) \leq \frac{\epsilon^2}{400}$$

Thus we have for any $t \in [\mathscr{T}/4, \mathscr{T}]$,

$$\|\mathcal{P}_{\mathcal{S}} \nabla f(\mathbf{x}_t)\| \leq \frac{\epsilon}{4}.$$

Using Lemma 20, we have

$$\mathbf{v}_t = \begin{pmatrix} 1 & -1 \end{pmatrix} \begin{pmatrix} \mathbf{x}_t \\ \mathbf{x}_{t-1} \end{pmatrix} = \begin{pmatrix} 1 & -1 \end{pmatrix} \mathbf{A}^t \begin{pmatrix} 0 \\ -\mathbf{v}_0 \end{pmatrix} - \eta \begin{pmatrix} 1 & -1 \end{pmatrix} \sum_{\tau=0}^{t-1} \mathbf{A}^{t-1-\tau} \begin{pmatrix} \hat{\nabla} f(\mathbf{0}) \\ \mathbf{0} \end{pmatrix}$$

$$- \eta \begin{pmatrix} 1 & -1 \end{pmatrix} \sum_{\tau=0}^{t-1} \mathbf{A}^{t-1-\tau} \begin{pmatrix} \delta_\tau \\ \mathbf{0} \end{pmatrix}$$

By Lemma 12, for $t \geq \mathscr{T}/4 = \Omega(\frac{c}{\theta} \log \frac{1}{\theta})$, we have

$$\|[\mathcal{P}_{\mathcal{S}}^{\mathsf{T}} \nabla^2 f(\mathbf{x}_0) \mathcal{P}_{\mathcal{S}}]^{1/2} \begin{pmatrix} 1 & -1 \end{pmatrix} \mathbf{A}^t \begin{pmatrix} 0 \\ -\mathbf{v}_0 \end{pmatrix} \|^2 = \sum_{j \in \mathcal{S}} |\lambda_j^{1/2} (b_t^{(j)} - b_{t-1}^{(j)}) \mathbf{v}_0^{(j)}|^2 \leq \ell (t+1)^2 (1-\theta)^t \|\mathbf{v}_0\|^2$$

$$\leq \mathcal{O}(\frac{\epsilon^2}{\ell} c^{-3}) \leq \frac{1}{3} \sqrt{\rho \epsilon} \mathscr{M}^2$$

By Lemma 10, we have

$$|\eta \lambda_j \begin{pmatrix} 1 & -1 \end{pmatrix} \sum_{\tau=0}^{t-1} \mathbf{A}_j^{t-1-\tau} \begin{pmatrix} 1 \\ 0 \end{pmatrix}| = |\eta \lambda_j \begin{pmatrix} 1 & 0 \end{pmatrix} \sum_{\tau=0}^{t-1} (\mathbf{A}_j^{t-1-\tau} - \mathbf{A}_j^{t-2-\tau}) \begin{pmatrix} 1 \\ 0 \end{pmatrix}|$$

$$= |\begin{pmatrix} 1 & 0 \end{pmatrix} (\mathbf{A}_j^t - \mathbf{A}_j^{t-1}) \begin{pmatrix} 1 \\ 0 \end{pmatrix}|.$$

By choosing $\mu \leq \tilde{\mathcal{O}}(\frac{\epsilon^{1/2}}{d^{1/4}})$, then we have

$$\|[\mathcal{P}_{\mathcal{S}}^{\mathsf{T}} \nabla^2 f(\mathbf{x}_0) \mathcal{P}_{\mathcal{S}}]^{1/2} \eta \begin{pmatrix} 1 & -1 \end{pmatrix} \sum_{\tau=0}^{t-1} \mathbf{A}^{t-1-\tau} \begin{pmatrix} \hat{\nabla} f(\mathbf{0}) \\ \mathbf{0} \end{pmatrix} \|^2$$

$$= \sum_{j \in \mathcal{S}} |\lambda_j^{-1/2} (a_t^{(j)} - a_{t-1}^{(j)} - b_t^{(j)} + b_{t-1}^{(j)}) \hat{\nabla} f(\mathbf{0})^{(j)}|^2$$

$$\leq \mathcal{O}(\frac{1}{\sqrt{\rho \epsilon}})(t+1)^2 (1-\theta)^t \|\hat{\nabla} f(\mathbf{0})\|^2 \leq \mathcal{O}(\frac{1}{\sqrt{\rho \epsilon}})(t+1)^2 (1-\theta)^t \cdot 2(\|\nabla f(\mathbf{0})\|^2 + \frac{\rho^2 d \mu^4}{36})$$

$$\leq \mathcal{O}(\frac{\epsilon^3}{\ell} c^{-3}) \leq \frac{1}{3} \sqrt{\rho \epsilon} \mathscr{M}^2.$$

By Lemma 13, for any $j \in \mathcal{S}$, we have

$$|(\nabla^2 f(\mathbf{0})^{\frac{1}{2}} \eta \begin{pmatrix} 1 & -1 \end{pmatrix} \sum_{\tau=0}^{t-1} \mathbf{A}^{t-1-\tau} \begin{pmatrix} \delta_\tau \\ \mathbf{0} \end{pmatrix})^{(j)}|$$

$$= |\eta \lambda_j^{1/2} \sum_{\tau=0}^{t-1} (a_\tau - a_{\tau-1}) \delta_{t-1-\tau}| \leq \sqrt{\eta}(|\delta_{t-1}^{(j)}| + \sum_{\tau=1}^{t-1} |\delta_\tau^{(j)} - \delta_{\tau-1}^{(j)}|).$$

Using Lemma 21 and choose $\mu \leq \tilde{\mathcal{O}}(\frac{\epsilon^{5/8}}{d^{1/4}})$, we have

$$\|[\mathcal{P}_{\mathcal{S}}^{\mathsf{T}} \nabla^2 f(\mathbf{x}_0) \mathcal{P}_{\mathcal{S}}]^{1/2} \eta \begin{pmatrix} 1 & -1 \end{pmatrix} \sum_{\tau=0}^{t-1} \mathbf{A}^{t-1-\tau} \begin{pmatrix} \delta_\tau \\ \mathbf{0} \end{pmatrix} \|^2 \leq \eta [2\|\delta_{t-1}\|^2 + 2t \sum_{\tau=1}^{t-1} \|\delta_\tau - \delta_{\tau-1}\|^2]$$

$$\leq \mathcal{O}(\eta \rho^2 \mathscr{S}^2) \leq \mathcal{O}(\frac{\epsilon^2}{\ell} c^{-6}) \leq \frac{1}{3} \sqrt{\rho \epsilon} \mathscr{M}^2$$

Thus we have

$$\mathbf{v}_t^{\mathsf{T}} [\mathcal{P}_{\mathcal{S}}^{\mathsf{T}} \nabla^2 f(\mathbf{x}_0) \mathcal{P}_{\mathcal{S}}] \mathbf{v}_t \leq \sqrt{\rho \epsilon} \mathscr{M}^2.$$

$\square$

**Lemma 5.** *If $\|\hat{\nabla} f(\mathbf{x}_\tau)\| \geq \frac{3\epsilon}{4}$ with $\mu \leq \mathcal{O}((\frac{3\epsilon}{2\rho\sqrt{d}})^{1/2})$ in Line 3 of Algorithm 1 for all $\tau \in [0, \mathscr{T}]$, then by running Algorithm 1 with $\mu \leq \tilde{\mathcal{O}}(\frac{\epsilon^{5/8}}{d^{1/4}})$ in Line 6 and $\mu \leq \tilde{\mathcal{O}}(\frac{\epsilon^{1/2}}{d^{1/4}})$ in Line 8, we have $E_{\mathscr{T}} - E_0 \leq -\mathscr{E}$.*

*Proof.* According to lemma 1, if we choose $\mu \leq \mathcal{O}((\frac{3\epsilon}{2\rho\sqrt{d}})^{1/2})$ in Line 3 of Algorithm 1. Then we get if $\|\hat{\nabla}f(\mathbf{x}_t)\| \leq \frac{3\epsilon}{4}$, then $\|\nabla f(\mathbf{x}_t)\| \leq \epsilon$, otherwise $\|\nabla f(\mathbf{x}_t)\| \geq \frac{\epsilon}{2}$. According to Algorithm 1, if $\|\hat{\nabla}f(\mathbf{x}_\tau)\| \geq \frac{\epsilon}{4}$, then for all $\tau \in [0, \mathcal{T}]$, the perturbation step is not reached.

According to Lemma 19, as long as the NCE step is reached, then we have the Hamiltonian we decrease by $\mathcal{E}$ in a single step. And according to Lemma 3 and Lemma 4, the Hamiltonian decrease monotonically in all steps, so we have Lemma 5 holds.

Then we prove that if the NCE step is never reached in all steps $\tau \in [0, \mathcal{T}]$, Lemma 5 holds. Let $t_1 = \arg\min_{t \in [0, \mathcal{T}]}\{t | \|\mathbf{v}_t \leq \mathcal{M} \text{ and} \|\nabla f(\mathbf{x}_t)\| \leq 2\ell\mathcal{M}\|\}$. When $t_1 \in [\mathcal{T}/4, \mathcal{T}]$, then we have $E_\mathcal{T} - E_0 \leq E_{\mathcal{T}/4} - E_0 \leq -\mathcal{E}$ according to Lemma 22. Then we discuss the case when $t_1 \in [0, \mathcal{T}/4]$. Using Lemma 24 by setting $t_1$ as a initial step, we have

$$\|\mathcal{P}_\mathcal{S}\nabla f(\mathbf{x}_t)\| \leq \frac{\epsilon}{4} \quad and \quad \mathbf{v}_t^\mathsf{T}[\mathcal{P}_\mathcal{S}^\mathsf{T}\nabla^2 f(\mathbf{x}_0)\mathcal{P}_\mathcal{S}]\mathbf{v}_t \leq \sqrt{\rho\epsilon}\mathcal{M}^2. \qquad \forall t \in [t_1 + \mathcal{T}/4, t_1 + \mathcal{T}/2].$$

Let $t_2 = \arg\min_{t \in [t_1 + \mathcal{T}/4, \mathcal{T}]}\{t | \|\mathbf{v}_t\| \leq \mathcal{M}\}$. If $t_2 \geq t_1 + \frac{\mathcal{T}}{2}$, then Hamiltonian will decrease by $\mathcal{E}$ by Lemma 22. Otherwise, $t_2 \in [t_1 + \mathcal{T}/4, t_1 + \mathcal{T}/2]$, we have $\|\mathcal{P}_\mathcal{S}\nabla f(\mathbf{x}_{t_2})\| \leq \frac{\epsilon}{4}$, by prediction of Lemma 5, we have $\|\nabla f(\mathbf{x}_{t_2})\| \geq \frac{3\epsilon}{4}$, so we have $\|\mathcal{P}_{\mathcal{S}^c}\nabla f(\mathbf{x}_{t_2})\| \geq \frac{\epsilon}{4}$. By Lemma 18, $\|\mathbf{x}_{t_1} - \mathbf{x}_{t_2}\| \leq 2\mathcal{S}$ holds, then we have

$$\mathbf{v}_{t_2}^\mathsf{T}[\mathcal{P}_{\mathcal{S}^\mathsf{T}}\nabla^2 f(\mathbf{x}_{t_2})\mathcal{P}_\mathcal{S}]\mathbf{v}_{t_2} \leq \mathbf{v}_{t_2}^\mathsf{T}[\mathcal{P}_{\mathcal{S}^\mathsf{T}}\nabla^2 f(\mathbf{x}_{t_1})\mathcal{P}_\mathcal{S}]\mathbf{v}_{t_2} + \|\nabla^2 f(\mathbf{x}_{t_1}) - \nabla^2 f(\mathbf{x}_{t_2})\|\|\mathbf{v}_{t_2}\|^2 \leq 2\sqrt{\rho\epsilon}\mathcal{M}^2$$

So according to Lemma 23, the Hamilton will decrease by $\mathcal{E}$. $\qquad\square$

## C.2 NEGATIVE CURVATURE

**Lemma 25.** *Suppose* $\|\hat{\nabla}f(\tilde{\mathbf{x}})\| \leq \frac{3\epsilon}{4}$ *( thus* $\|\hat{\nabla}f(\tilde{\mathbf{x}})\| \leq \epsilon$*) and* $\lambda_{\min}(\nabla^2 f(\tilde{\mathbf{x}})) \leq -\sqrt{\rho\epsilon}$*.* $\mathbf{x}_0$ *and* $\mathbf{x}_0'$ *are at distance at most* $r$ *from* $\tilde{\mathbf{x}}$*. Let* $\mathbf{x}_0 - \mathbf{x}_0' = r_0\mathbf{e}_1$ *and* $\mathbf{v}_0 = \mathbf{v}_0' = \tilde{v}$ *where* $\mathbf{e}_1$ *is the minimum eigen-direction of* $\nabla^2 f(\tilde{\mathbf{x}})$ *and* $r_0 \geq \frac{\delta\mathcal{E}r}{2\Delta_f\sqrt{d}}$*. Then, running zeroth-order AGD starting at* $(\mathbf{x}_0, \mathbf{v}_0)$ *and* $(\mathbf{x}_0', \mathbf{v}_0')$ *respectively and set* $\mu \leq \tilde{\mathcal{O}}(\frac{\epsilon^{13/8}}{d^{1/2}})$*, we have*

$$\min\{E_\mathcal{T} - \tilde{E}, E_\mathcal{T}' - \tilde{E}\} \leq -\mathcal{E}.$$

*Proof.* Assume that

$$\min\{E_\mathcal{T} - E_0, E_\mathcal{T}' - E_0'\} \geq -2\mathcal{E},$$

where $E_0$ and $E_0'$ are Hamiltonians at $(\mathbf{x}_0, \mathbf{v}_0)$ and $(\mathbf{x}_0', \mathbf{v}_0')$, respectively. By Lemma 18 and choose $\mu \leq \tilde{\mathcal{O}}(\frac{\epsilon^{3/8}}{d^{1/4}})$, we have for any $t \leq \mathcal{T}$,

$$\max\{\|\mathbf{x}_t - \tilde{x}\|, \|\mathbf{x}_t' - \tilde{x}\|\} \leq \max\{\|\mathbf{x}_t - \mathbf{x}_0 + \mathbf{x}_0 - \tilde{x}\|, \|\mathbf{x}_t' - \mathbf{x}_0' + \mathbf{x}_0' - \tilde{x}\|\}$$

$$\leq r + \max\{\|\mathbf{x}_t - \mathbf{x}_0\|, \|\mathbf{x}_t' - \mathbf{x}_0'\|\} \leq r + \sqrt{\frac{4\eta\mathcal{E}\mathcal{T}}{\theta} + \mathcal{T}^2\frac{\eta\rho^2 d\mu^4}{48}} \leq 2\mathcal{S}.$$

Let $\tilde{x} = \mathbf{0}$ be the origin. Let $\mathbf{w}_t = \mathbf{x}_t - \mathbf{x}_t'$, according to lemma 20, we have

$$\begin{pmatrix} \mathbf{w}_{t+1} \\ \mathbf{w}_t \end{pmatrix} = \mathbf{A}^t \begin{pmatrix} \mathbf{w}_1 \\ \mathbf{w}_0 \end{pmatrix} - \eta \sum_{\tau=1}^t \mathbf{A}^{t-\tau} \begin{pmatrix} \xi_\tau \\ 0 \end{pmatrix} = \mathbf{A}^{t+1} \begin{pmatrix} \mathbf{w}_0 \\ \mathbf{w}_{-1} \end{pmatrix} - \eta \sum_{\tau=0}^t \mathbf{A}^{t-\tau} \begin{pmatrix} \xi_\tau \\ 0 \end{pmatrix},$$

where $\xi_t = \hat{\nabla}f(\mathbf{y}_t) - \hat{\nabla}f(\mathbf{y}_t') - \nabla^2 f(\mathbf{0})(\mathbf{y}_t - \mathbf{y}_t') =:$. Let $\Delta_t = \int_0^1 (\nabla^2 f(\phi\mathbf{y}_t + (1-\phi)\mathbf{y}_t') - \nabla^2 f(\mathbf{0}))d\phi$, then we have

$$\xi_t = \Delta_t(\mathbf{y}_t - \mathbf{y}_t') + e_t - e_t' = \Delta_t((1-\theta)\mathbf{w}_t - (1-\theta)\mathbf{w}_{t-1}) + e_t - e_t',$$

where $e_t = \hat{\nabla}f(\mathbf{y}_t) - \nabla f(\mathbf{y}_t), e_t' = \hat{\nabla}f(\mathbf{y}_t') - \nabla f(\mathbf{y}_t')$. Since $\mathbf{v}_0 = \mathbf{v}_0'$, we have $\mathbf{w}_{-1} = \mathbf{w}_0$, $\|\Delta_t\| \leq \rho\max\{\|\mathbf{x}_t - \tilde{x}\|, \|\mathbf{x}_t' - \tilde{x}\|\} \leq 2\rho\mathcal{S}$ and $\|\xi_t\| \leq 6\rho\mathcal{S}(\|\mathbf{w}_\tau\| + \|\mathbf{w}_{\tau-1}\|) + \frac{\rho\sqrt{d}\mu^2}{3}$. Then we prove by induction that

$$\|\eta\,(\mathbf{I} \quad 0)\sum_{\tau=0}^{t-1} \mathbf{A}^{t-1-\tau} \begin{pmatrix} \xi_\tau \\ 0 \end{pmatrix}\| \leq \frac{1}{2}\|(\mathbf{I} \quad 0)\mathbf{A}^t \begin{pmatrix} \mathbf{w}_0 \\ \mathbf{w}_0 \end{pmatrix}\|$$

For reasonably small $\mu$, it is easy to check the base case holds for $t = 1$ as $\|\mathbf{A}\| \leq \ell = 4\eta$. Then we assume that for all steps less than or equal to $t$, the induction assumption holds. Then we have

$$\|\mathbf{w}_t\| = \|(\mathbf{I} \quad 0)\,\mathbf{A}^t \begin{pmatrix} \mathbf{w}_0 \\ \mathbf{w}_0 \end{pmatrix} - \eta\,(\mathbf{I} \quad 0)\sum_{\tau=0}^{t-1}\mathbf{A}^{t-1-\tau}\begin{pmatrix} \xi_\tau \\ 0 \end{pmatrix}\| \leq \|(\mathbf{I} \quad 0)\,\mathbf{A}^t \begin{pmatrix} \mathbf{w}_0 \\ \mathbf{w}_0 \end{pmatrix}\| + \|\eta\,(\mathbf{I} \quad 0)\sum_{\tau=0}^{t-1}\mathbf{A}^{t-1-\tau}\begin{pmatrix} \xi_\tau \\ 0 \end{pmatrix}\|$$

$$\leq 2\|(\mathbf{I} \quad 0)\,\mathbf{A}^t \begin{pmatrix} \mathbf{w}_0 \\ \mathbf{w}_0 \end{pmatrix}\|,$$

then we have

$$\|\xi_t\| \leq \mathcal{O}(\rho\mathscr{S})(\|\mathbf{w}_t\| + \|\mathbf{w}_{t-1}\|) + \frac{\rho\sqrt{d}\mu^2}{3} \leq \mathcal{O}(\rho\mathscr{S})(\|(\mathbf{I} \quad 0)\,\mathbf{A}^t \begin{pmatrix} \mathbf{w}_0 \\ \mathbf{w}_0 \end{pmatrix}\| + \|(\mathbf{I} \quad 0)\,\mathbf{A}^{t-1} \begin{pmatrix} \mathbf{w}_0 \\ \mathbf{w}_0 \end{pmatrix}\|) + \frac{\rho\sqrt{d}\mu^2}{3}$$

$$\leq \mathcal{O}(\rho\mathscr{S})\|(\mathbf{I} \quad 0)\,\mathbf{A}^t \begin{pmatrix} \mathbf{w}_0 \\ \mathbf{w}_0 \end{pmatrix}\| + \frac{\rho\sqrt{d}\mu^2}{3},$$

where the last inequality uses Lemma 16. For the case $t + 1$, we have

$$\|\eta\,(\mathbf{I} \quad 0)\sum_{\tau=0}^{t}\mathbf{A}^{t-\tau}\begin{pmatrix} \xi_\tau \\ 0 \end{pmatrix}\| \leq \eta\sum_{\tau=0}^{t}\|(\mathbf{I} \quad 0)\,\mathbf{A}^{t-\tau}\begin{pmatrix} \mathbf{I} \\ 0 \end{pmatrix}\|\|\xi_\tau\|$$

$$\leq \eta\sum_{\tau=0}^{t}\|(\mathbf{I} \quad 0)\,\mathbf{A}^{t-\tau}\begin{pmatrix} \mathbf{I} \\ 0 \end{pmatrix}\|\left(\mathcal{O}(\rho\mathscr{S})\|(\mathbf{I} \quad 0)\,\mathbf{A}^{\tau}\begin{pmatrix} \mathbf{w}_0 \\ \mathbf{w}_0 \end{pmatrix}\| + \frac{\rho\sqrt{d}\mu^2}{3}\right)$$

Without loss of generality, assume that the minimum eigenvector direction of $\nabla^2 f(\tilde{\mathbf{x}})$ is along the first coordinate $\mathbf{e}_1$ with the corresponding $2 \times 2$ matrix $\mathbf{A}_1$. Let $\begin{pmatrix} a_t^{(1)} & -b_t^{(1)} \end{pmatrix} = (1 \quad 0)\,\mathbf{A}_1$. If we choose $\mu \leq \tilde{\mathcal{O}}(\frac{\epsilon^{13/8}}{d^{1/2}})$, then

$$\|\eta\,(\mathbf{I} \quad 0)\sum_{\tau=0}^{t}\mathbf{A}^{t-\tau}\begin{pmatrix} \xi_\tau \\ 0 \end{pmatrix}\| \leq \eta\sum_{\tau=0}^{t}a_{t-\tau}^{(1)}\left(\mathcal{O}(\rho\mathscr{S})(a_\tau^{(1)} - b_\tau^{(1)})\|\mathbf{w}_0\| + \frac{\rho\sqrt{d}\mu^2}{3}\right)$$

$$\leq \eta\sum_{\tau=0}^{t}a_{t-\tau}^{(1)}\left(\mathcal{O}(\rho\mathscr{S})(a_\tau^{(1)} - b_\tau^{(1)})\|\mathbf{w}_0\|\right)$$

$$\leq \mathcal{O}(\eta\rho\mathscr{S})\sum_{\tau=0}^{t}(\frac{2}{\theta} + t + 1)|a_{t+1}^{(1)} - b_{t+1}^{(1)}|\|\mathbf{x}_0\|$$

$$\leq \mathcal{O}(\eta\rho\mathscr{S})\|(\mathbf{I} \quad 0)\,\mathbf{A}^{t+1}\begin{pmatrix} \mathbf{w}_0 \\ \mathbf{w}_0 \end{pmatrix}\| \leq \frac{1}{2}\|(\mathbf{I} \quad 0)\,\mathbf{A}^{t+1}\begin{pmatrix} \mathbf{w}_0 \\ \mathbf{w}_0 \end{pmatrix}\|,$$

where the second inequality used Lemma 16 that $|a_\tau^{(1)} - b_\tau^{(1)}| \geq \frac{\theta}{2}$ and $\mu \leq \tilde{\mathcal{O}}(\frac{\epsilon^{13/8}}{d^{1/2}})$; the third inequality used Lemma 14 and the fourth inequality used $\frac{1}{\theta} \leq \mathscr{S}$. Then we finished the proof of the induction. Then we have

$$\|\mathbf{w}_t\| \geq \|(\mathbf{I} \quad 0)\,\mathbf{A}^t \begin{pmatrix} \mathbf{w}_0 \\ \mathbf{w}_0 \end{pmatrix}\| - \|\eta\,(\mathbf{I} \quad 0)\sum_{\tau=0}^{t-1}\mathbf{A}^{t-1-\tau}\begin{pmatrix} \xi_\tau \\ 0 \end{pmatrix}\| \geq \frac{1}{2}\|(\mathbf{I} \quad 0)\,\mathbf{A}^t \begin{pmatrix} \mathbf{w}_0 \\ \mathbf{w}_0 \end{pmatrix}\| \geq \frac{1}{4}(1 + \Omega(\theta))^t r_0,$$

where the last inequality Lemma 16 and $\lambda_{\min}(\nabla^2 f(\tilde{x}))$. Since $r_0 \geq \frac{\delta\mathscr{E}r}{2\Delta_f\sqrt{d}}$, $\mathscr{T} = \Omega(\frac{1}{\theta}\chi c)$ Then we have

$$\|\mathbf{w}_{\mathscr{T}}\| = \|\mathbf{x}_{\mathscr{T}} - \mathbf{x}'_{\mathscr{T}}\| \geq \frac{1}{4}(1 + \omega(\theta))^{\mathscr{T}} r_0 \geq 4\mathscr{S},$$

which is contradicted with $\forall t \leq \mathscr{T}, \max\{\|\mathbf{x}_t - \tilde{x}\|, \|\mathbf{x}'_t - \tilde{x}\|\} \leq 2\mathscr{S}$. Therefore the following inqualty holds

$$\min\{E_{\mathscr{T}} - E_0, E'_{\mathscr{T}} - E'_0\} \leq -2\mathscr{E}.$$

Since $\max\{E_0 - \tilde{E}, \mathbb{E}'_0 - \tilde{E}\} = \max\{f(\mathbf{x}_0) - f(\tilde{\mathbf{x}}), f(\mathbf{x}'_0) - f(\tilde{\mathbf{x}})\} \leq \epsilon r + \frac{\ell r^2}{2} \leq \mathscr{E}$. Then we have

$$\min\{E_{\mathscr{T}} - \tilde{E}, E'_{\mathscr{T}} - \tilde{E}\} \leq -\mathscr{E}.$$

$\square$

**Lemma 6.** *Suppose* $\|\hat{\nabla}f(\mathbf{x}_t)\| \leq \frac{3\epsilon}{4}$ *( thus* $\|\nabla f(\mathbf{x}_t)\| \leq \epsilon$*),* $\lambda_{\min}(\nabla^2 f(\mathbf{x}_t)) \leq -\sqrt{\rho\epsilon}$ *and no perturbation is added in iterations* $[t - \mathscr{T}, t]$*. Then by running Algorithm 1, we have* $E_{\mathscr{T}} - E_0 \leq -\mathscr{E}$ *with probability at least* $1 - \frac{\delta\mathscr{E}}{2\Delta_f}$*.*

*Proof.* According to precondition of Lemma 6, a perturbation will be added at iteration 0, then the Hamiltonian will increase by at most $\mathscr{E}$. According to Lemma 19, Hamiltonian will decrease by $2\mathscr{E}$ if at least one NCE step is called and thus $E_{\mathscr{T}} - E_0 \leq -\mathscr{E}$. Otherwise NCE step is never reached in iterations $[0, \mathscr{T}]$. In this case, denote by $\mathbb{B}_{\mathbf{x}_0}(r)$ the ball with radius $r$ around $\mathbf{x}_0$. Let $\mathcal{X} \subset \mathbb{B}_{\mathbf{x}_0}(r)$ be the region where Hamiltonian will not decrease by $\mathscr{E}$ if the AGD sequences started from at a point $\mathbf{x} \in \mathcal{X}$. Then by Lemma 25, the width of region is no more than $r_0 = \frac{\delta\mathscr{E}r}{2\Delta_f\sqrt{d}}$, then we have

$$\frac{\mathrm{Vol}(\mathcal{X})}{\mathrm{Vol}(\mathbb{B}_{\mathbf{x}_0}(r))} \leq \frac{r_0 \times \mathrm{Vol}(\mathbb{B}_{\mathbf{x}_0}^{(d-1)}(r))}{\mathrm{Vol}(\mathbb{B}_{\mathbf{x}_0}^{(d)}(r))} = \frac{r_0\Gamma(d/2+1)}{r\sqrt{\pi}\Gamma(d/2+1/2)} \leq \frac{\delta\mathscr{E}}{2\Delta_f}.$$

Then, with probability at least $1 - \frac{\delta\mathscr{E}}{2\Delta_f}$, we have $E_{\mathscr{T}} - E_0 \leq -\mathscr{E}$.

$\square$

## C.3 PROOF OF THEOREM 1

*Proof.* Consider the set $\mathcal{H} = \{\tau | \tau \in [0, \mathscr{T}] \text{and} \|\hat{\nabla}f(\mathbf{x}_\tau)\| \leq \frac{3\epsilon}{4}\}$ and suppose that all $\mathbf{x}_\tau$ are not $\epsilon$-approximate SOSPs. If $\mathcal{H} = \emptyset$, then no perturbation is added and by Lemma 5, we have $E_{\mathscr{T}} - E_0 \leq -\mathscr{E}$. Else if $\mathcal{H} \neq \emptyset$, then define $\tau' = \arg\min \mathcal{H}$. Then by Lemma 6, we have $E_{\tau'+\mathscr{T}} - E_0 \leq E_{\tau'+\mathscr{T}} - E_{\tau'} \leq -\mathscr{E}$. Thus the Hamiltonian will decrease by at least $\mathscr{E}/(2\mathscr{T})$ per step and the total steps is no more than $\frac{2\mathscr{T}\Delta_f}{\mathscr{E}}$. In all $\frac{2\mathscr{T}\Delta_f}{\mathscr{E}}$ steps, Lemma 6 is called at most $\frac{2\Delta_f}{\mathscr{E}}$ times. Denote by $A$ the event that the argument of Theorem 1 is true and denote by $A_i, i \in \{1, \ldots, \lfloor\frac{2\mathscr{T}\Delta_f}{\mathscr{E}}\rfloor\}$ the event that the argument of Lemma 6 is true. Then by union bound, we have $\Pr(A) \geq \Pr(\bigcap_i A_i) = 1 - \Pr(\bigcup_i \bar{A}_i) \geq 1 - \sum_i \Pr(\bar{A}_i) \geq 1 - \frac{2\Delta_f}{\mathscr{E}} \cdot \frac{\delta\mathscr{E}}{2\Delta_f} = 1 - \delta$.

$\square$

# D PROOF OF MAIN RESULTS OF ALGORITHM 3

---

**Algorithm 4** Zeroth-Order Accelerated Negative Curvature Finding without Renormalization$(\tilde{\mathbf{x}}, r', \mathscr{T}')$

---

1: $\mathbf{x}_0 \leftarrow \mathrm{Unif}(\mathbb{B}_{\tilde{\mathbf{x}}}(r'))$
2: $\mathbf{y}_0 \leftarrow \mathbf{x}_0$
3: **for** $t = 0, \ldots, \mathscr{T}'$ **do**
4:      $\mathbf{x}_{t+1} = \mathbf{y}_t - \eta \frac{\|\mathbf{y}_t - \tilde{\mathbf{x}}\|}{r'}\left(\hat{\nabla}f(r'\frac{\mathbf{y}_t - \tilde{\mathbf{x}}}{\|\mathbf{y}_t - \tilde{\mathbf{x}}\|} + \tilde{\mathbf{x}}) - \hat{\nabla}f(\tilde{\mathbf{x}})\right)$
5:      $\mathbf{v}_{t+1} = \mathbf{x}_{t+1} - \mathbf{x}_t$
6:      $\mathbf{y}_{t+1} = \mathbf{x}_{t+1} + (1-\theta)\mathbf{v}_{t+1}$
    **return** $\frac{\mathbf{x}_{\mathscr{T}'} - \tilde{\mathbf{x}}}{\|\mathbf{x}_{\mathscr{T}'} - \tilde{\mathbf{x}}\|}$.

---

**Lemma 26.** *The output of the algorithm 4 is the same as the unit* $\hat{\mathbf{e}}$ *in Algorithm 3. Denote the sequence of* $\{\mathbf{x}_r\}$ *obtained by Algorithm 4 and Algorithm 3 by* $\{\mathbf{x}_{1,0}, \mathbf{x}_{1,1}, \ldots, \mathbf{x}_{1,\mathscr{T}'}\}$ *and* $\{\mathbf{x}_{2,0}, \mathbf{x}_{2,1}, \ldots, \mathbf{x}_{2,\mathscr{T}'}\}$*, respectively. Then we have*

$$\frac{\mathbf{x}_{1,\mathscr{T}'} - \tilde{\mathbf{x}}}{\|\mathbf{x}_{1,\mathscr{T}'} - \tilde{\mathbf{x}}\|} = \frac{\mathbf{x}_{2,\mathscr{T}'} - \tilde{\mathbf{x}}}{\|\mathbf{x}_{2,\mathscr{T}'} - \tilde{\mathbf{x}}\|}.$$

*Proof.* We prove this by induction that

$$\frac{\mathbf{x}_{2,k} - \tilde{\mathbf{x}}}{\|\mathbf{y}_{2,k} - \tilde{\mathbf{x}}\|} = \frac{\mathbf{x}_{1,k} - \tilde{\mathbf{x}}}{r'}, \qquad \frac{\mathbf{y}_{2,k} - \tilde{\mathbf{x}}}{\|\mathbf{y}_{2,k} - \tilde{\mathbf{x}}\|} = \frac{\mathbf{y}_{1,k} - \tilde{\mathbf{x}}}{r'}.$$

It is easy to check that the base case holds for $k = 0$. Then we assume that the above equations holds for all $k \leq t$. Then we have

$$\mathbf{x}_{2,t+1} - \tilde{\mathbf{x}} = \mathbf{y}_{2,t} - \tilde{\mathbf{x}} - \eta \frac{\|\mathbf{y}_{2,t} - \tilde{\mathbf{x}}\|}{r'} \left( \hat{\nabla} f(r' \frac{\mathbf{y}_{2,t} - \tilde{\mathbf{x}}}{\|\mathbf{y}_{2,t} - \tilde{\mathbf{x}}\|} + \tilde{\mathbf{x}}) - \hat{\nabla} f(\tilde{\mathbf{x}}) \right)$$

$$= \mathbf{y}_{2,t} - \tilde{\mathbf{x}} - \eta \frac{\|\mathbf{y}_{2,t} - \tilde{\mathbf{x}}\|}{r'} \left( \hat{\nabla} f(\mathbf{y}_{1,t}) - \hat{\nabla} f(\tilde{\mathbf{x}}) \right) = \frac{\|\mathbf{y}_{2,t} - \tilde{\mathbf{x}}\|}{r'} \left( \mathbf{y}_{1,t} - \tilde{\mathbf{x}} - \eta \left( \hat{\nabla} f(\mathbf{y}_{1,t}) - \hat{\nabla} f(\tilde{\mathbf{x}}) \right) \right)$$

Denote by $\mathbf{x}'_{1,t+1}, \mathbf{y}'_{1,t+1}$ the value of $\mathbf{x}_{1,t+1}, \mathbf{y}_{1,t+1}$ before renormalization, then

$$\mathbf{x}'_{1,t+1} = \mathbf{y}_{1,t} - \eta(\hat{\nabla} f(\mathbf{y}_{1,t}) - \hat{\nabla} f(\tilde{\mathbf{x}})), \quad \mathbf{y}'_{1,t+1} = \mathbf{x}'_{1,t+1} + (1 - \theta)(\mathbf{x}'_{1,t+1} - \mathbf{x}_{1,t})$$

Then we have

$$\mathbf{x}_{2,t+1} - \tilde{\mathbf{x}} = \frac{\|\mathbf{y}_{2,t} - \tilde{\mathbf{x}}\|}{r'} \left( \mathbf{y}_{1,t} - \tilde{\mathbf{x}} - \eta \left( \hat{\nabla} f(\mathbf{y}_{1,t}) - \hat{\nabla} f(\tilde{\mathbf{x}}) \right) \right) = \frac{\|\mathbf{y}_{2,t} - \tilde{\mathbf{x}}\|}{r'} \left( \mathbf{x}'_{1,t+1} - \tilde{\mathbf{x}} \right),$$

and

$$\mathbf{y}_{2,t+1} - \tilde{\mathbf{x}} = \mathbf{x}_{2,t+1} - \tilde{\mathbf{x}} + (1 - \theta)(\mathbf{x}_{2,t+1} - \tilde{\mathbf{x}} - (\mathbf{x}_{2,t} - \tilde{\mathbf{x}})) = \frac{\|\mathbf{y}_{2,t} - \tilde{\mathbf{x}}\|}{r'} (\mathbf{y}'_{1,t+1} - \tilde{\mathbf{x}}),$$

then we have

$$\|\mathbf{y}_{2,t+1} - \tilde{\mathbf{x}}\| = \frac{\|\mathbf{y}_{2,t} - \tilde{\mathbf{x}}\|}{r'} \|\mathbf{y}'_{1,t+1} - \tilde{\mathbf{x}}\|.$$

So we have

$$\mathbf{x}_{2,t+1} - \tilde{\mathbf{x}} = \frac{\|\mathbf{y}_{2,t} - \tilde{\mathbf{x}}\|}{r'} (\mathbf{x}'_{1,t+1} - \tilde{\mathbf{x}}) = \frac{\|\mathbf{y}_{2,t} - \tilde{\mathbf{x}}\|}{r'} \frac{\|\mathbf{y}'_{1,t+1} - \tilde{\mathbf{x}}\|}{r'} (\mathbf{x}_{1,t+1} - \tilde{\mathbf{x}}) = \frac{\|\mathbf{y}_{2,t+1} - \tilde{\mathbf{x}}\|}{r'} (\mathbf{x}_{1,t+1} - \tilde{\mathbf{x}}).$$

Then we finish the proof of the induction.

$\square$

Note that $\nabla^2 f(\tilde{\mathbf{x}})$ has the following eigendecomposition: $\nabla^2 f(\tilde{\mathbf{x}}) = \sum_{i=1}^n \lambda_i \mathbf{u}_i \mathbf{u}_i^\mathsf{T}$, where $\{\mathbf{u}_{i=1}^n\}$ forms an orthonormal basis of $\mathbb{R}^d$. Without loss of generality, assume that $\lambda_1 \leq \lambda_2 \leq \cdots \leq \lambda_d$ and $\lambda_1 \leq -\sqrt{\rho\epsilon}$. If $\lambda_d \leq -\sqrt{\rho\epsilon}/2$, then Lemma holds directly. Then we prove the case when $\lambda_d \geq -\sqrt{\rho\epsilon}/2$ and assume that $\lambda_p \leq -\sqrt{\rho\epsilon} \leq \lambda_{p+1}(p > 1)$. Let $\mathcal{S}$ be the subspace of $\mathbb{R}^d$ spanned by $\{\mathbf{u}_1, \mathbf{u}_2, \ldots, \mathbf{u}_p\}$ and $\mathcal{S}^c$ be subspace spanned by $\{\mathbf{u}_{p+1}, \mathbf{u}_{p+2}, \ldots, \mathbf{u}_d\}$. Then we have the following lemma:

**Lemma 27.** *Denote* $\alpha_t = \frac{\|\mathbf{x}_{t,\mathcal{S}} - \tilde{\mathbf{x}}\|}{\|\mathbf{x}_t - \tilde{\mathbf{x}}\|}$, *where* $\mathbf{x}_{t,\mathcal{S}}$ *is the component of* $\mathbf{x}_t$ *in the subspace* $\mathcal{S}$. *Then, during all the* $\mathcal{T}'$ *iterations of Algorithm 4, we have* $\alpha_t \geq \alpha_{\min} = \frac{\delta_0}{8} \sqrt{\frac{\pi}{d}}$, *given that* $\alpha_0 = \sqrt{\frac{\pi}{d}} \delta_0$.

*Proof.* Define $\mathbf{x}_{-1} = \mathbf{x}_0 - \mathbf{v}_0$. Without loss of generality, we assume that $\tilde{\mathbf{x}} = 0$. Consider the worst case that $\alpha_0 = \sqrt{\frac{\pi}{d}} \delta_0$ and the component $x_{0,d}$ along $\mathbf{u}_d$ equals $0$. Assume that the eigenvalues satisfy

$$\lambda_2 = \cdots = \lambda_p = \lambda_{p+1} = \cdots = \lambda_{d-1} = -\sqrt{\rho\epsilon}.$$

Define $\Delta = \frac{\|\mathbf{y}_t\|}{r'} \left( \hat{\nabla} f(\mathbf{y}_t \frac{r'}{\|\mathbf{y}_t\|}) - \hat{\nabla} f(\mathbf{0}) - \nabla^2 f(\mathbf{0}) \frac{r'}{\|\mathbf{y}_t\|} \mathbf{y}_t \right)$ and assume that $\Delta$ lies in the direction that make $\alpha_t$ as small as possible. Then, the component $\Delta_{\mathcal{S}}$ in $\mathcal{S}$ should be in the opposite direction to $\mathbf{v}_{\mathcal{S}}$, and the component $\Delta_{\mathcal{S}^c}$ in $\mathcal{S}^c$ should be in the direction of $\mathbf{v}_{\mathcal{S}^c}$. Then we have both $\|\mathbf{x}_{t,\mathcal{S}^c}\|/\|\mathbf{x}_t\|$ and $\|\mathbf{y}_{t,\mathcal{S}^c}\|/\|\mathbf{y}_t\|$ being non-decreasing. Note that

$$\mathbf{x}_{t+2} = \mathbf{x}_{t+1} + (1 - \theta)(\mathbf{x}_{t+1} - \mathbf{x}_t) - \eta\Delta - \eta\nabla^2 f(\mathbf{0})(\mathbf{x}_{t+1} + (1 - \theta)(\mathbf{x}_{t+1} - \mathbf{x}_t))$$

Then we consider the following recurrence formula:

$$\|\mathbf{x}_{t+2,\mathcal{S}^c}\| \leq (1 + \eta\sqrt{\rho\epsilon})(\|\mathbf{x}_{t+1,\mathcal{S}^c}\| + (1 - \theta)(\|\mathbf{x}_{t+1,\mathcal{S}^c}\| - \|\mathbf{x}_{t,\mathcal{S}^c}\|)) + \eta\|\Delta_{\mathcal{S}^c}\|.$$

Since $\|\mathbf{x}_{t,\mathcal{S}^c}\|/\|\mathbf{x}_t\|$ is non-decreasing, we have

$$\frac{\|\Delta_{\mathcal{S}^c}\|}{\|\mathbf{x}_{t+1,\mathcal{S}^c}\|} \leq \frac{\|\Delta\|}{\|\mathbf{x}_{t+1,\mathcal{S}^c}\|} \leq \frac{\|\Delta\|}{\|\mathbf{x}_{t+1}\|} \frac{\|\mathbf{x}_0\|}{\|\mathbf{x}_{0,\mathcal{S}^c}\|} \leq \frac{\|\Delta\|}{\|\mathbf{x}_{t+1}\|} \frac{\|\mathbf{x}_0\|}{\|\mathbf{x}_0\| - \|\mathbf{x}_{0,\mathcal{S}}\|} = \frac{\|\Delta\|}{\|\mathbf{x}_{t+1}\|} \frac{1}{1 - \alpha_0}$$

$$\leq \frac{2\|\Delta\|}{\|\mathbf{x}_{t+1}\|} \leq \frac{2}{r'}\rho(\frac{r'^2}{2} + \frac{\sqrt{d}\mu^2}{3}) \leq 2\rho r',$$

where the last second step uses Lemma 2 and the last step is due to our choice of $\mu$ such that $\sqrt{d}\mu^2 \leq r'^2$. Then we have

$$\|\mathbf{x}_{t+2,\mathcal{S}^c}\| \leq (1 + \eta\sqrt{\rho\epsilon} + 2\eta\rho r')((2-\theta)\|\mathbf{x}_{t+1,\mathcal{S}^c}\| - (1-\theta)\|\mathbf{x}_{t,\mathcal{S}^c}\|).$$

Then by Lemma 17, we have

$$
\begin{aligned}
\|\mathbf{x}_{t,\mathcal{S}^c}\| \leq & (\frac{1+\kappa_{\mathcal{S}^c}}{2})^t \left( (-\frac{2-\theta-\mu_{\mathcal{S}^c}}{2\mu_{\mathcal{S}^c}}\|\mathbf{x}_{0,\mathcal{S}^c}\| + \frac{1}{(1+\kappa_{\mathcal{S}^c})\mu_{\mathcal{S}^c}}(1+\kappa_{\mathcal{S}^c})\|\mathbf{x}_{0,\mathcal{S}^c}\|) \cdot (2-\theta+\mu_{\mathcal{S}^c})^t \right. \\
& \left. + (\frac{2-\theta+\mu_{\mathcal{S}^c}}{2\mu_{\mathcal{S}^c}}\|\mathbf{x}_{0,\mathcal{S}^c}\| - \frac{1}{(1+\kappa_{\mathcal{S}^c})\mu_{\mathcal{S}^c}}(1+\kappa_{\mathcal{S}^c})\|\mathbf{x}_{0,\mathcal{S}^c}\|) \cdot (2-\theta-\mu_{\mathcal{S}^c})^t \right) \\
\leq & (\frac{1+\kappa_{\mathcal{S}^c}}{2})^t \left( (-\frac{2-\theta-\mu_{\mathcal{S}^c}}{2\mu_{\mathcal{S}^c}}\|\mathbf{x}_{0,\mathcal{S}^c}\| + \frac{\|\mathbf{x}_{0,\mathcal{S}^c}\|}{\mu_{\mathcal{S}^c}}) + (\frac{2-\theta+\mu_{\mathcal{S}^c}}{2\mu_{\mathcal{S}^c}}\|\mathbf{x}_{0,\mathcal{S}^c}\| - \frac{\|\mathbf{x}_{0,\mathcal{S}^c}\|}{\mu_{\mathcal{S}^c}}) \right) \cdot (2-\theta+\mu_{\mathcal{S}^c})^t \\
= & (\frac{1+\kappa_{\mathcal{S}^c}}{2})^t\|\mathbf{x}_{0,\mathcal{S}^c}\|(2-\theta+\mu_{\mathcal{S}^c})^t,
\end{aligned}
$$

where $\kappa_{\mathcal{S}^c} = \eta\sqrt{\rho\epsilon} + 2\eta\rho r'$, $\mu_{\mathcal{S}^c} = \sqrt{((2-\theta)^2 - \frac{4(1-\theta)}{1+\kappa_{\mathcal{S}^c}})}$. Suppose for some value $t$, we have $\alpha_k \geq \alpha_{\min}$ for any $1 \leq k \leq t+1$. Then we have

$$\|x_{t+2,\mathcal{S}}\| \geq (1+\eta\sqrt{\rho\epsilon}) \geq (1+\eta\sqrt{\rho\epsilon})(\|\mathbf{x}_{t+1,\mathcal{S}}\| + (1-\theta)(\|\mathbf{x}_{t+1,\mathcal{S}}\| - \|\mathbf{x}_{t,\mathcal{S}}\|)) - \eta\|\Delta_{\mathcal{S}}\|.$$

Since $\|\mathbf{x}_{t+1,\mathcal{S}}\|/\|\mathbf{x}_{t+1}\| \geq \alpha_{\min}$ holds for all $t > 0$, we have $\frac{\|\mathbf{y}_{t+1,\mathcal{S}}\|}{\|\mathbf{y}_{t+1}\|} \geq \alpha_{\min}$, then

$$\frac{\|\Delta_{\mathcal{S}}\|}{\|\mathbf{y}_{t+1,\mathcal{S}}\|} \leq \frac{\|\Delta\|}{\alpha_{\min}\|\mathbf{y}_{t+1}\|} \leq \frac{1}{\alpha_{\min}}\rho(\frac{r'^2}{2} + \frac{\sqrt{d}\mu^2}{3}) \leq \frac{\rho r'}{\alpha_{\min}},$$

where the last second step uses Lemma 2 and the last step is due to our choice of $\mu$ such that $\sqrt{d}\mu^2 \leq r'^2$. Then we have

$$\|x_{t+2,\mathcal{S}}\| \geq (1+\eta\sqrt{\rho\epsilon} - \frac{\eta\rho r'}{\alpha_{\min}})((2-\theta)\|\mathbf{x}_{t+1,\mathcal{S}}\| - (1-\theta)\|\mathbf{x}_{t,\mathcal{S}}\|).$$

Then by Lemma 17, we have

$$
\begin{aligned}
\|\mathbf{x}_{t,\mathcal{S}}\| \geq & (\frac{1+\kappa_{\mathcal{S}}}{2})^t \left( (-\frac{2-\theta-\mu_{\mathcal{S}}}{2\mu_{\mathcal{S}}}\|\mathbf{x}_{0,\mathcal{S}}\| + \frac{1}{(1+\kappa_{\mathcal{S}})\mu_{\mathcal{S}}}(1+\kappa_{\mathcal{S}})\|\mathbf{x}_{0,\mathcal{S}}\|) \cdot (2-\theta+\mu_{\mathcal{S}})^t \right. \\
& \left. + (\frac{2-\theta+\mu_{\mathcal{S}}}{2\mu_{\mathcal{S}}}\|\mathbf{x}_{0,\mathcal{S}}\| - \frac{1}{(1+\kappa_{\mathcal{S}})\mu_{\mathcal{S}}}(1+\kappa_{\mathcal{S}})\|\mathbf{x}_{0,\mathcal{S}}\|) \cdot (2-\theta-\mu_{\mathcal{S}})^t \right) \\
\geq & (\frac{1+\kappa_{\mathcal{S}}}{2})^t \cdot (-\frac{2-\theta-\mu_{\mathcal{S}}}{2\mu_{\mathcal{S}}}\|\mathbf{x}_{0,\mathcal{S}}\| + \frac{\|\mathbf{x}_{0,\mathcal{S}}\|}{\mu_{\mathcal{S}}}) \cdot (2-\theta+\mu_{\mathcal{S}})^t \\
= & (\frac{1+\kappa_{\mathcal{S}}}{2})^t \cdot \frac{\|\mathbf{x}_{0,\mathcal{S}}\|}{2} \cdot (2-\theta+\mu_{\mathcal{S}})^t
\end{aligned}
$$

where $\kappa_{\mathcal{S}} = \eta\sqrt{\rho\epsilon} - \frac{\eta\rho r'}{\alpha_{\min}}$, $\mu_{\mathcal{S}} = \sqrt{((2-\theta)^2 - \frac{4(1-\theta)}{1+\kappa_{\mathcal{S}}})}$. Then we have

$$\frac{\|\mathbf{x}_{t,\mathcal{S}}\|}{\|\mathbf{x}_{t,\mathcal{S}^c}\|} \geq (\frac{1+\kappa_{\mathcal{S}}}{1+\kappa_{\mathcal{S}^c}})^t \frac{\|\mathbf{x}_{0,\mathcal{S}}\|}{2\|\mathbf{x}_{0,\mathcal{S}^c}\|}(\frac{2-\theta+\mu_{\mathcal{S}}}{2-\theta+\mu_{\mathcal{S}^c}})^t,$$

where

$$\frac{1+\kappa_{\mathcal{S}}}{1+\kappa_{\mathcal{S}^c}} \geq (1+\kappa_{\mathcal{S}})(1-\kappa_{\mathcal{S}^c}) = 1 - (\frac{1}{\alpha_{\min}}+2)\eta\rho r' - \kappa_{\mathcal{S}}\kappa_{\mathcal{S}^c} \geq 1 - 2\eta\rho r'/\alpha_{\min},$$

$$
\begin{aligned}
\frac{2-\theta+\mu_{\mathcal{S}}}{2-\theta+\mu_{\mathcal{S}^c}} \geq & (1+\frac{\mu_{\mathcal{S}}}{2-\theta})(1-\frac{\mu_{\mathcal{S}^c}}{2-\theta}) = (1+\frac{1}{2-\theta}\sqrt{(2-\theta)^2 - \frac{4(1-\theta)}{1+\kappa_{\mathcal{S}}}})(1-\frac{1}{2-\theta}\sqrt{(2-\theta)^2 - \frac{4(1-\theta)}{1+\kappa_{\mathcal{S}^c}}}) \\
= & (1+\frac{1}{2-\theta}\sqrt{\frac{\theta^2 + \kappa_{\mathcal{S}}(2-\theta)^2}{1+\kappa_{\mathcal{S}}}})(1-\frac{1}{2-\theta}\sqrt{\frac{\theta^2 + \kappa_{\mathcal{S}^c}(2-\theta)^2}{1+\kappa_{\mathcal{S}^c}}}) \geq 1 - \frac{2(\kappa_{\mathcal{S}^c}-\kappa_{\mathcal{S}})}{\theta} \geq 1 - \frac{3\eta\rho r'}{\alpha_{\min}\theta}.
\end{aligned}
$$

Then we have

$$\frac{\|\mathbf{x}_{t,\mathcal{S}}\|}{\|\mathbf{x}_{t,\mathcal{S}^c}\|} \geq \frac{\|\mathbf{x}_{0,\mathcal{S}}\|}{2\|\mathbf{x}_{0,\mathcal{S}^c}\|}(1 - \frac{4\rho r'}{\alpha_{\min}\theta})^t \geq \frac{\|\mathbf{x}_{0,\mathcal{S}}\|}{2\|\mathbf{x}_{0,\mathcal{S}^c}\|}(1 - 1/\mathscr{T}')^t \geq \frac{\|\mathbf{x}_{0,\mathcal{S}}\|}{2\|\mathbf{x}_{0,\mathcal{S}^c}\|}\exp(-\frac{t}{\mathscr{T}'-1}) \geq \frac{\|\mathbf{x}_{0,\mathcal{S}}\|}{4\|\mathbf{x}_{0,\mathcal{S}^c}\|}.$$

So we have

$$\alpha_t = \frac{\|\mathbf{x}_{t,\mathcal{S}}\|}{\sqrt{\|\mathbf{x}_{t,\mathcal{S}}\|^2 + \|\mathbf{x}_{t,\mathcal{S}^c}\|^2}} \geq \frac{\|\mathbf{x}_{0,\mathcal{S}}\|}{8\|\mathbf{x}_{0,\mathcal{S}^c}\|} \geq \alpha_{\min}.$$

Thus for all $t \leq \mathscr{T}'$, we have $\alpha_t \geq \alpha_{\min}$. $\qquad\square$

**Proof of Lemma 7**

*Proof.* As stated above, we only need to prove the case when $\lambda_d \geq -\frac{\sqrt{\rho\epsilon}}{2}$. Then there exist some $p'$ such that $\lambda_{p'} \leq -\sqrt{\rho\epsilon}/2 \leq \lambda_{p'+1}$. Let $\mathcal{S}'$ be the subspace of $\mathbb{R}^d$ spanned by $\{\mathbf{u}_1, \mathbf{u}_2, \ldots, \mathbf{u}_{p'}\}$ and $\mathcal{S}'^c$ be the complementary subspace. Define $\mathbf{x}_{t,\mathcal{S}'} = \sum_{i=1}^{p'} \langle \mathbf{u}_i, \mathbf{x}_t \rangle \mathbf{u}_i$, $\mathbf{x}_{t,\mathcal{S}'^c} = \sum_{i=p'+1}^{d} \langle \mathbf{u}_i, \mathbf{x}_t \rangle \mathbf{u}_i$, and let $\alpha_t = \|\mathbf{x}_{t,\mathcal{S}}\|/\|\mathbf{x}_t\|$. We know that with probability at least

$$1 - \sqrt{\frac{\pi}{d}}\delta_0 \frac{\mathrm{Vol}(\mathbb{B}_0^{d-1}(1))}{\mathrm{Vol}(\mathbb{B}_0^d(1))} \geq 1 - \sqrt{\frac{\pi}{d}}\delta_0\sqrt{\frac{d}{\pi}} = 1 - \delta_0,$$

we have $\alpha_0 \geq \sqrt{\frac{\pi}{d}}\delta_0$. Then we prove that there exists some $t_0$ with $1 \leq t \leq \mathscr{T}'$ such that

$$\frac{\|\mathbf{x}_{t_0,\mathcal{S}'^c}\|}{\|\mathbf{x}_{t_0}\|} \leq \frac{\sqrt{\rho\epsilon}}{8\ell}.$$

Assume the contrary holds that for any $1 \leq t \leq \mathscr{T}'$, $\frac{\|\mathbf{x}_{t_0,\mathcal{S}'^c}\|}{\|\mathbf{x}_{t_0}\|} > \frac{\sqrt{\rho\epsilon}}{8\ell}$ and $\frac{\|\mathbf{y}_{t_0,\mathcal{S}'^c}\|}{\|\mathbf{y}_{t_0}\|} > \frac{\sqrt{\rho\epsilon}}{8\ell}$. Then we consider the case when $\|\mathbf{x}_{t,\mathcal{S}'^c}\|$ achieves the largest possible value and we have the following recurrence formula:

$$\|\mathbf{x}_{t+2,\mathcal{S}'^c}\| \leq (1 + \eta\sqrt{\rho\epsilon}/2)(\|\mathbf{x}_{t+1,\mathcal{S}'^c}\| + (1-\theta)(\|\mathbf{x}_{t+1,\mathcal{S}'^c}\| - \|\mathbf{x}_{t,\mathcal{S}'^c}\|)) + \eta\|\Delta_{\mathcal{S}'^c}\|.$$

Since $\frac{\|\mathbf{y}_{t_0,\mathcal{S}'^c}\|}{\|\mathbf{y}_{t_0}\|} > \frac{\sqrt{\rho\epsilon}}{8\ell}$ for any $1 \leq k \leq t+1$, then we have

$$\frac{\eta\|\Delta_{\mathcal{S}'^c}\|}{\|\mathbf{x}_{t+1,\mathcal{S}'^c}\| + (1-\theta)(\|\mathbf{x}_{t+1,\mathcal{S}'^c}\| - \|\mathbf{x}_{t,\mathcal{S}'^c}\|)} \leq \frac{\|\Delta_{\mathcal{S}'^c}\|}{4\ell\|\mathbf{y}_{\mathcal{S}'^c}\|} \leq \frac{2\rho}{\sqrt{\rho\epsilon}r'}(\frac{r'^2}{2} + \frac{\sqrt{d}\mu^2}{3}) \leq \frac{2\rho r'}{\sqrt{\rho\epsilon}},$$

where the last is step is due to Lemma 2 and our choice of $\mu$ such that $\sqrt{d}\mu^2 \leq r'^2$. Then we have

$$\|\mathbf{x}_{t+2,\mathcal{S}'^c}\| \leq (1 + \eta\sqrt{\rho\epsilon}/2 + 2\rho r'/\sqrt{\rho\epsilon})((2-\theta)\|\mathbf{x}_{t+1,\mathcal{S}'^c}\| - (1-\theta)\|\mathbf{x}_{t,\mathcal{S}'^c}\|).$$

Then we have

$$\|\mathbf{x}_{t,\mathcal{S}'^c}\| \leq \|\mathbf{x}_{0,\mathcal{S}'^c}\|(\frac{1+\kappa_{\mathcal{S}'^c}}{2})^t(2 - \theta + \mu_{\mathcal{S}'^c})^t,$$

where $\kappa_{\mathcal{S}'^c} = \eta\sqrt{\rho\epsilon}/2 + 2\rho r'/\sqrt{\rho\epsilon}$, $\mu_{\mathcal{S}'^c} = \sqrt{(2-\theta)^2 - \frac{4(1-\theta)}{1+\kappa_{\mathcal{S}'^c}}}$. By Lemma 27, we have $\|\mathbf{x}_{t,\mathcal{S}}\| \geq (\frac{1+\kappa_{\mathcal{S}}}{2})^t \cdot \frac{\|\mathbf{x}_{0,\mathcal{S}}\|}{2} \cdot (2 - \theta + \mu_{\mathcal{S}})^t$ for any $1 \leq t \leq \mathscr{T}'$. Then we have

$$\frac{\|\mathbf{x}_{\mathscr{T}',\mathcal{S}'^c}\|}{\|\mathbf{x}_{\mathscr{T}',\mathcal{S}}\|} \leq \frac{2\|\mathbf{x}_{0,\mathcal{S}'^c}\|}{\|\mathbf{x}_{0,\mathcal{S}}\|}(\frac{1+\kappa_{\mathcal{S}'^c}}{1+\kappa_{\mathcal{S}}})^{\mathscr{T}'}(\frac{2-\theta+\mu_{\mathcal{S}'^c}}{2-\theta+\mu_{\mathcal{S}}})^{\mathscr{T}'}$$

$$\leq \frac{2}{\alpha_0}(\frac{1+\kappa_{\mathcal{S}'^c}}{1+\kappa_{\mathcal{S}}})^{\mathscr{T}'}(\frac{2-\theta+\mu_{\mathcal{S}'^c}}{2-\theta+\mu_{\mathcal{S}}})^{\mathscr{T}'} \leq \frac{2}{\delta_0}\sqrt{\frac{d}{\pi}}(\frac{1+\kappa_{\mathcal{S}'^c}}{1+\kappa_{\mathcal{S}}})^{\mathscr{T}'}(\frac{2-\theta+\mu_{\mathcal{S}'^c}}{2-\theta+\mu_{\mathcal{S}}})^{\mathscr{T}'},$$

where

$$\frac{1+\kappa_{\mathcal{S}'^c}}{1+\kappa_{\mathcal{S}}} \leq \frac{1}{(1+\kappa_{\mathcal{S}})(1-\kappa_{\mathcal{S}'^c})} \leq \frac{1}{1+(\kappa_{\mathcal{S}}-\kappa_{\mathcal{S}'^c})/2} = 1 - \frac{(\kappa_{\mathcal{S}}-\kappa_{\mathcal{S}'^c})/2}{1+(\kappa_{\mathcal{S}}-\kappa_{\mathcal{S}'^c})/2} \leq 1 - \frac{\kappa_{\mathcal{S}}-\kappa_{\mathcal{S}'^c}}{4}$$

$$\leq 1 - \frac{\eta\sqrt{\rho\epsilon}/2 - \rho r'(\frac{1}{\sqrt{\rho\epsilon}} + \frac{1}{\alpha_{\min}})}{4} \leq 1 - \frac{\eta\sqrt{\rho\epsilon}}{16}$$

$$\frac{2-\theta+\mu_{\mathcal{S}'^c}}{2-\theta+\mu_{\mathcal{S}}} = \frac{1+\frac{\mu_{\mathcal{S}'^c}}{2-\theta}}{1+\frac{\mu_{\mathcal{S}}}{2-\theta}} \leq \frac{1}{(1+\frac{\mu_{\mathcal{S}}}{2-\theta})(1-\frac{\mu_{\mathcal{S}'^c}}{2-\theta})} = \frac{1}{(1+\sqrt{1-\frac{4(1-\theta)}{(1+\kappa_{\mathcal{S}})(2-\theta)^2}})(1-\sqrt{1-\frac{4(1-\theta)}{(1+\kappa_{\mathcal{S}'^c})(2-\theta)^2}})}$$

$$\leq \frac{1}{1+\frac{2(\kappa_{\mathcal{S}}-\kappa_{\mathcal{S}'^c})}{\theta}} = 1 - \frac{\frac{2(\kappa_{\mathcal{S}}-\kappa_{\mathcal{S}'^c})}{\theta}}{1+\frac{2(\kappa_{\mathcal{S}}-\kappa_{\mathcal{S}'^c})}{\theta}} \leq 1 - \frac{\kappa_{\mathcal{S}}-\kappa_{\mathcal{S}'^c}}{\theta} \leq 1 - \frac{\eta\sqrt{\rho\epsilon}}{4\theta} = 1 - \frac{(\rho\epsilon)^{1/4}}{16\sqrt{\ell}}.$$

Then we have

$$\frac{\|\mathbf{x}_{\mathcal{T}',\mathcal{S}'^c}\|}{\|\mathbf{x}_{\mathcal{T}',\mathcal{S}}\|} \leq \frac{2}{\delta_0}\sqrt{\frac{d}{\pi}}(1-\frac{(\rho\epsilon)^{1/4}}{16\sqrt{\ell}})^{\mathcal{T}'} \leq \frac{\sqrt{\rho\epsilon}}{8\ell}.$$

So we conclude that there exist some $1 \leq t_0 \leq \mathcal{T}'$ such that $\frac{\|\mathbf{x}_{t_0,\mathcal{S}'^c}\|}{\|\mathbf{x}_{t_0}\|} \leq \frac{\sqrt{\rho\epsilon}}{8\ell}$. Consider the normalized vector $\hat{\mathbf{e}} = \frac{\mathbf{x}_{t_0}}{r}$, then we have $\|\hat{\mathbf{e}}_{\mathcal{S}'^c}\| \leq \frac{\sqrt{\rho\epsilon}}{8\ell}$ and $\|\hat{\mathbf{e}}_{\mathcal{S}'}\| \geq \|\hat{\mathbf{e}}\| - \|\hat{\mathbf{e}}_{\mathcal{S}'^c}\| \geq 1 - \frac{\sqrt{\rho\epsilon}}{8\ell}$.
Then we have

$$\hat{\mathbf{e}}^\mathsf{T}\nabla^2 f(\mathbf{0})\hat{\mathbf{e}} = (\hat{\mathbf{e}}_{\mathcal{S}'^c} + \hat{\mathbf{e}}_{\mathcal{S}'})^\mathsf{T}\nabla^2 f(\mathbf{0})(\hat{\mathbf{e}}_{\mathcal{S}'^c} + \hat{\mathbf{e}}_{\mathcal{S}'}) = \hat{\mathbf{e}}_{\mathcal{S}'^c}^\mathsf{T}\nabla^2 f(\mathbf{0})\hat{\mathbf{e}}_{\mathcal{S}'^c} + \hat{\mathbf{e}}_{\mathcal{S}'}^\mathsf{T}\nabla^2 f(\mathbf{0})\hat{\mathbf{e}}_{\mathcal{S}'}$$

$$\leq \ell\|\hat{\mathbf{e}}_{\mathcal{S}'^c}\|^2 - \frac{\sqrt{\rho\epsilon}}{2}\|\hat{\mathbf{e}}_{\mathcal{S}'}\|^2 \leq \frac{\rho\epsilon}{64\ell} - \frac{\sqrt{\rho\epsilon}}{2}(1-\frac{\sqrt{\rho\epsilon}}{8\ell})^2 \leq -\frac{\sqrt{\rho\epsilon}}{4}.$$

$\square$

### D.1 Proof of Theorem 2

*Proof.* Recall the parameters setting of Algorithm 3:

$$\delta_0 = \frac{\delta}{384\Delta_f}\sqrt{\frac{\epsilon^3}{\rho}}, \quad \eta = \frac{1}{4\ell}, \quad \theta = \frac{1}{4\sqrt{\kappa}}, \quad \gamma = \frac{\theta^2}{\eta}, \quad s = \frac{\gamma}{4\rho}, \quad \mathcal{T}' = 32\sqrt{\kappa}\log(\frac{\ell\sqrt{d}}{\delta_0\sqrt{\rho\epsilon}}),$$

$$\mathscr{E} = \sqrt{\frac{\epsilon^3}{\rho}}c_A^{-7}, \quad r' = \frac{\delta_0\epsilon}{32}\sqrt{\frac{\pi}{\rho d}},$$

where $c_A$ is a large enough constant. Define a new parameter $\tilde{\mathcal{T}} = \sqrt{\kappa}c_A$. From Lemma 5 we know if $\|\hat{\nabla}f(\mathbf{x}_\tau)\| \geq \frac{3\epsilon}{4}$ for any $\tau \in [0, \tilde{\mathcal{T}}]$, then by running Algorithm 3 we have $E_{\tilde{\mathcal{T}}} - E_0 \leq -\mathscr{E}$. Then we first assume that for each time we can escape saddle points successfully, *i.e.*, after $\mathcal{T}'$ iterations of the perturbation step, we have $\hat{\mathbf{e}}^\mathsf{T}\nabla^2 f(\tilde{\mathbf{x}})\hat{\mathbf{e}} \leq -\frac{\sqrt{\rho\epsilon}}{4}$. Then by Lemma 8, we have $\min\{f(\tilde{\mathbf{x}} - \frac{1}{4}\sqrt{\frac{\epsilon}{\rho}}\hat{\mathbf{e}}), f(\tilde{\mathbf{x}} + \frac{1}{4}\sqrt{\frac{\epsilon}{\rho}}\hat{\mathbf{e}})\} \leq f(\tilde{\mathbf{x}} - \frac{f'_{\hat{\mathbf{e}}}(\tilde{\mathbf{x}})}{4|f'_{\hat{\mathbf{e}}}(\tilde{\mathbf{x}})|}\sqrt{\frac{\epsilon}{\rho}}\hat{\mathbf{e}}) \leq f(\tilde{\mathbf{x}}) - \frac{1}{384}\sqrt{\frac{\epsilon^3}{\rho}}$ and the total times of random perturbations is no more than $384(f(\mathbf{x}_0) - f^*)\sqrt{\frac{\rho}{\epsilon^3}}$. By union bound, the probability that at least one time the negative curvature finding fails to escape saddle points is upper bounded by $384(f(\mathbf{x}_0) - f^*)\sqrt{\frac{\rho}{\epsilon^3}}\delta_0 \leq \delta$.

Then we assume that we never encounter a SOSP in the rest steps. Set the total number of iterations to be $T = \max\{\frac{4\Delta_f(\tilde{\mathcal{T}}+\mathcal{T}')}{\mathscr{E}}, 768\Delta_f\mathcal{T}'\sqrt{\frac{\rho}{\epsilon^3}}\} = \mathcal{O}\left(\frac{\Delta_f\ell^{1/2}\rho^{1/4}}{\epsilon^{7/4}}\log(\frac{\ell\sqrt{d}\Delta_f}{\delta\epsilon^2})\right)$. Denote by the $N_{\tilde{\mathcal{T}}}$ the number of periods containing only large gradient steps, then we have

$$N_{\tilde{\mathcal{T}}} \geq \frac{T}{2(\tilde{\mathcal{T}}+\mathcal{T}')} - 384(f(\mathbf{x}_0) - f^*)\sqrt{\frac{\rho}{\epsilon^3}} \geq (2c_A^7 - 84)\Delta_f\sqrt{\frac{\rho}{\epsilon^3}} \geq \frac{\Delta_f}{\mathscr{E}}.$$

By Lemma 5 we have the Hamiltonian will decrease by $N_{\tilde{\mathcal{T}}}\mathscr{E} \geq \Delta_f$, which cause a contradiction. Thus we have with probability at least $1 - \delta$, we must encounter an $\epsilon$-approximate SOSP during the $T$ iterations. $\square$

### E Parameter Settings of the Numerical Experiments

Table 2: Parameter settings of the cubic regularization problem experiment.

| Algorithm | Parameters |
|---|---|
| **d = 20, 100, 200, 1000** | |
| PAGD | $\ell = 10, \eta = 1/\ell, r = 0.01, t_{\text{thresh}} = 30, g_{\text{thresh}} = e/100$ |
| ZO-Perturbed-AGD | $\ell = 10, \rho = 1, \eta = 1/\eta, r = 0.001, \mu = 0.001$ |
| ZO-Perturbed-AGD-ANCF | $\ell = 10, \rho = 1, \eta = 1/\eta, r = 0.001, \mu = 0.001$ |

Table 3: Parameter settings of the cubic quartic function experiment.

| Algorithm | Parameters |
|---|---|
| **d = 20** | |
| PAGD | $\ell = 20, \eta = 1/\ell, r = 10^{-3}, t_{\text{thresh}} = 10, g_{\text{thresh}} = e/100$ |
| RSPI | $\ell = 20, \sigma_1 = 1, \sigma_2 = 0.6, \rho_{\sigma_1} = 0.8, T_{\sigma_1} = 5, T_{\text{DFPI}} = 20$ |
| RSPI(SPSA) | $\ell = 20, \sigma_1 = 1.75, \sigma_2 = 0.65, \rho_{\sigma_1} = 0.8, T_{\sigma_1} = 15, T_{\text{DFPI}} = 100$ |
| ZO-GD-NCF | $\ell = 20, \rho = 10, \eta = 1/\ell$ |
| ZO-Perturbed-AGD | $\ell = 20, \rho = 10, \eta = 1/\ell, r = 0.01, \mu = 0.001$ |
| ZO-Perturbed-AGD-ANCF | $\ell = 20, \rho = 10, \eta = 1/\ell, r = 0.01, \mu = 0.01$ |
| **d = 100** | |
| PAGD | $\ell = 100, \eta = 1/\ell, r = 10^{-3}, t_{\text{thresh}} = 10, g_{\text{thresh}} = e/100$ |
| RSPI | $\ell = 100, \sigma_1 = 1, \sigma_2 = 0.65, \rho_{\sigma_1} = 0.95, T_{\sigma_1} = 15, T_{\text{DFPI}} = 20$ |
| RSPI(SPSA) | $\ell = 100, \sigma_1 = 1.75, \sigma_2 = 0.65, \rho_{\sigma_1} = 0.95, T_{\sigma_1} = 15, T_{\text{DFPI}} = 100$ |
| ZO-GD-NCF | $\ell = 100, \rho = 10, \eta = 1/\ell$ |
| ZO-Perturbed-AGD | $\ell = 100, \rho = 10, \eta = 1/\ell, r = 0.01, \mu = 0.001$ |
| ZO-Perturbed-AGD-ANCF | $\ell = 2 = 100, \rho = 10, \eta = 1/\ell, r = 0.01, \mu = 0.01$ |
| **d = 200** | |
| PAGD | $\ell = 110, \eta = 1/\ell, r = 10^{-3}, t_{\text{thresh}} = 10, g_{\text{thresh}} = e/100$ |
| RSPI | $\ell = 200, \sigma_1 = 1.5, \sigma_2 = 0.65, \rho_{\sigma_1} = 0.96, T_{\sigma_1} = 15, T_{\text{DFPI}} = 20$ |
| RSPI(SPSA) | $\ell = 200, \sigma_1 = 1.75, \sigma_2 = 0.65, \rho_{\sigma_1} = 0.98, T_{\sigma_1} = 15, T_{\text{DFPI}} = 100$ |
| ZO-GD-NCF | $\ell = 150, \rho = 10, \eta = 1/\ell$ |
| ZO-Perturbed-AGD | $\ell = 200, \rho = 10, \eta = 1/\ell, r = 0.01, \mu = 0.001$ |
| ZO-Perturbed-AGD-ANCF | $\ell = 200, \rho = 10, \eta = 1/\ell, r = 0.01, \mu = 0.01$ |

