# OpenReview forum: "Faster Gradient-Free Methods for Escaping Saddle Points"
_ICLR.cc/2023/Conference — ICLR 2023 notable top 25%_

### Official Review · Reviewer_kuK4 · 2022-10-20

**Confidence:** 5
**Correctness:** 4
**Technical Novelty And Significance:** 4
**Empirical Novelty And Significance:** 3
**Recommendation:** 8

**Clarity, Quality, Novelty And Reproducibility:**

From my perspective, the clarity and quality of this paper is in general good, though the paper still has space to improve as I mentioned above. Novelty is excellent -- this is the first paper proving the 1/eps^1.75 convergence rate for escaping from saddle points using zeroth-order methods, matching that of first-order methods. Reproducibility is also excellent -- theoretical results are stated with detailed proofs and the numerical experiments are provided with codes and detailed explanations.

**Strength And Weaknesses:**

From my perspective, this paper has the following strengths:

- This is the first work which gives zeroth-order methods for escaping saddle points with matching eps dependence compared to first-order methods. This is a great contribution to me and significantly improves our understanding of zeroth-order methods.

- Two algorithms are proposed to achieve the result of ~O(1/eps^1.75) iteration complexity and ~O(d/eps^1.75) query complexity. The second one based on negative curvature finding only has one logarithmic term in d.

- The theory results are corroborated by numerical experiments on two instances: cubic regularization and quartic functions. Codes are also provided in the supplementary material.

Nevertheless, the paper may still have space to improve from the following aspects:

- It would be very helpful if the authors can illustrate more on the practical cases where both escaping from saddle points is a vital problem and we have to use zeroth-order methods instead of first-order methods. In the introduction, the authors mentioned the situations where the calculation of explicit gradients is expensive or even infeasible, such as black-box adversarial attack on deep neural networks. This is still a bit vague to me; would be helpful to explain how we have the function evaluation access in such problems.

- The writing in this paper is a bit sloppy in general and has space to further polish. From a relatively superficial reading, I can already spot the following glitches:

- - Second paragraph in Page 1: non-convex problem 1 -> non-convex problem (1). This also applies to a few following places.

- - Third paragraph in Page 1: O(1/eps^2) -> Theta(1/eps^2) (Here, the authors are talking about the optimality of gradient descent, so a lower bound or tight bound should be reflected here.)

- - Second half of Page 2: proposed a first-order Negative curvature finding framework -> … negative (should be small letter instead of capital letter here)

- - End of Page 2: with fewer function query complexity -> with smaller function query complexity, or with fewer function queries

- - Beginning of Page 3, second bullet point in contributions: Due to the efficiency of the negative curvature finding for finding the most negative curvature direction near a saddle. We further study -> …, we further study (currently, the first sentence is not complete)

- - Beginning of Page 4: By the property Hessian Lipschitz -> By the Hessian-Lipschitz property

- - Middle of Page 5: The second part of the Algorithm 1 is the Nesterov’s accelerated gradient descent steps with its gradients estimated by 2 -> … with its gradients estimated by Eq. (2) (otherwise it reads like the gradient is estimated by the value 2)

- - Page 5, after Eq. (6): Specifically, when 6 doesn’t hold -> Specifically, when Eq. (6) does not hold (in formal English writing, abbreviations are supposed to be open). Same change shall be applied to the statement of Lemma 3: when 6 does not holds -> when Eq. (6) does not hold

In addition, I also have a couple of questions for the authors:

- It would be helpful to give detailed comparison between Algorithm 1 and Algorithm 3. Algorithm 1 gives slightly worse query complexity results (extra poly-log d overhead); so why is this of general interest over Algorithm 3?

- In Eq. (2) in Section 2.3, a central difference coordinate-wise gradient estimator is applied. In fact, in numerical analysis, higher-order central difference formulae have been studied, see for instance Li https://www.sciencedirect.com/science/article/pii/S0377042704006454?via%3Dihub It would be of general interest to discuss whether using a higher-order central difference formula could further improve the results.

**Summary Of The Paper:**

This paper studies the problem of escaping from saddle points, a basic problem in nonconvex optimization. Specifically, this paper considers function evaluations, i.e., zero-order inputs, and is able to give algorithms with ~O(1/eps^1.75) iteration complexity and ~O(d/eps^1.75) query complexity. Although this 1/eps^1.75 dependence has been achieved by gradient-descent based methods, this paper is first one to achieve the same dependence for gradient-free methods. Finally, numerical experiments are conducted to corroborate the theory results.

**Summary Of The Review:**

Overall, I think this is a novel and outstanding work in studying zeroth-order methods for escaping from saddle points, and I’m happy to recommend acceptance for ICLR 2023. Nevertheless, the paper still has space to improve as mentioned above.

---

> ### Author Response · Authors · 2022-11-17
> **Response to Reviewer kuK4**
>
> **Q1: It would be very helpful if the authors can illustrate more on the practical cases.**
>
> On the one hand, saddle points are ubiquitous in high-dimensional non-convex optimization problems, especially those related to deep neural networks. On the other hand, most real world systems do not release their internal configurations (including network structure and weights), and one can only access the input and out of such black-box models.  Take black-box adversarial attack on deep neural networks as an example. Since we do not know the explicit structure and weights of the DNN model, while we would like to maximize the loss regarding with imperceptible perturbation for a clean sample. In this case, the explicit derivative of the attack loss with respect to the perturbation cannot be accessed. However, we can use the zeroth order oracle (the objective function value $f(x)$ at any $x$, i.e., the perturbation value of the input and the attack loss of the output.) to estimate the gradient and then optimize the model.
>
> **Q2: I can already spot the following glitches.**
>
> Thanks for pointing these glitches out, we have fixed all of these glitches in the revised version.
>
> **Q3: It would be helpful to give detailed comparison between Algorithm 1 and Algorithm 3.**
>
> The main difference between Algorithm 1 and Algorithm 3 is the way in which random perturbations are added. Specifically, in Algorithm 1, we add a uniform random perturbation nearby a first-order stationary point. If it is a saddle point, then by running the zeroth-order accelerated gradient descent for $\mathscr{T} = \sqrt{\kappa} \chi c = \tilde{\Theta}(\sqrt{\kappa})$ steps, the value of Hamiltonian function will decrease by $\mathscr{E}:=\sqrt{\frac{\epsilon^3}{\rho}}\chi^{-5}c^{-7} =\tilde{\Theta}(\sqrt{\frac{\epsilon^3}{\rho}})$. In Algorithm 3, the perturbation is added along an approximate negative curvature direction, which is obtained by running $\mathscr{T}' = 32\sqrt{\kappa} \log(\frac{\ell \sqrt{d}}{\delta_0 \sqrt{\rho \epsilon}})$ steps of zeroth-order accelerated negative curvature finding (Line 11-13). Then moving along the negative curvature direction, the value of Hamiltonian function will decrease by $\frac{1}{384}\sqrt{\frac{\epsilon^3}{\rho}}$ (no more log term as in $\mathscr{E}$). We have added the above content in Remark 5 of the revised version.
>
> **Q4: It would be of general interest to discuss whether using a higher-order central difference formula could further improve the results.**
>
> Thanks for pointing this point out. After reading the paper you cited, I know that higher-order central difference formula of order $2n$ can approximate the gradient with approximation error up to $O(\mu^{2n})$. However, this requires both higher-order differentiability of the objective function and higher function query complexity. On the other hand, zeroth-order optimization algorithms which uses zeroth-order oracle to approximate the gradient has comparable convergence rate to their first-order counterparts, which is generally independent of what kinds of zeroth-order gradient estimators are used.

---

### Official Review · Reviewer_sdTo · 2022-10-22

**Confidence:** 4
**Correctness:** 3
**Technical Novelty And Significance:** 3
**Empirical Novelty And Significance:** 3
**Recommendation:** 8

**Clarity, Quality, Novelty And Reproducibility:**

Clarity: The contribution looks clear and brief. This paper is generally clear and well organized. Some points need to be clarified as shown below.

(1) What do $c$ and $\Delta_f$ mean at the end of page 4? Though $\Delta_f$ is defined in Theorem 1, it is better to define it at its first appearance.

(2) At the beginning of Lemma 6, should the second $\widehat{\nabla} f(\mathbf{x}_t)$ be $\nabla f(\mathbf{x}_t)$?

(3) Just double check, Do Lemmas 3-5 hold almost surely instead of high probability?

(4) In Theorem 1, by "one of the iterates will be an $\epsilon$-approximate SOSP", do you mean $\mathbf{x}_t$ or $\mathbf{y}_t$? The lemmas seems to imply $\mathbf{x}_t$ but it is better to clarify it in Theorems 1 and 2.

(5) In Lemma 7, $\lambda_{\min}(\nabla^2 f(\mathbf{x}_t))\le ??$.

(6) In Algorithms 1 and 3, variables like $\mathbf{x}_t$, $\tilde{\mathbf{x}}$ have been assigned multiple times. Is it better to use $\leftarrow$ instead of $=$ for such variables, which includes all the assignment statements in Algorithms 1 and 3?

Also, do $\mathbf{x}_t$ in Theorem 1 and Lemmas 3-6 refer to its value right after implementing line 4, 6 or 9 of Algorithm 1? Similar issue exists in Section 3.2. You may clarify it in these Lemmas and theorms, or add variables such that each variable is assigned only once?

(7) In Line 15 of Algorithm 3, should the first $\tilde{\mathbf{x}}$ be $(\tilde{\mathbf{x}}, \tilde{\mathbf{x}})$ to match dimensionality?

(8) In Theorem 2, you could substitute the value of $\delta_0$ into the complexity, since $\delta_0$ relies on $\epsilon, \delta, \rho$.

(9) The complexities in Theorems 1 and 2 match oracle complexity in the abstract, but the iteration complexity is not found in the theorems. Could you add them? Also, you may say "The total number of function queries (oracle complexity)" in the theorems to be consistent with abstract.

(10) I did not find out the complexity value expressed in $\mathcal{O}$ in the proof of Theorem 1. In the proof of Theorem 2, $T=\widetilde{O}\big(\frac{\Delta_f}{\epsilon^{1.75}}\log d\big)$ does not match the complexity in Theorem 2. Could you explain that? Usually when proving complexity, it is recommended to express the complexity in terms of hyperparameters and then substitute the hyperparameter values.


Quality:

This work is complete and generally solid. However, the claim in the abstract that "our methods achieve a comparable convergence rate to their first-order counterparts and have fewer oracle complexity compared to prior derivative-free methods for finding second-order stationary points." is not well supported, unless you compare all these complexities in terms of $\mathcal{O}$, either in the introduction or after Theorems 1 and 2.

In the experiment, why is RSPI not implemented in Figure 1?


Novelty:  This work gives positive answer to a novel research problem: can Nesterov's momentum accelerate the convergence of gradient-free methods for finding second-order stationary point?


Reproducibility: To ensure reproducibility of the experiments, it is better to tell the values used for all the inputs (hyperparameter choices). For example, what are the candidate values in the coarse grid search for $\ell$ and $\rho$?

Minor comments:

(1) In the abstract, I think "smaller oracle complexity" is better than "fewer oracle complexity".

(2) At the beginning of Section 3.1, use "three parts" instead of "three part".

(3) Use bolded $\mathbf{x}_t$ in line 5 of Algorithm 1.

(4) Line 8-9 of Algorithm 1 could be moved to right before line 6.

(5) In and right after Lemma 3, you could use "eq. (6)" instead of "(6)". This also applies to equation citations elsewhere.

(6) You could use big { } in Lemma 4. Similarly, use big ( ) on the left side of the centered equation in Lemma 8.

(7) In Theorem 1 and 2, "query" could be changed into "queries".

(8) A typo in Theorem 2: "set set".

(9) You could also cite
Zhang, H., Xiong, H., & Gu, B. (2022). Zeroth-Order Negative Curvature Finding: Escaping Saddle Points without Gradients. arXiv preprint arXiv:2210.01496.

**Strength And Weaknesses:**

Pros: This paper studies a significant novel research problem. Theorems and experiments look solid. The presentation looks generally clear, brief and neat.

Cons: There are some points that need to be clarified, especially those about complexity results, See "Clarity, Quality, Novelty And Reproducibility" for detail.

**Summary Of The Paper:**

The authors provide to their knowledge the first two variants of Nesterov's accelerated gradient-free algorithms for escaping saddle points of minimization problem, with comparable convergence rate to their first-order counterparts and smaller oracle complexity compared to prior derivative-free methods for finding second-order stationary points.


**Summary Of The Review:**

Based on "Strength And Weaknesses" above, I recommend borderline accept for this paper. After my major concerns above about the complexities are solved by the authors, I would like to raise my rating.

---

> ### Author Response · Authors · 2022-11-17
> **Response to Reviewer sdTo**
>
> Dear Reviewer, thank you for taking the time to review our submission. We sincerely appreciate your valuable comments. Please find detailed responses to your questions below.
>
> **Re Clarity:**
>
> **(1)** $c$ is a constant and $f(x_0) - f(x^*) \le \Delta_f < \infty$ is a basic assumption in nonconvex optimization. In the revised version, we have defined it at its first appearance.
>
> **(2)** Yes, it should be $\nabla f(x_t)$ here. In the revised version, we have fixed this typo.
>
> **(3)** Lemmas 3-5 holds surely. This is because we consider the deterministic optimization in this paper and the zeroth-order gradient estimator is also deterministic. Hence, no randomness should be considered in Lemmas 3-5.
>
> **(4)** It means $x_t$. In the revised version, we have clarified it in Theorems 1 and 2.
>
> **(5)** It is $\lambda_{\min} (\nabla^2 f(x_t)) \le -\sqrt{\rho \epsilon}$. In the revised version, we have added it.
>
> **(6)** Thanks for your suggestions, in the revised version, we changed it to $\gets$. $x_t$ in Theorem 1 and Lemmas 3-6 always refer to its value after the $t$-th iteration of Algorithm 1. Specifically, Lemma 3 and 4 are one iteration analysis of Algorithm 1 under the condition that Eq. (6) doesn't hold and holds, respectively. Lemmas 5 and 6 are descent lemmas of $\mathscr{T}$ iterations. All preconditions of these lemmas has already discussed before the corresponding lemmas. In the revised version, we will make it more clear.
>
> **(7)** Thanks for pointing this out. Yes, it would be better to write it as $(\tilde{x}, \tilde{x})$.
>
> **(8)** In the revised version, we substituted the value of $\delta_0$ into the complexity.
>
> **(9)** Yes, in the revised version, we will add the iteration complexity in the theorems.
>
> **(10)** On the main claim of the proof of Theorem 1 is that ``the total steps is no more than $\frac{2 \mathscr{T}\Delta_f}{\mathscr{E}}$''. By substituting the values of $\mathscr{T}, \mathscr{E}$ we get the total number of iterations is $\mathcal{O}\left(\frac{\Delta_f \ell^{1/2} \rho^{1/4}}{\epsilon^{7/4}} \log^6(\frac{d\ell\Delta_f}{\rho\epsilon\delta}) \right)$. Since each iteration needs $\mathcal{O}(d)$ function queries, the total oracle complexity is $\mathcal{O}\left(\frac{ d \Delta_f \ell^{1/2} \rho^{1/4}}{\epsilon^{7/4}} \log^6(\frac{d\ell\Delta_f}{\rho\epsilon\delta}) \right)$. In proof of Theorem 2, notation $\tilde{\mathcal{O}}$ hides the log dependent on $\epsilon$. In the revised version, we have rewritten it as $\mathcal{O}\left(\frac{\Delta_f \ell^{1/2} \rho^{1/4}}{\epsilon^{7/4}} \log(\frac{\ell \sqrt{d} \Delta_f}{ \delta \epsilon^2 })\right)$ by substituting the corresponding hyperparameter values.
>
> **Re Quality:** In table 1, we have compared our complexity results with prior zeroth-order methods for finding second-order stationary points. We also discussed these complexities in the third paragraph of the introduction.
>
> **Re ``In the experiment, why is RSPI not implemented in Figure 1?''** Actually, RSPI is not a single loop algorithm but a nested loop algorithm. In Figure 1, we just simply compared our algorithms with a single loop algorithm PAGD and showed the efficiency in escaping from saddle points. Considering that RSPI is also an important algorithm for escaping saddle points using only the zeroth-order information, we compared it with other algorithms in the second experiment. In the revised version, we also added a new nested loop algorithm ZO-GD-NCF you mentioned for comparison in the second experiment.
>
> **Re Reproducibility:** Thanks for pointing this out, in the revised version, we added an additional section at the end of the appendix to detail the values used for all the hyperparameters. The grid search region of $\ell$ and $\rho$ has already been added in the section of numerical experiments. The experimental code has been attached to the supplementary material of our first submission.
>
> **Re Minor comments:** We have fixed all the minor issues you mentioned in the revised version and also cited the Zhang, H., Xiong, H., \& Gu, B. (2022). Zeroth-Order Negative Curvature Finding: Escaping Saddle Points without Gradients. arXiv preprint arXiv:2210.01496.

---

> > ### Comment · Reviewer_sdTo · 2022-11-18
> > **Reviewer sdTo's 2nd reply**
> >
> > Thanks the authors for their reply and revision which well solves all my questions, so I increase the score to 8.

---

### Official Review · Reviewer_Y99G · 2022-10-26

**Confidence:** 3
**Correctness:** 3
**Technical Novelty And Significance:** 3
**Empirical Novelty And Significance:** 3
**Recommendation:** 8

**Clarity, Quality, Novelty And Reproducibility:**

This paper is well-written, and the techniques are effective and well-supported by the empirical results.

**Strength And Weaknesses:**

Strength

Overall, the paper is well written and the related work are well cited. The introduction is clearly organized. The derived convergence results match the state-of-the-art one and are faster than previous zeroth-order algorithms for escaping saddle points. Numerical results illustrate the effectiveness of the proposed algorithms.

Weaknesses

1. Since the work focused only on the deterministic optimization, this may restrict the application scenarios of the algorithm. It'd be nice if the authors have further insights for stochastic  optimization as well.

2. The algorithms use $\mathcal{O}(d)$ function evaluations per iteration to enable a high accuracy approximation of the gradient,  which may also not
be practical in realistic applications when the dimension of the problem is large. I wonder if such a gradient estimator is necessary?

3. Since the algorithms are similar to the Jin et al. and Zhang et al., it would be helpful to explain what novel theoretical analysis techniques are used in this work.

4. In theoretical results, the value of parameter $\mu$ is set to be very small, is it the same in the experiment? Since in practice, too mall $\mu$ may cause system error. It would be better for the authors to explain how to choose the parameter u in the experiment.

5. Minor issues:

it seems that $\mathcal{H}_f$ only appear in section 2.4, where is it used?

Algorithm 2 is from Jin et al. 2018, you should add a cite here or delete it and add a footnote.

**Summary Of The Paper:**

This work proposes two zeroth-order Nesterov’s accelerated gradient descent based algorithms that can escape saddle points and converge to second order stationary points. The main idea of the work is to use the central finite difference gradient estimator to approximate the gradient. The theoretical results show that the proposed can converge to second-order stationary points with fewer function queries.


**Summary Of The Review:**

Overall, this is a good paper, and I would recommend an "accept".

---

> ### Author Response · Authors · 2022-11-17
> **Response to Reviewer Y99G**
>
> Dear Reviewer, thank you for taking the time to review our submission. We sincerely appreciate your valuable comments. Please find detailed responses to your questions below.
>
> **Q1:  It'd be nice if the authors have further insights for stochastic optimization as well**
>
> Yes, in this paper we only consider the deterministic optimization. In the stochastic setting, things may be more complicated since we only have access to the stochastic zeroth-order oracle. To the best of our knowledge, some work of this line has been studied in https://arxiv.org/abs/2210.01496. Recently, an accelerated zeroth-order momentum method was proposed that can convergence to a first-order stationary point with smaller query complexity https://www.jmlr.org/papers/volume23/20-924/20-924.pdf. It is then interesting to explore if such momentum-based zeroth-order method combined with the negative curvature finding subroutine can also find a second-order stationary point with smaller query complexity.
>
> **Q2: I wonder if such a gradient estimator is necessary?**
>
> Yes, using $O(d)$ function queries to estimate the gradient may not be practical in realistic applications. Thus, it would be interesting to study the momentum-based zeroth-order algorithms with two-point zeroth-order gradient estimator for converging to a second-order stationary point. We will explore it in the future work.
>
> **Q3: it would be helpful to explain what novel theoretical analysis techniques are used in this work.**
>
> The main technique is to analyse the error term of induced by the zeroth-order oracle and proper smoothing parameter choice in theory. Specifically, we analyse the properties of zeroth-order gradient estimators  and  under the Hessian Lipschitz assumption, which is essential for studying the second-order convergence properties with zeroth-order methods. Also, we study the approximation error of the zeroth-order Hessian-vector product under the Hessian Lipschitz assumption, which is essential for analysing the zeroth-order negative curvature finding subroutine.
>
> **Q4: It would be better for the authors to explain how to choose the parameter $\mu$ in the experiment.**
>
> That is indeed an important point: yes, indeed, in theory, $\mu$ should be taken as small as possible to reduce the error term. However, it is not possible to do so in practice, due to numerical precision errors. Therefore, $\mu$ should be tuned empirically, so as to obtain the best results: it should be small enough so that the error term is not too large, but large enough to avoid any numerical precision errors.
>
> **Q5: Minor issues**
>
> $\mathcal{H}_f$ is the notation of the zeroth-order Hessian-vector product, which is estimated by the difference of two coordinate-wise gradient estimator and used in Line 11 of Algorithm 3.

---

> > ### Comment · Reviewer_Y99G · 2022-12-07
> > **thank you**
> >
> > I would like to thank the authors' responses and revisions. In the current version, my previous concerns have been addressed. I am positive about this paper and would recommend it for acceptance.

---

### Official Review · Reviewer_85d1 · 2022-10-31

**Confidence:** 4
**Correctness:** 4
**Technical Novelty And Significance:** 3
**Empirical Novelty And Significance:** 3
**Recommendation:** 6

**Clarity, Quality, Novelty And Reproducibility:**

Minor issues:
-- The title of Table 1 should be changed
-- Page 6: “Hessian-vector product estimator in 3.” -> “Equation (3)”.


**Strength And Weaknesses:**

My main concern about the paper is the lack of explanations:
-- The algorithms should be explained in more details. I’m familiar with Jin et al’18, so I had a good idea what happens in Algorithms 1 and 2. However, while I could parse Algorithm 3, I can’t say I understood what happens there.
-- Similarly, it would be great to explain the choice of parameters (I mean, why their values make sense), especially μ in Theorem 1.
-- Proof outline for Section 3 should be added.


**Summary Of The Paper:**

The paper shows convergence to an ε-second-order stationary point for zeroth-order methods. They substantially improve existing bounds, achieving ε^-7/4 dependence on ε and quasi-linear dependence on problem dimension. They present two algorithms: the first algorithm and its analysis are based on the Jin et al’18. The second algorithm, based on the Hessian-vector product idea, substantially improves the logarithmic factors. One of the main component in their analysis is better estimation of discrepancy between the true gradient and its approximation, based on the Lipschitz Hessian property.



**Summary Of The Review:**

Solid paper, accept.

---

> ### Author Response · Authors · 2022-11-17
> **Response to Reviewer 85d1**
>
> Dear Reviewer, thank you for taking the time to review our submission. We sincerely appreciate your valuable comments. Please find detailed responses to your questions below.
>
> **Q1: The algorithms should be explained in more details.**
>
> Thanks for pointing this out. In the revised version, we have added more details of the difference between Algorithm 1 and Algorithm 3 in Remark 5.
>
> **Q2: Similarly, it would be great to explain the choice of parameters (I mean, why their values make sense), especially $\mu$ in Theorem 1.**
>
> That is indeed an important point: yes, indeed, in theory, $\mu$ should be taken as small as possible to reduce the error term. In the theorem 1, parameters $\mu$ in different lines of Algorithm 1 are given different values to ensure that the approximation error is not too large and the convergence of the algorithm. Specially, we set $\mu = \tilde{\mathcal{O}}(\frac{\epsilon^{1/2}}{d^{1/4}})$ in Line 3 to satisfy the pre-condition of Proposition 1; we set $\mu = \tilde{\mathcal{O}}(\frac{\epsilon^{13/8}}{d^{1/2}})$ in Line 6 such that the approximation error will not increase too quickly during the AGD process; we set $\mu = \tilde{\mathcal{O}}(\frac{\epsilon^{1/2}}{d^{1/4}})$ in Line 8 to ensure the performance of the Negative Curvature Exploitation step. All of the above values of $\mu$ are based on specific theoretical analysis.  In practice, due to the existence of the numerical precision errors,  $\mu$ should be tuned empirically so as to obtain the best results: it should be small enough so that the error term is not too large, but large enough to avoid the numerical precision errors.
>
> **Q3: Proof outline for Section 3 should be added.**
>
> Thanks for pointing this out. In the revised version, we have added some proof outlines in Section 3.
>
> **Q4: Minor issues**
>
> In the revised version, we have fixed all the minor issues you mentioned. Thanks for pointing them out.

---

### Decision · Program_Chairs · 2023-01-20

**Decision:**

Accept: notable-top-25%

**Justification For Why Not Higher Score:**

It's somewhere between a spotlight and oral, I'm not sure which way it should go.

**Justification For Why Not Lower Score:**

The paper gets almost tight guarantees for gradient free nonconvex optimization I think it deserves at least a spotlight.

**Metareview: Summary, Strengths And Weaknesses:**

This paper gives efficient gradient free algorithms for finding approximate second order stationary points. The running time and query complexity bounds are very strong and they significantly improved over previous results. The proof technique is also novel and interesting. Reviewers agreed that this is a strong result, with some suggestions on clarify and presentation. The author response has addressed most of the concerns. Overall this is a solid result in nonconvex optimization.

**Note From Pc:**

if the above contains the word "oral" or "spotlight" please see: "oral" presentation means -> notable-top-5% and "spotlight" means -> notable-top-25%. As stated in our emails, we are disassociating presentation type from AC recommendations